# DISTRIBUTED ESTIMATION OF SPARSE COVARIANCE MATRIX UNDER HEAVY-TAILED DATA

## ABSTRACT

In this paper, we study high-dimensional covariance matrix estimation over a network of interconnected agents, where the data are distributed and may exhibit heavy-tailed behavior. To address this challenge, we propose a new estimator that integrates the Huber loss to mitigate outliers with a non-convex regularizer to promote sparsity. To the best of our knowledge, this is the first framework that simultaneously accounts for high dimensionality, heavy tails, and distributed data in covariance estimation. We begin by analyzing a proximal gradient descent algorithm to solve this non-convex and non-globally Lipschitz smooth problem in the centralized setting to set the stage for the distributed case. In the distributed setting, where bandwidth, storage, and privacy constraints preclude agents from directly sharing raw data, we design a decentralized algorithm aligned with the centralized one, building on the principle of gradient tracking. We prove that, under mild conditions, both algorithms converge linearly to the same solution. Moreover, we establish that the resulting covariance estimates attain the oracle statistical rate in Frobenius norm, representing the state of the art for high-dimensional covariance estimation under heavy-tailed distributions. Numerical experiments corroborate our theoretical findings and demonstrate that the proposed estimator outperforms existing baselines in both estimation accuracy and robustness.

## 1 INTRODUCTION

Covariance matrix estimation is a fundamental problem in multivariate data analysis, with wide-ranging applications in fields such as machine learning (Jolliffe, 2002), biology (Schäfer & Strimmer, 2005), and finance (Markowitz, 1952). In many practical scenarios, the dimension of the covariance matrix can far exceed the number of samples. For example, in functional genomics, gene expression microarray studies often involve estimating a covariance matrix for tens of thousands of genes based on only a few hundred samples (Schäfer & Strimmer, 2005). However, in high-dimensional settings where $d/N \to \infty$, the sample covariance matrix performs poorly, leading to significant challenges in downstream tasks (Bai & Yin, 2008). Consequently, accurate estimation of covariance matrices in high dimensions has become an active and important area of research.

A widely adopted assumption in high-dimensional covariance matrix estimation is sparsity, i.e., most off-diagonal entries are close to zero, which significantly reduces the number of parameters to estimate (Bickel & Levina, 2008). This assumption also enhances interpretability in many applications. For example, in portfolio analysis, once common factors are removed, stocks from unrelated sectors typically exhibit near-zero marginal correlations (Fan et al., 2013). Sparse covariance estimation has been extensively studied; see Pourahmadi (2013); Fan et al. (2016); Lam (2020); Wei & Zhao (2023) for comprehensive overviews. Among existing methods, applying an $\ell_1$ penalty to promote sparsity is well established and achieves the minimax optimal statistical rate (Xue et al., 2012; Rothman, 2012). However, while the $\ell_1$ penalty encourages sparsity, it introduces estimation bias (Fan & Li, 2001; Fan et al., 2016). To address this issue, non-convex penalties such as smoothly clipped absolute deviation (SCAD) (Fan & Li, 2001) and minimax concave penalty (MCP) (Zhang, 2010) have been proposed. These methods preserve the sparsity-inducing effect of the $\ell_1$ penalty for small coefficients while reducing shrinkage on large coefficients, thereby mitigating bias. Recently, Wei & Zhao (2023) applies non-convex penalties to sparse covariance estimation and shows that, under (sub-)Gaussian assumptions, the resulting estimator attains the oracle rate.

In many modern applications, data are collected independently by multiple agents, such as geographically dispersed sensors, satellites in different orbits, or institutions across continents (Xia et al., 2025). Due to communication and storage overhead, privacy concerns, and regulatory constraints, transmitting distributed data to a central processor can be inefficient or even infeasible (Bertsekas & Tsitsiklis, 2015; Boyd et al., 2011; Nedić et al., 2018). For example, agents may be unwilling to share private data in collaborative research among laboratories (Forero et al., 2010), and low-power devices are often restricted to communicate only with physically nearby neighbors in wireless networks (Predd et al., 2009). These challenges underscore the need for distributed estimation methods that enable network-wide analysis while preserving data locality and privacy (Maros & Scutari, 2022; Ji et al., 2023; Xia et al., 2025). Moreover, many practical datasets may follow heavy-tailed distributions or contain outliers. For instance, return distributions in finance usually exhibit power-law behavior (Cont, 2001), and measurement limitations, e.g., in biological imaging, often lead to heavy-tailed noise (Fan et al., 2021). Consequently, numerous robust estimation methods have been developed to address these challenges (Huber & Ronchetti, 2011; Maronna et al., 2019). A representative strategy treats each entry of the covariance matrix as a location parameter and estimates it using robust techniques such as Huber's M-estimator (Huber, 1964). Compared to the traditional squared loss, the Huber loss adopts a quadratic form for small residuals and a linear form for large residuals, thereby limiting the influence of outliers while maintaining convexity and smoothness.

To address the aforementioned challenges, we study robust sparse covariance matrix estimation in a distributed setting, where data samples are spread across networked agents and may exhibit heavy-tailed behavior. Specifically, our main contributions are as follows:

- **New problem formulation.** We propose a novel covariance matrix estimation problem as the minimization of a Huber loss combined with a log-determinant barrier and a non-convex penalty, which is the first framework that simultaneously handles the high-dimensional, heavy-tailed, and distributed setting in covariance matrix estimation. The resulting problem is non-convex and non-globally Lipschitz smooth, and the lack of access to data from other agents further complicates the estimation task in the distributed setting.

- **Algorithmic design and convergence analysis.** We develop both centralized and decentralized single-loop algorithms for the proposed problem. In the centralized case, we introduce a proximal gradient method, which serves as the foundation for the distributed setting. In the decentralized case, we integrate proximal gradient updates with the gradient tracking scheme (Di Lorenzo & Scutari, 2016). We show that, under mild conditions, both algorithms converge linearly to the same estimate, and that the convergence rate of the decentralized algorithm differs from its centralized counterpart only by constant factors.

- **Sharp statistical guarantees.** We prove that both the decentralized and centralized algorithms attain an oracle statistical rate $O(\sqrt{s/N})$ in Frobenius norm ($s$ is the sparsity level and $N$ is the sample size), which is the state-of-the-art statistical guarantee for high-dimensional covariance estimation under heavy-tailed distributions. We validate the theoretical findings through numerical experiments, demonstrating that our method outperforms baseline methods in terms of both estimation accuracy and robustness.

## 2 RELATED WORK

**Covariance estimation under heavy tails.** In the work of Fan et al. (2016), each entry of the covariance matrix is estimated using Huber's M-estimator under the assumption of zero-mean data. For the unknown-mean case, Avella-Medina et al. (2018) apply Huber's M-estimator elementwise to estimate the first and second moments separately and then combines them to construct a robust covariance estimator. To avoid the potential accumulation of errors introduced by separately estimating and combining moments, Ke et al. (2019) propose using pairwise differences to eliminate the mean effect directly. Under the assumption of finite fourth moments, these Huber-loss-based estimators are shown to achieve optimal deviation bounds. Beyond Huber-based approaches, several other elementwise estimators have been developed based on robust location estimation (e.g., truncation (Ke et al., 2019), median-of-means (Avella-Medina et al., 2018; Ke et al., 2019)), robust scale estimation (e.g., median absolute deviation (Gnanadesikan & Kettenring, 1972), interquartile range (Lu et al., 2021)), and combined first- and second-moment estimation (e.g., rank-based methods (Liu et al., 2012; Xue & Zou, 2012; Avella-Medina et al., 2018)). Furthermore, M–estimators based on

the Mahalanobis distance with different robust functions (e.g., Tyler's M–estimator (Tyler, 1987), Maronna's M–estimator (Maronna, 1976), the minimum volume ellipsoid estimator (Rousseeuw, 1985), the minimum covariance determinant estimator (Rousseeuw & Driessen, 1999), S–estimator (Davies, 1987)), and estimators combining projection pursuit with one-dimensional robust scale estimation (Donoho, 1982; Li & Chen, 1985) have also been considered.

**Sparse covariance estimation under heavy tails.** Existing robust sparse covariance matrix estimators are generally based on two-stage procedures: they first construct a robust covariance estimate and then enforce sparsity through hard or soft thresholding. For example, Avella-Medina et al. (2018) employ Huber's M-estimator, the mean-of-medians, and rank-based methods to obtain robust covariance estimates, followed by hard thresholding to induce sparsity, and subsequently projects the result onto the positive semidefinite cone. Under the finite fourth-moments assumption, both Huber's M-estimator and rank-based methods achieve the minimax rate, while the mean-of-medians further requires finite sixth moments. For Huber's M-estimator, Li et al. (2023) introduce soft thresholding combined with a positive-definiteness constraint to simultaneously promote sparsity and ensure positive definiteness, and further establishes support recovery rates and sign consistency for compositional data. The work of Goes et al. (2020) combines Tyler's M-estimator with hard thresholding and achieves the minimax rate under elliptical distribution assumptions. Similarly, Lu et al. (2021) use the interquartile range and soft thresholding to construct a robust sparse covariance estimator and prove the minimax rate under pair-elliptical models. In contrast to these two-stage procedures, Chen et al. (2018) propose a one-step estimator based on maximizing matrix depth, which simultaneously achieves robustness and sparsity; however, this approach currently lacks a polynomial-time algorithm for practical implementation. Notably, none of the existing robust sparse covariance estimation methods have been shown to attain the oracle statistical rate $O(\sqrt{s/N})$.

**Distributed covariance estimation.** Research on distributed covariance matrix estimation remains limited, with most existing efforts focused on precision (inverse covariance) matrix estimation. When a central server is available, divide-and-conquer approaches have been applied to the debiased graphical lasso (Nezakati & Pircalabelu, 2023), debiased CLIME (Xu et al., 2016), and D-trace loss penalized lasso (Wang & Cui, 2021) to obtain high-dimensional precision matrix estimates. In particular, Xu et al. (2016) and Wang & Cui (2021) incorporate rank-based methods to derive robust covariance estimates with favorable statistical properties under the trans-elliptical model and a finite number of agents. The work of Dong & Liu (2024) removes the constraint on the number of agents by iteratively solving the D-trace loss penalized lasso through communication with a central server. However, a common limitation of these methods is the lack of guarantees on positive definiteness, which can lead to invalid covariance estimates. To address this, Xia et al. (2025) propose a decentralized graphical lasso algorithm that eliminates the need for a central processor and ensures both linear convergence and positive definiteness. Nonetheless, this method relies on (sub-)Gaussian assumptions and does not guarantee sparsity in the estimated covariance matrix. Xia et al. (2024) directly impose an $\ell_1$ penalty on the covariance matrix under the Gaussian maximum likelihood framework. However, this method still relies on the (sub-)Gaussian assumption, achieves only sublinear convergence, and lacks statistical guarantees.

## 3 PROBLEM FORMULATION

We study the estimation problem where data are distributed across a network of $m$ agents. The network is modeled as a time-invariant undirected graph $\mathcal{G} = (\mathcal{V}, \mathcal{E})$, where $\mathcal{V} = \{1, \ldots, m\}$ denotes the set of agents and $\mathcal{E}$ the set of communication links. For each agent $i$, its neighborhood is defined as $\mathcal{N}_i = \{j \mid (i,j) \in \mathcal{E}\} \cup \{i\}$. The interactions among agents are encoded in a weight matrix $\mathbf{W} \in \mathbb{R}^{m \times m}$. We impose the following assumptions on the network.

**Assumption 1** (Nedić et al. (2018))**.** *The network $\mathcal{G}$ and the weight matrix $\mathbf{W}$ satisfy the following conditions: (a) $\mathcal{G}$ is connected; (b) $\mathbf{W}$ is compliance with $\mathcal{G}$, i.e., $W_{ij} > 0$ if $j \in \mathcal{N}_i$, otherwise $W_{ij} = 0$; (c) $\mathbf{W}$ is doubly stochastic, i.e., $\mathbf{W} = \mathbf{W}^\top$ and $\mathbf{W}\mathbf{1} = \mathbf{1}$.*

Assumption 1 implies that each agent is only allowed to exchange information with its neighbors, which is standard in the distributed optimization literature and widely adopted in practical applications. See Cattivelli & Sayed (2009) for representative examples of weight matrices $\mathbf{W}$ that satisfy this assumption. Note that our network setting is allowed to be fully decentralized, as Assumption 1

does not require the presence of a central coordinator. Let $\rho = \|\mathbf{W} - \mathbf{J}\|_2$, where $\mathbf{J} = \frac{1}{m}\mathbf{1}\mathbf{1}^\top$. Assumption 1 guarantees that $\rho \in [0, 1)$ (Sun et al., 2022a). The parameter $\rho$ characterizes the connectivity of the network $\mathcal{G}$: as $\rho \to 0$, the graph $\mathcal{G}$ becomes increasingly connected; as $\rho \to 1$, it approaches a disconnected topology.

Let $\mathbf{x}$ be a zero-mean $d$-dimensional random vector. We assume that each component $x_k$, for $k = 1, \ldots, d$, possesses a finite fourth moment. This assumption, which is standard in robust covariance matrix estimation (Rothman et al., 2009; Avella-Medina et al., 2018; Ke et al., 2019), accommodates heavy-tailed distributions of $\mathbf{x}$ and is formally stated below.

**Assumption 2.** *For all $k = 1, \ldots, d$, we have* $\mathrm{E}\left(|x_k|^{4(1+\nu)}\right) \leq \sigma^{2(1+\nu)} < +\infty$*, where $\sigma > 0$ provides a uniform bound on moments, and $\nu > 0$ determines the order of the moment that is finite.*

A commonly used loss function in robust estimation is the Huber loss, formally defined as follows.

**Definition 1** (Huber loss (Huber, 1964))**.** The Huber loss function, denoted as $h$, is defined as

$$h\left(x\right) = \begin{cases} \frac{1}{2}x^2, & |x| \leq a, \\ a\left|x\right| - \frac{1}{2}a^2, & |x| > a, \end{cases} \tag{1}$$

where $a > 0$ is the robustification parameter.

In contrast to the conventional squared loss, the Huber loss down-weights extreme outliers, thereby offering enhanced robustness. Notably, the Huber loss mediates between the squared and absolute error loss functions: in the limit as $a \to +\infty$, the Huber loss recovers the squared loss, whereas as $a \to 0$, it approximates the absolute loss.

Let $\{\mathbf{x}_j\}_{j=1}^N$ be $N$ independent and identically distributed observations that are stored over $m$ agents. We assume the total sample size $N$ is much smaller than the feature dimension $d$ in the high-dimensional estimation regime. Without loss of generality, we assume that each agent $i \in \{1, \ldots, m\}$ holds a local subset $\{\mathbf{x}_j\}_{j \in \mathcal{J}_i}$ of $n$ observations, where the index sets $\{\mathcal{J}_i\}_{i=1}^m$ are disjoint and satisfy $\bigcup_{i=1}^m \mathcal{J}_i = \{1, 2, \ldots, N\}$ and $|\mathcal{J}_i| = n$, so that $N = mn$. We formulate the following optimization problem for distributed sparse covariance matrix estimation:

$$\underset{\mathbf{\Sigma} \succeq \mathbf{0}}{\text{minimize}} \quad \frac{1}{m}\sum_{i=1}^m \mathcal{H}_i\left(\mathbf{\Sigma}\right) - \tau \log\det\left(\mathbf{\Sigma}\right) + \mathcal{P}\left(\mathbf{\Sigma}\right), \tag{2}$$

where

$$\mathcal{H}_i\left(\mathbf{\Sigma}\right) = \frac{1}{n}\sum_{j \in \mathcal{J}_i}\sum_{k=1}^d\sum_{l=1}^d h\left(\Sigma_{kl} - x_{jk}x_{jl}\right) \text{ and } \mathcal{P}\left(\mathbf{\Sigma}\right) = \sum_{k=1}^d\sum_{l=1,l\neq k}^d p_\lambda\left(\Sigma_{kl}\right). \tag{3}$$

$\mathcal{H}_i$ is the Huber loss for agent $i$ based on its local dataset, the log-determinant barrier term, with $\tau > 0$, ensures the solution to be positive definite, and $\mathcal{P}$ introduces sparsity on the off-diagonal elements of $\mathbf{\Sigma}$ with the elementwise penalty function $p_\lambda$ being non-convex and decomposable. Problem 2 is non-convex due to the penalty $\mathcal{P}$. On the penalty function $p_\lambda$, we impose the following assumption.

**Assumption 3** (Regularity condition on $p_\lambda$)**.** *The elementwise penalty function $p_\lambda : \mathbb{R} \to \mathbb{R}$ can be decomposed as $p_\lambda\left(x\right) = \lambda\left|x\right| - q\left(x\right)$, where $q$ is a convex function. Besides, the following conditions are satisfied: (a) there exists a constant $b \geq 0$ such that $p_\lambda'\left(x\right) = 0$ for $|x| \geq b\lambda$; (b) $q$ is symmetric, i.e., $q\left(x\right) = q\left(-x\right)$ for any $x$; (c) $q$ and $q'$ pass through the origin, i.e., $q\left(0\right) = q'\left(0\right) = 0$; (d) $q'$ is bounded, i.e., $|q'\left(x\right)| \leq \lambda$ for any $x$; (e) there exist a constant $L_q \geq 0$ such that for any $x_1$ and $x_2$, we have $0 \leq \left(q'\left(x_1\right) - q'\left(x_2\right)\right)/\left(x_1 - x_2\right) \leq L_q$.*

Compared with the commonly used $\ell_1$ penalty $\lambda|x|$ (Tibshirani, 1996), the function $p_\lambda(x)$ in Assumption 3 can be viewed as the $\ell_1$ norm minus a convex function. This construction reduces the penalty on large coefficients, thereby mitigating the estimation bias of $\ell_1$, while still preserving its sparsity-inducing property. Condition (a) ensures that $p_\lambda(x)$ remains constant for $|x| \geq b\lambda$, and condition (e) regulates the curvature of the convex function $q$ through the parameter $L_q$. A number of popular sparse regularizers satisfy Assumption 3, including the SCAD penalty (Fan & Li, 2001) and MCP (Zhang, 2010).

Since the dataset is stored locally, agent $i$ can only access its own loss function $\mathcal{H}_i$ and thus cannot solve 2 independently. In addition, the objective function is non-convex and non-globally Lipschitz smooth, which further increases the difficulty of optimization. In this paper, our goal is to design an algorithm that computes an estimator while avoiding exchange of local data to solve problem 2.

## 4 WARM-UP: ROBUST SPARSE COVARIANCE MATRIX ESTIMATION

Before introducing the decentralized algorithm for solving problem 2, we first present a centralized optimization algorithm under the setting where all data are aggregated on a single server, and establish its theoretical convergence guarantees, which serve as a benchmark for the distributed setting.

**Proximal gradient algorithm.** According to Assumption 3, the regularizer $\mathcal{P}(\boldsymbol{\Sigma})$ can be written as $\mathcal{P}(\boldsymbol{\Sigma}) = \lambda \|\boldsymbol{\Sigma}\|_{1,\text{off}} - \mathcal{Q}(\boldsymbol{\Sigma})$, where $\mathcal{Q}(\boldsymbol{\Sigma}) = \sum_{k=1}^{d}\sum_{l=1,l\neq k}^{d} q(\Sigma_{kl})$. Let $\mathcal{F}(\boldsymbol{\Sigma}) = \frac{1}{m}\sum_{i=1}^{m}\mathcal{H}_i(\boldsymbol{\Sigma}) - \tau\log\det(\boldsymbol{\Sigma}) - \mathcal{Q}(\boldsymbol{\Sigma})$. A standard approach for solving problem equation 2 is the proximal gradient method, as summarized in Algorithm 1. At each iteration $t$, the estimate is updated according to

---
**Algorithm 1** Robust Sparse Covariance Matrix Estimation

---
**given** $\boldsymbol{\Sigma}^{(0)} \succeq \mathbf{0}, \gamma > 0, t = 0$
**while** not converge, **do**
$\quad \boldsymbol{\Sigma}^{(t+1)} = \text{ST}_{\frac{\lambda}{\gamma}}\left(\boldsymbol{\Sigma}^{(t)} - \frac{1}{\gamma}\nabla\mathcal{F}\left(\boldsymbol{\Sigma}^{(t)}\right)\right)$
$\quad t = t+1$
**end while**
**return** $\boldsymbol{\Sigma}^{(t)}$.

---

$$\boldsymbol{\Sigma}^{(t+1)} \in \arg\min_{\boldsymbol{\Sigma} \succ \mathbf{0}}\left\{\mathcal{F}\left(\boldsymbol{\Sigma}^{(t)}\right) + \left\langle\nabla\mathcal{F}\left(\boldsymbol{\Sigma}^{(t)}\right), \boldsymbol{\Sigma} - \boldsymbol{\Sigma}^{(t)}\right\rangle + \frac{\gamma}{2}\left\|\boldsymbol{\Sigma} - \boldsymbol{\Sigma}^{(t)}\right\|_F^2 + \lambda\|\boldsymbol{\Sigma}\|_{1,\text{off}}\right\}, \quad (4)$$

where $\gamma > 0$ is the step size. The combination of the positive definiteness constraint and the $\ell_1$ penalty in equation 4 precludes a closed-form solution. By relaxing the positive definiteness requirement, the update reduces to a simple soft-thresholding step: $\boldsymbol{\Sigma}^{(t+1)} = \text{ST}_{\frac{\lambda}{\gamma}}\left(\boldsymbol{\Sigma}^{(t)} - \frac{1}{\gamma}\nabla\mathcal{F}\left(\boldsymbol{\Sigma}^{(t)}\right)\right)$, where for any matrix $\mathbf{X} \in \mathbb{R}^{d\times d}$ and threshold $\omega > 0$, $\text{ST}_\omega(\mathbf{X})$ is defined elementwise as $[\text{ST}_\omega(\mathbf{X})]_{ij} = \text{sign}(X_{ij})\max\{|X_{ij}| - \omega, 0\}$ for $i \neq j$, and $[\text{ST}_\omega(\mathbf{X})]_{ii} = X_{ii}$ for $i = j$. As we will show in the next subsection, with an appropriate choice of $\gamma$, the iterates $\boldsymbol{\Sigma}^{(t)}$ remain positive definite for all $t$. Consequently, Algorithm 1 eliminates the need for a costly optimization subroutine to solve equation 4 at each iteration.

**Convergence analysis.** We now present the convergence guarantee of Algorithm 1.

**Theorem 1** (Convergence property of Algorithm 1). *Suppose Assumption 3 holds for problem 2. Define constants $\overline{r} \geq \underline{r} > 0$ as the two roots of the following equation in variable $y$:*

$$ay - \tau\log y + (d-1)\left(\tau - \tau\log\frac{\tau}{a}\right) - \frac{a}{N}\sum_{j=1}^{N}\left\|\mathbf{x}_j\mathbf{x}_j^\top\right\|_1 - \frac{1}{2}a^2 d^2 - \mathcal{L}\left(\boldsymbol{\Sigma}^{(0)}\right) = 0,$$

*where $\mathcal{L}$ denotes the objective function in problem equation 2, and $\boldsymbol{\Sigma}^{(0)}$ is any initialization for Algorithm 1. Denote the condition number $\kappa = L/\mu > 1$ with $L = 1 + 4\tau\underline{r}^{-2}$ and $\mu = \tau(\overline{r} + \underline{r}/2)^{-2} - L_q$. Suppose that $\overline{r} + \underline{r}/2 < \sqrt{\tau/L_q}$, then the sequence $\{\boldsymbol{\Sigma}^{(t)}\}$ from Algorithm 1 with step size parameter $\gamma \geq \max\left\{2\left(ad + \tau\sqrt{d}/\underline{r} + 2\lambda d\right)/\underline{r}, L\right\}$, satisfies $\underline{r}\mathbf{I} \preceq \boldsymbol{\Sigma}^{(t)} \preceq \overline{r}\mathbf{I}$ for all $t$, and the following linear convergence rate holds:*

$$\left\|\boldsymbol{\Sigma}^{(t)} - \widehat{\boldsymbol{\Sigma}}\right\|_F^2 \leq C_1\left(1 - \frac{1}{C_2\kappa}\right)^t,$$

*where $\widehat{\boldsymbol{\Sigma}}$ is the unique minimizer of problem 2 in the set $\{\boldsymbol{\Sigma} \mid \mathbf{0} \preceq \boldsymbol{\Sigma} \prec \sqrt{\tau/L_q}\mathbf{I}\}$, $C_1 = \left\|\boldsymbol{\Sigma}^{(0)} - \widehat{\boldsymbol{\Sigma}}\right\|_F^2$, and $C_2 = \gamma/L$.*

Theorem 1 implies that if the largest eigenvalue of the initialization $\boldsymbol{\Sigma}^{(0)}$ is smaller than $\sqrt{\tau/L_q}$, then the sequence $\boldsymbol{\Sigma}^{(t)}$ generated by Algorithm 1 converges linearly to the unique solution $\widehat{\boldsymbol{\Sigma}}$ of problem equation 2, with a rate of $O\left(1 - 1/(C_2\kappa)\right)$. Moreover, with an appropriately chosen step size $\gamma$, the iterates $\boldsymbol{\Sigma}^{(t)}$ remain positive definite throughout, even without explicitly enforcing the positive definiteness constraint.

---

**Algorithm 2** Robust Sparse Covariance Matrix Estimation over Networks

---

**given** $\mathbf{\Sigma}_i^{(0)} \succ 0$, $\mathbf{Y}_i^{(0)} = \nabla \mathcal{F}_i \left( \mathbf{\Sigma}_i^{(0)} \right)$, for $i = 1, \ldots, m$, $\mathbf{W}$, $\gamma > 0$, $\theta \in (0, 1]$, $t = 0$

**while** not converge, each agent $i$ **do**

$$\mathbf{\Sigma}_i^{\left(t+\frac{1}{2}\right)} = \mathrm{ST}_{\frac{\lambda}{\gamma}} \left( \mathbf{\Sigma}_i^{(t)} - \frac{1}{\gamma} \mathbf{Y}_i^{(t)} \right) \qquad\qquad\text{(Local optimization)}$$

$$\mathbf{\Sigma}_i^{(t+1)} = \sum_{j=1}^m W_{ij} \left( \mathbf{\Sigma}_j^{(t)} + \theta \left( \mathbf{\Sigma}_j^{\left(t+\frac{1}{2}\right)} - \mathbf{\Sigma}_j^{(t)} \right) \right) \qquad\text{(Variable tracking) (6)}$$

$$\mathbf{Y}_i^{(t+1)} = \sum_{j=1}^m W_{ij} \left( \mathbf{Y}_j^{(t)} + \nabla \mathcal{F}_j \left( \mathbf{\Sigma}_j^{(t+1)} \right) - \nabla \mathcal{F}_j \left( \mathbf{\Sigma}_j^{(t)} \right) \right) \qquad\text{(Gradient tracking) (7)}$$

$t = t + 1$

**end while**

**return** $\mathbf{\Sigma}_i^{(t)}$, for $i = 1, \ldots, m$.

---

## 5 ROBUST SPARSE COVARIANCE MATRIX ESTIMATION OVER NETWORKS

In this section, we propose an efficient decentralized algorithm to solve problem equation 2 over a network, and provide a convergence analysis.

**Distributed proximal gradient with tracking algorithm.** In a distributed estimation setting, each agent lacks access to the global objective in problem equation 2 and thus cannot solve it independently. To address this challenge, we propose a distributed optimization algorithm, summarized in Algorithm 2, which extends Algorithm 1 to decentralized networks via a gradient tracking scheme (Di Lorenzo & Scutari, 2016; Nedic et al., 2017; Xu et al., 2017; Sun et al., 2022b). Algorithm 2 iteratively repeats two stages: local optimization and information mixing, progressively driving each local copy to approximate the true global information.

LOCAL OPTIMIZATION. For each agent $i = 1, \ldots, m$, define the local objective function as $\mathcal{F}_i(\mathbf{\Sigma}) = \mathcal{H}_i(\mathbf{\Sigma}) - \tau \log \det(\mathbf{\Sigma}) - \mathcal{Q}(\mathbf{\Sigma})$. Let $\mathbf{\Sigma}_i$ denote the local estimate of $\mathbf{\Sigma}$ and let $\mathbf{Y}_i$ denote the local auxiliary variable that aims to asymptotically track the global gradient $\frac{1}{m} \sum_{i=1}^m \nabla \mathcal{F}_i(\mathbf{\Sigma}_i)$. Similar to proximal gradient method, the local optimization step for agent $i$ at iteration $t$ is given by

$$\mathbf{\Sigma}_i^{\left(t+\frac{1}{2}\right)} \in \arg \min_{\mathbf{\Sigma}_i \succ \mathbf{0}} \left\{ \mathcal{F}_i \left( \mathbf{\Sigma}_i^{(t)} \right) + \left\langle \mathbf{Y}_i^{(t)}, \mathbf{\Sigma}_i - \mathbf{\Sigma}_i^{(t)} \right\rangle + \frac{\gamma}{2} \left\| \mathbf{\Sigma}_i - \mathbf{\Sigma}_i^{(t)} \right\|_F^2 + \lambda \left\| \mathbf{\Sigma} \right\|_{1,\mathrm{off}} \right\}. \quad (5)$$

Compared to the centralized update in equation 4, the local optimization step in equation 5 replaces the inaccessible global variable and gradient with locally maintained auxiliary variables. Similar to Section 4, we solve equation 4 using a soft-thresholding operation: $\mathbf{\Sigma}_i^{\left(t+\frac{1}{2}\right)} = \mathrm{ST}_{\frac{\lambda}{\gamma}} \left( \mathbf{\Sigma}_i^{(t)} - \frac{1}{\gamma} \mathbf{Y}_i^{(t)} \right)$. As in the centralized case, we will later show that, with a properly chosen step size $\frac{1}{\gamma}$, the positive definiteness constraint is automatically satisfied.

INFORMATION MIXING. After local optimization, each agent $i$ collects information from its neighbors and updates both $\mathbf{\Sigma}_i$ and $\mathbf{Y}_i$ according to equation 6 and equation 7, where the weights $W_{ij}$ are defined in Assumption 1 and $\theta \in (0, 1]$ is a step size parameter. Under Assumption 1, each update corresponds to a weighted average of the auxiliary variables received from $\mathcal{N}_i$.

**Convergence analysis.** We now present the convergence property of Algorithm 2.

**Theorem 2** (Convergence property of Algorithm 2). *Suppose Assumptions 1 and 3 hold for problem 2. Define constants $\overline{r} \geq \underline{r} > 0$ and $e > 0$, which depend on the object function $\mathcal{L}$ and the initialization $\left\{ \mathbf{\Sigma}_i^{(0)} \right\}_{i=1}^m$; the explicit dependencies are detailed in the proof. Denote the condition number $\kappa = L/\mu > 1$ with $L = 1 + 4\tau \underline{r}^{-2}$ and $\mu = \tau \left( \overline{r} + \underline{r}/2 \right)^{-2} - L_q$. Suppose that $\overline{r} + \underline{r}/2 < \sqrt{\tau/L_q}$ and the network connectivity parameter $\rho$ satisfies $\rho \leq \left( \left( \sqrt{\kappa^2 + (12\kappa - 2)(\kappa - 1)} - \kappa \right)/(6\kappa - 1) \right)^2$,*

*then the sequence $\left\{\mathbf{\Sigma}_i^{(t)}\right\}_{i=1}^m$ from Algorithm 2 with step size parameters*

$$\gamma \geq \max\left\{2\underline{r}^{-1}\left(\sqrt{m\left(ad + \frac{\sqrt{d}\tau}{\underline{r}} + \lambda d\right)^2 + e} + \lambda d\right), L + \frac{48L^2 m\sqrt{\rho}}{\mu(1-\rho)^2}\right\}, \tag{8}$$

$$\theta \leq \min\left\{\left(\sqrt{\frac{L^2}{16} + 32L\gamma\left(\frac{\rho^2(1+\rho^2)}{(1-\rho^2)^2} + 4\frac{\rho^4(1+\rho^2)^2}{(1-\rho^2)^4}\right)} + \frac{L}{4}\right)^{-1}\gamma, 1\right\}, \tag{9}$$

*satisfies $\underline{r}\mathbf{I} \preceq \mathbf{\Sigma}_i^{(t)} \preceq \bar{r}\mathbf{I}$ for all $i$ and $t$, and the following linear convergence rate holds:*

$$\sum_{i=1}^m \left\|\mathbf{\Sigma}_i^{(t)} - \widehat{\mathbf{\Sigma}}\right\|_F^2 \leq C_1'\left(1 - \frac{1}{C_2'\kappa}\right)^t, \tag{10}$$

*where $\widehat{\mathbf{\Sigma}}$ is the unique solution of problem 2 in the set $\left\{\mathbf{\Sigma} \mid \mathbf{0} \preceq \mathbf{\Sigma} \prec \sqrt{\tau/L_q}\mathbf{I}\right\}$, $C_1'$ is a constant related to $\left\{\mathbf{\Sigma}_i^{(0)}\right\}_{i=1}^m$, and $C_2' = 2\gamma/((2-\rho)\theta L)$.*

Theorem 2 implies that the sequences $\left\{\mathbf{\Sigma}_i^{(t)}\right\}$, for all $i = 1, \ldots, m$, generated by Algorithm 2 converge to $\widehat{\mathbf{\Sigma}}$ and reach consensus with a linear rate. Note that $\widehat{\mathbf{\Sigma}}$ is the unique solution to problem 2 when $\mathbf{\Sigma} \prec \sqrt{\tau/L_q}\mathbf{I}$. Hence, Algorithm 2 can obtain the same solution as Algorithm 1. In the distributed estimation setting, the linear convergence rate of Algorithm 2 is influenced by the network connectivity. In particular, as indicated by equation 8, equation 9, and equation 10, a smaller value of $\rho$ (corresponding to stronger network connectivity) leads to a smaller lower bound on $\gamma$ and a larger upper bound on $\theta$, thereby accelerating convergence. Moreover, from the expression of $C_2'$, a smaller $\rho$ directly contributes to a faster convergence rate.

## 6    STATISTICAL GUARANTEE

In this section, we analyze the statistical estimation performance of $\widehat{\mathbf{\Sigma}}$, the estimator produced by Algorithms 1 and 2. We assume that the true covariance matrix $\mathbf{\Sigma}^\star \succ \mathbf{0}$ of $\mathbf{x}$ is $s$-sparse; that is, it contains at most $s$ nonzero entries. Let $\mathcal{S} = \text{supp}(\mathbf{\Sigma}^\star)$ denote the support of $\mathbf{\Sigma}^\star$. We establish a nonasymptotic bound on the estimation error $\left\|\widehat{\mathbf{\Sigma}} - \mathbf{\Sigma}^\star\right\|_F$.

**Theorem 3** (Statistical guarantee)**.** *Suppose Assumptions 2 and 3 hold for problem 2. Define $K = \max\left\{\sigma^2, (2\sigma)^{2(1+\nu)}\right\}$ and $\mu_0 \in (0, 2)$ and choose parameters satisfying*

$$a = c_a\sqrt{\frac{KN}{\log d}}, \qquad \tau \leq c_\tau \min\left\{\left\|\left((\mathbf{\Sigma}^\star)^{-1}\right)_S\right\|_F^{-1}\sqrt{\frac{s}{N}}, \left\|(\mathbf{\Sigma}^\star)^{-1}\right\|_{\max}^{-1}\sqrt{\frac{\log d}{N}}\right\},$$

$$\lambda = c_\lambda\left(\left(\sqrt{6} + 2c_a + \frac{1}{c_a}\right)\sqrt{K} + c_\tau\right)\sqrt{\frac{\log d}{N}}, \quad b \leq \lambda^{-1}|\Sigma_{kl}^\star|, \ \forall (k, l) \in \mathcal{S}, \quad L_q \leq \frac{c_q\mu_0}{\sqrt{s}},$$

*where $c_a > 0$, $c_\tau > 0$, $c_\lambda > 1$, and $c_q \in (0, \sqrt{s})$ are universal constants. If the sample size satisfies*

$$N > \max\left\{\frac{(\log d)^{1+\frac{1}{\nu}}}{c_a^{2(1+\frac{1}{\nu})}K}, \frac{16c_N \log d}{(2-\mu_0)^2}, \frac{38c_\lambda}{\mu_0 c_a \sqrt{K}}\left(\left(\sqrt{6} + 2c_a + \frac{1}{c_a}\right)\sqrt{K} + c_\tau\right)\sqrt{s}\log d\right\},$$

*where $c_N > \max\{1, 1/((2-\mu_0)c_a^2)\}$ is a universal constant, then with high probability,*

$$\left\|\widehat{\mathbf{\Sigma}} - \mathbf{\Sigma}^\star\right\|_F \leq C_s\sqrt{\frac{s}{N}},$$

*where $C_s = (\beta\sqrt{K} + c_\tau)/(\mu_0 - L_q)$ with $\beta > \sqrt{2}$.*

Theorem 3 indicates that, under appropriate conditions, the estimator $\widehat{\mathbf{\Sigma}}$ produced by both Algorithms 1 and 2 achieves the oracle rate $\mathbf{\Sigma}^\star$ with oracle rate $O\left(\sqrt{s/N}\right)$ in Frobenius norm. The oracle rate refers to the convergence rate of the estimation error that can be achieved by an estimator when the true support set $\mathcal{S}$ is known in advance (Wainwright, 2019). By combining Theorems 1, 2, and 3, we conclude that both algorithms converge within $O(\log(1/\varepsilon)/\log(1-1/\kappa))$ iterations to within an $\varepsilon$-neighborhood of a statistically optimal estimate, attaining an error of order $O(\sqrt{s/N})$.

**Refined convergence rate.** In Theorems 1 and 2, the condition number $\kappa$ depends on $\underline{r}$ and $\bar{r}$. Since these quantities are affected by the initialization, sample size, and dimension, the exact convergence rate under high-dimensional settings and varying initializations remains unclear. To address this issue, we leverage Theorem 3 to refine the convergence rates. The resulting rates, reported in Corollaries 1 and 2, are independent of the initialization and better suited to high-dimensional scenarios.

**Corollary 1** (Refined convergence rate for Algorithm 1). *Suppose Assumptions 2 and 3 hold for problem 2, and all the conditions in Theorems 1 and 3 are satisfied. Then, for $t > T$ where $T = \max\left\{0, \left\lceil 2\log\left(c_h/\sqrt{C_1}\right)/\log\left(1 - 1/(C_2\kappa)\right)\right\rceil\right\}$, Algorithm 1 converges with a refined rate*

$$\left\|\mathbf{\Sigma}^{(t)} - \widehat{\mathbf{\Sigma}}\right\|_F^2 \leq C_3 \left(1 - \frac{1}{C_4\kappa_r}\right)^{t-T}$$

*with high probability, where $C_3 = C_1 (1 - 1/(C_2\kappa))^T$, $C_4 = \gamma/(1 + \tau c_4)$, and $\kappa_r = (1 + 4\tau c_4)/(\mu_0 - L_q)$. When $\lambda_{\min}(\mathbf{\Sigma}^\star) - a/2 > 0$, we have $c_h = \frac{a}{2} - \|\widehat{\mathbf{\Sigma}} - \mathbf{\Sigma}^\star\|_F$ and $c_4 = 1/(\lambda_{\min}(\mathbf{\Sigma}^\star) - a/2)^2$; when $\lambda_{\min}(\mathbf{\Sigma}^\star) - a/2 \leq 0$, suppose that $C_s\sqrt{s/N} < \lambda_{\min}(\mathbf{\Sigma}^\star)/(12c_r)$, where $c_r > 1/12$, we have $c_h = \lambda_{\min}(\widehat{\mathbf{\Sigma}}) - (12c_r - 1)\lambda_{\min}(\mathbf{\Sigma}^\star)/12c_r$ and $c_4 = 144c_r^2/((12c_r - 1)\lambda_{\min}(\mathbf{\Sigma}^\star))^2$.*

**Corollary 2** (Refined convergence rate for Algorithm 2). *Suppose Assumptions 1, 2, and 3 hold for problem 2, and all the conditions in Theorems 2 and 3 are satisfied. Assume that $\rho \leq \left(\left(\sqrt{\kappa_r^2 + (12\kappa_r - 2)(\kappa_r - 1)} - \kappa_r\right)/(6\kappa_r - 1)\right)^2$ and $\gamma \geq 1 + \tau c_4 + 48(1 + \tau c_4)\kappa m\sqrt{\rho}/(1 - \rho)^2$. Then, for $t > T$ where $T = \max\left\{0, \left\lceil 2\log\left(c_h/\sqrt{C_3''}\right)/\log\left(1 - 1/(C_2'\kappa)\right)\right\rceil\right\}$ with $C_3''$ a constant related to $\left\{\mathbf{\Sigma}_i^{(0)}\right\}_{i=1}^m$, Algorithm 2 converges with*

$$\left\|\mathbf{\Sigma}^{(t)} - \widehat{\mathbf{\Sigma}}\right\|_F^2 \leq C_3' \left(1 - \frac{1}{C_4'\kappa_r}\right)^{t-T}$$

*with high probability, where $C_3' = C_3'' (1 - 1/(C_2'\kappa))^T$ and $C_4' = 2\gamma/((2 - \rho)\theta(1 + \tau c_4))$.*

Corollaries 1 and 2 show that, after $T$ iterations of Algorithms 1 and 2, the convergence rate depends on the condition number $\kappa_r$, which is solely determined by $\lambda_{\min}(\mathbf{\Sigma}^\star)$. In both corollaries, $c_h$ is an interval parameter associated with $\widehat{\mathbf{\Sigma}}$ and $\mathbf{\Sigma}^\star$, while $C_1$ and $C_3''$ are coefficients related to the initialization distance from $\widehat{\mathbf{\Sigma}}$. Taking Corollary 1 as an example, when $c_h \geq \sqrt{C_1}$, we have $\left\lceil 2\log\left(c_h/\sqrt{C_1}\right)/\log\left(1 - 1/(C_2\kappa)\right)\right\rceil \leq 0$. This implies that if the initialization is sufficiently close to $\widehat{\mathbf{\Sigma}}$, the corollaries apply starting from $T = 0$.

# 7 NUMERICAL EXPERIMENTS

In this section, we demonstrate the convergence of the proposed algorithms and assess their estimation performance on synthetic and real data. We choose MCP as the non-convex penalty function $p_\lambda$, and set $\tau = 0.1$ and $b = 2$ as recommended by Wei & Zhao (2023). We set the initialization $\mathbf{\Sigma}_i^{(0)} = 0.1\mathbf{I}$ so that $\mathbf{\Sigma}_i^{(0)} \prec \sqrt{\tau/L_q}\mathbf{I}$, where $L_q = 0.5$ for MCP with $b = 2$. The parameters $a$ and $\lambda$ are selected via five-fold cross-validation. The step size parameters $\gamma$ and $\theta$ are tuned to ensure the convergence of the algorithm. Specifically, we fix $\theta = 0.1$ and set $\gamma = \eta^k$, where $\eta > 1$ and $k$ is the smallest integer ensuring the convergence of the proposed algorithms. We compare the performance of the proposed methods with two high-dimensional sparse covariance matrix estimators (PD_MCP (Wei & Zhao, 2023) and NetGGM (Xia et al., 2025)), as well as several robust sparse covariance estimators (Adaptive Huber (Avella-Medina et al., 2018), Reg_TME (Goes et al., 2020), PD_gQNE (Lu et al., 2021), and M-COAT (Li et al., 2023)). The hyperparameters for all baseline methods are selected according to the procedures described in their respective original references. We consider two $d$-dimensional sparse covariance matrix models with $d = 100$: 1) banded structure: $\Sigma_{ij}^\star = 1 - |i - j|/10$ if $|i - j| \leq 10$, otherwise $\Sigma_{ij}^\star = 0$; 2) block structure: The indices $1, 2, \ldots, d$ are partitioned into 10 ordered groups of equal size with $\Sigma_{ij}^\star = 1$ if $i = j$, $\Sigma_{ij}^\star = 0.6$ if $i$ and $j$ are in the same group, otherwise $\Sigma_{ij}^\star = 0$. Moreover, To evaluate robustness under different distributions, we generate data from three models within the Gaussian scale mixture framework: $\mathbf{x}_i = \phi_i\sqrt{\mathbf{\Sigma}^\star}\zeta_i$,

Table 1: Performance Comparison of Estimators Under Different Distributions

| Banded structure | | Adaptive Huber | RegTME | PD_gQNE | M-COAT | PD_MCP | NetGGM | Proposed |
|---|---|---|---|---|---|---|---|---|
| Gaussian | NMSE | 0.2551(0.0028) | 0.2501(0.0022) | 0.2145(0.0037) | 0.2346(0.0041) | 0.1591(0.0021) | 0.1833(0.0041) | **0.1574(0.0024)** |
| | F1-Score | 0.3065(0.0001) | 0.7339(0.0030) | 0.4847(0.0023) | 0.5944(0.0026) | 0.7601(0.0061) | 0.3065(0.0001) | **0.7627(0.0073)** |
| $t$ | NMSE | 0.2873(0.0033) | 0.2795(0.0039) | 0.2231(0.0037) | 0.2793(0.0021) | 0.2491(0.0039) | 0.4167(0.0332) | **0.2139(0.0047)** |
| | F1-Score | 0.3065(0.0001) | 0.7036(0.0040) | 0.5249(0.0041) | 0.5096(0.0063) | 0.7123(0.0028) | 0.3065(0.0001) | **0.7390(0.0158)** |
| Laplace | NMSE | 0.2343(0.0027) | 0.2429(0.0032) | 0.2089(0.0021) | 0.2377(0.0031) | 0.2539(0.0023) | 0.6734(0.0374) | **0.1821(0.0026)** |
| | F1-Score | 0.3065(0.0001) | 0.6847(0.0037) | 0.6067(0.0034) | 0.4953(0.0016) | 0.6720(0.0024) | 0.3065(0.0001) | **0.7042(0.0101)** |
| Block structure | | Adaptive Huber | RegTME | PD_gQNE | M-COAT | PD_MCP | NetGGM | Proposed |
| Gaussian | NMSE | 0.2503(0.0034) | 0.2366(0.0022) | 0.2336(0.0039) | 0.2223(0.0038) | 0.1489(0.0019) | 0.2310(0.0027) | **0.1442(0.0026)** |
| | F1-Score | 0.1818(0.0001) | 0.8231(0.0034) | 0.5386(0.0025) | 0.6354(0.0038) | 0.8117(0.0079) | 0.1818(0.0001) | **0.8280(0.0083)** |
| $t$ | NMSE | 0.2592(0.0038) | 0.2562(0.0044) | 0.2541(0.0041) | 0.2452(0.0158) | 0.2689(0.0017) | 0.4952(0.0441) | **0.1949(0.0050)** |
| | F1-Score | 0.1818(0.0001) | 0.8697(0.0035) | 0.4165(0.0048) | 0.5395(0.0117) | 0.8499(0.0041) | 0.1818(0.0001) | **0.8829(0.0053)** |
| Laplace | NMSE | 0.2235(0.0029) | 0.2411(0.0021) | 0.2331(0.0029) | 0.2610(0.0121) | 0.2849(0.0027) | 0.6811(0.0425) | **0.1735(0.0038)** |
| | F1-Score | 0.1818(0.0001) | 0.7818(0.0029) | 0.6165(0.0024) | 0.5473(0.0023) | 0.7614(0.0029) | 0.1818(0.0001) | **0.7984(0.0023)** |

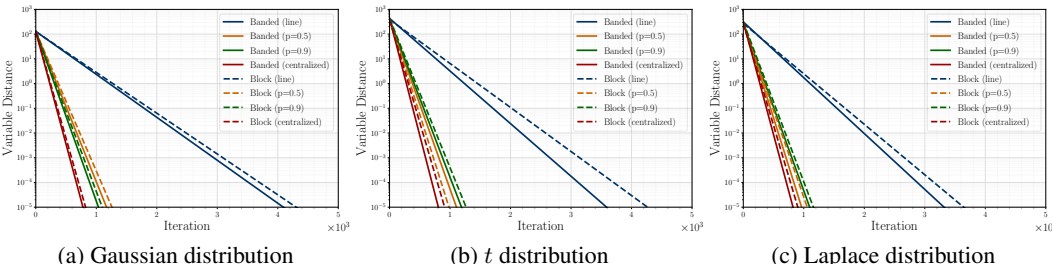

(a) Gaussian distribution     (b) $t$ distribution     (c) Laplace distribution

Figure 1: Convergence of the proposed algorithms on different networks.

$i = 1, 2, \ldots, N$, where $\zeta_i \sim \mathcal{N}(0, \mathbf{I})$ and $\phi_i$ is a random variable drawn from one of the following distributions: 1) Gaussian distribution: $\phi_i \sim \mathcal{N}(0, 1)$; 2) $t$ distribution: $\phi_i = \sqrt{5/\nu_i}$, where $\nu_i \sim \chi_5^2$ and hence $\mathbf{x}_i \sim t_5(\mathbf{0}, \mathbf{\Sigma})$. 3) Laplace distribution: $\phi_i \sim \mathrm{Laplace}(0, 1)$. A total of $N = 50$ samples are drawn and evenly distributed among $m = 25$ agents. In the distributed setting, we evaluate the performance of Algorithm 2 over three different connected, time-invariant undirected networks: two Erdős–Rényi random graphs (Erdős & Rényi, 1959), where each pair of agents is connected independently with probability $p = 0.9$, and $p = 0.5$, respectively; and a line graph, where agent $i$ is connected to agent $i-1$ and agent $i+1$ for $i = 2, \ldots, m-1$. The weight matrix $\mathbf{W}$ for each network is constructed using the Metropolis rule (Xiao et al., 2005): $W_{ij} = 1/(\max(d_i, d_j) + 1)$ if $i \neq j$ and $(i, j) \in \mathcal{E}$, $W_{ij} = 0$ if $i \neq j$ and $(i, j) \notin \mathcal{E}$, and $W_{ij} = 1 - \sum_{i \neq l} W_{il}$ if $i = j$, where $d_i$ denotes the degree of agent $i$.

We first compare the estimation performance of the proposed method with several baselines. The performance is evaluated using the normalized mean squared error (NMSE), defined as $\mathrm{NMSE}(\mathbf{\Sigma}) = \|\mathbf{\Sigma} - \mathbf{\Sigma}^\star\|^2 / \|\mathbf{\Sigma}^\star\|^2$ and the F1-score (Witten et al., 2005). The results, averaged over 100 Monte Carlo trials, are reported in Table 1. As shown in the table, Algorithm 1 performs comparably to PD_MCP (Wei & Zhao, 2023) under the Gaussian model and outperforms all baseline methods in the non-Gaussian settings. These results highlight the superior accuracy and robustness of the proposed method. Next, we evaluate the convergence behavior of the proposed algorithms. Figure 1 shows the decrease in variable distances $\|\mathbf{\Sigma}^{(t)} - \widehat{\mathbf{\Sigma}}\|_F^2$ and $\frac{1}{m} \sum_{i=1}^m \|\mathbf{\Sigma}_i^{(t)} - \widehat{\mathbf{\Sigma}}\|_F^2$ for Algorithms 1 and 2 over three data generating models, where $\widehat{\mathbf{\Sigma}}$ denotes the final estimate obtained by Algorithm 1. The results confirm that both algorithms converge linearly to the same optimal solution. Moreover, the convergence rate of Algorithm 2 improves as the connectivity of the underlying network increases.

To evaluate the effectiveness of our proposed methods, we conducted experiments using the Leukemia dataset Golub et al. (1999). This dataset contains 72 gene expression profiles: 47 samples from patients with acute lymphoblastic leukemia (ALL) and 25 samples from patients with acute myeloid leukemia (AML). Each sample is represented by 7,129 gene expression levels. Following the approaches outlined in Rothman et al. (2009); Cui et al. (2016); Xia et al. (2025), we first computed the $F$ statistic for each gene $j$ as follows:

$$F(x_j) = \frac{\frac{1}{K-1} \sum_{l=1}^K N_{(l)} (\bar{x}_j(l) - \bar{x}_j)}{\frac{1}{N-K} \sum_{l=1}^K N_{(l)} (N_{(l)} - 1) \hat{\sigma}_{(l)}^2},$$

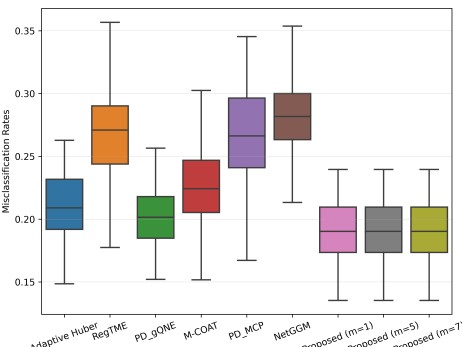

Figure 2: Misclassification rates of QDA on Leukemia dataset using different methods.

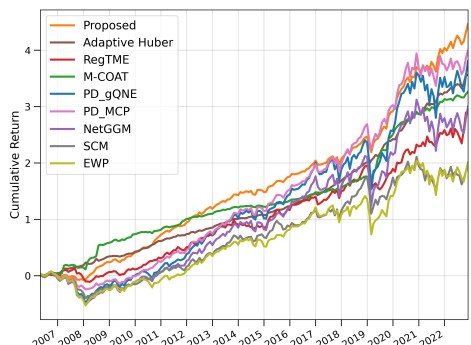

Figure 3: Cumulative returns of the GMVP constructed from different methods.

where $K = 2$ denotes the number of classes, $N_{(l)}$ is the sample size of class $l$, $\bar{x}$ and $\bar{x}_{j(l)}$ represents the overall mean and the mean of class $l$. We ranked the genes by their F-statistic and selected the top 75 and the bottom 25 genes, resulting in a total of $d = 100$ genes. Next, we randomly partitioned the dataset into 100 different subsets, each containing 35 training samples (from 23 ALL and 12 AML) and 37 test samples (from 24 ALL and 13 AML). In real-world scenarios, such datasets may be distributed across separate hospitals and cannot be directly shared due to privacy regulations. To simulate this condition, we distributed the training samples randomly among $m = \{1, 5, 7\}$ agents and then estimated the covariance matrices. We incorporated the estimated covariance matrices into a quadratic discriminant analysis (QDA) model, as described in Hastie et al. (2009), and evaluated the classification performance based on the misclassification rate. As the estimates obtained by the RegTME (Goes et al., 2020) do not always guarantee positive definiteness, we present results only for cases with positive definite estimates. As shown in Figure 2, regardless of the number of agents, our proposed method consistently achieves the lowest misclassification rate. This indicates that our decentralized algorithm achieves estimation performance equivalent to the centralized approach across diverse networks and demonstrates the superior estimation performance of our methodology.

We further conduct experiments on financial time-series data to assess the performance of different covariance estimators. A standard approach for evaluating the quality of an estimated covariance matrix is to examine the returns of portfolios constructed from it. In this study, we focus on the global minimum variance portfolio (GMVP) under a no-short-sales constraint, which is formulated as:

$$\min_{\boldsymbol{\omega} \in \mathbb{R}^d} \quad \boldsymbol{\omega}^\top \boldsymbol{\Sigma} \boldsymbol{\omega} \qquad \text{subject to} \quad \boldsymbol{\omega}^\top \mathbf{1} = 1, \quad \boldsymbol{\omega} \geq \mathbf{0},$$

This optimization problem can be efficiently solved using CVX (Grant et al., 2008). We collect historical monthly stock prices for the components of the S&P 100 Index over a 240-month period, from December 2002 to December 2022. After excluding stocks with missing data, we obtain monthly returns for 78 companies ($d = 78$). We evaluate the estimators using a rolling-window scheme, where each window consists of 40 months for training and one subsequent month for testing. Portfolio performance is assessed by comparing the cumulative returns over the test period. Figure 3 shows the cumulative returns of the GMVP based on different covariance estimators, including the equal-weighted portfolio (EWP, $\boldsymbol{\omega} = \mathbf{1}/d$) as a heuristic baseline. The results indicate that the EWP yields the lowest cumulative return, confirming its ineffectiveness as a strategy. Moreover, the SCM method, which does not incorporate regularization, performs substantially worse than regularized estimators. Among the regularized methods, our proposed approach achieves the highest cumulative return, demonstrating a clear advantage over competing methods.

## 8 CONCLUSION

In this paper, we have studied the problem of distributed sparse covariance matrix estimation under heavy-tailed data. The estimation task has been formulated as a non-convex and non-globally Lipschitz smooth problem that minimizes a Huber loss function with a log-determinant barrier and a sparsity-inducing non-convex penalty. We have proposed both centralized and decentralized algorithms, and established that both methods converge linearly to the same solution, which achieves the oracle statistical rate in Frobenius norm. Simulation results validated the theoretical guarantees and demonstrated the superior accuracy and robustness of the proposed estimator.

ETHICS STATEMENT

All authors of this paper have read and agree to adhere to the ICLR Code of Ethics. Our work does not involve human subjects, sensitive data, or applications that could directly cause harm. The datasets used in our experiments are publicly available and do not contain personally identifiable information. We have taken care to ensure that our methodology does not introduce or reinforce unfair bias, and we discuss potential limitations and societal impacts in the main text. There are no conflicts of interest or external sponsorships that could influence the results or their interpretation.

REPRODUCIBILITY STATEMENT

We are committed to ensuring the reproducibility of our results. All experimental details, including model architectures, parameters, training procedures, and evaluation metrics, are described in the main paper and appendix. We provide anonymized source code and preprocessing scripts as supplementary material, where the data and simulation code are available at https://anonymous.4open.science/r/distributed_covariance-4E65. The datasets used are publicly accessible, and we include links and version information in the appendix. For theoretical claims, full proofs and assumptions are provided in the appendix. We further conduct ablation and sensitivity analyses to demonstrate the robustness of our approach.

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

# Contents

## A   LLM USAGE STATEMENT

During the preparation of this manuscript, we used large language models (LLMs) as a general-purpose writing and editing aid. LLMs were used to improve clarity, grammar, and structure of the text, but they did not contribute to the research ideation, experimental design, or scientific content. No LLM was used to generate new ideas, proofs, or experimental results. All content, including any text suggested by LLMs, was reviewed and verified by the authors, who take full responsibility for the final manuscript.

## B   NOTATION

The following notation is adopted throughout the paper. Standard lowercase or uppercase letters represent scalars, while boldface lowercase and uppercase letters denote vectors and matrices, respectively. $A_{ij}$ (or $[A]_{ij}$) stands for the $(i, j)$-th entry of matrix $\mathbf{A}$. $\mathbf{1}$ stands for vector with all elements equal to one. $\mathbf{0}$ stands for all zero matrix and $\mathbf{I}$ stands for the identity matrix. For a matrix $\mathbf{A}$, let $\lambda_{\min}(\mathbf{A})$ and $\lambda_{\max}(\mathbf{A})$ denote its smallest and largest eigenvalues, respectively; $\lambda_k(\mathbf{A})$ denotes its $k$-th largest eigenvalue. $\|\mathbf{A}\|_F$ denotes the Frobenius norm, defined as the square root of the sum of the squares of all elements; $\|\mathbf{A}\|_1$ denotes the sum of the absolute values of all elements; $\|\mathbf{A}\|_{\max}$ denotes the largest absolute value among all elements; $\|\mathbf{A}\|_2$ denotes the induced matrix 2-norm, i.e., the maximum absolute column sum; and $\|\mathbf{A}\|_\infty$ denotes the induced matrix $\infty$-norm, i.e., the maximum absolute row sum.

For two functions $f(x)$ and $g(x)$, we use $f(x) = O(g(x))$ to indicate that there exists a positive constant $M$ and a constant $x_0$ such that $|f(x)| \leq M |g(x)|$ for any $x \geq x_0$. For a function $f$, $\nabla f(\mathbf{x})$ and denote the Jacobian and the Hessian of $f$ at $\mathbf{x}$, respectively. For a convex function $f$, $\partial f(\mathbf{x})$ stands for the subdifferential of $f$ at $\mathbf{x}$. $\otimes$ stands for the Kronecker product.

Table 2: Summary of main notation used in the paper.

| Symbol | Description |
|---|---|
| **Dimensions, indices, and sets** | |
| $d$ | Dimension of the random vector $\boldsymbol{x}$ and covariance matrix $\boldsymbol{\Sigma}$. |
| $N$ | Total number of samples. |
| $m$ | Number of agents in the network. |
| $n$ | Local sample size at each agent; $N = mn$. |
| $J_i$ | Index set of samples stored at agent $i$, with $\bigcup_{i=1}^m J_i = \{1, \ldots, N\}$. |
| $S$ | Support set of nonzero entries of $\boldsymbol{\Sigma}^\star$. |
| $S^c$ | Complement of $S$. |
| $s$ | Sparsity level; number of nonzero entries in $\boldsymbol{\Sigma}^\star$. |
| $\mathbb{S}_{++}^d$ | Cone of $d \times d$ symmetric positive definite matrices. |
| **Random variables and covariance matrices** | |
| $\mathbf{x} \in \mathbb{R}^d$ | Zero-mean random vector with covariance $\boldsymbol{\Sigma}^\star$. |
| $x_k$ | $k$-th coordinate of $\boldsymbol{x}$. |
| $\{\mathbf{x}_j\}_{j=1}^N$ | I.i.d. samples of $\boldsymbol{x}$. |
| $x_{jk}$ | $k$-th entry of the $j$-th sample $\boldsymbol{x}_j$. |
| $\boldsymbol{\Sigma}^\star$ | True covariance matrix of $\boldsymbol{x}$. |
| $\boldsymbol{\Sigma}$ | Generic covariance matrix in $\mathbb{S}_{++}^d$ (optimization variable). |
| $\boldsymbol{\Sigma}^{(t)}$ | Iterate of the centralized Algorithm 1 at iteration $t$. |
| $\boldsymbol{\Sigma}_i^{(t)}$ | Local iterate of agent $i$ in the decentralized algorithm at iteration $t$. |
| $\bar{\boldsymbol{\Sigma}}^{(t)}$ | Network average $\bar{\boldsymbol{\Sigma}}^{(t)} = \frac{1}{m} \sum_{i=1}^m \boldsymbol{\Sigma}_i^{(t)}$. |
| **Network and communication** | |
| $\mathcal{G} = (\mathcal{V}, \mathcal{E})$ | Undirected communication graph of agents. |
| $\mathcal{V} = \{1, \ldots, m\}$ | Node (agent) index set. |
| $\mathcal{E}$ | Edge set of $G$. |
| $N_i$ | Neighborhood of agent $i$, including $i$ itself. |
| $\mathbf{W} = [W_{ij}] \in \mathbb{R}^{m \times m}$ | Doubly stochastic weight matrix compliant with $G$. |

**Table 2 (continued)**

| Symbol | Description |
|---|---|
| $\mathbf{J}$ | Averaging matrix, $J = \frac{1}{m}\mathbf{1}\mathbf{1}^\top$. |
| $\rho$ | Spectral gap parameter $\rho = \|\mathbf{W} - \mathbf{J}\|_2 \in [0,1)$. |

**Objective functions and penalties**

| Symbol | Description |
|---|---|
| $\mathcal{H}_i(\boldsymbol{\Sigma})$ | Local Huber loss at agent $i$. |
| $\mathcal{H}(\boldsymbol{\Sigma})$ | Average Huber loss, $\mathcal{H}(\boldsymbol{\Sigma}) = \frac{1}{m}\sum_{i=1}^m \mathcal{H}_i(\boldsymbol{\Sigma})$. |
| $a$ | Robustification parameter in the Huber loss (Definition 1). |
| $\mathcal{P}(\boldsymbol{\Sigma})$ | Nonconvex sparsity-inducing penalty in problem (2). |
| $p_\lambda(\cdot)$ | Elementwise penalty function with tuning parameter $\lambda > 0$. |
| $q(\cdot)$ | Convex function such that $p_\lambda(x) = \lambda|x| - q(x)$. |
| $\lambda$ | Regularization parameter controlling sparsity. |
| $b$ | Threshold parameter in the nonconvex penalty (cf. Assumption 3). |
| $\tau$ | Log-determinant regularization parameter in problem (2). |
| $\mathcal{F}_i(\boldsymbol{\Sigma})$ | Smooth part of the local objective at agent $i$. |
| $\mathcal{F}(\boldsymbol{\Sigma})$ | Smooth part of the centralized objective, $\mathcal{F}(\boldsymbol{\Sigma}) = \frac{1}{m}\sum_{i=1}^m \mathcal{F}_i(\boldsymbol{\Sigma})$. |
| $\mathcal{L}(\boldsymbol{\Sigma})$ | Centralized objective in (2): $\mathcal{H}(\boldsymbol{\Sigma}) - \tau\log\det(\boldsymbol{\Sigma}) + \mathcal{P}(\boldsymbol{\Sigma})$. |
| $L_q$ | Lipschitz constant of $q'$ in Assumption 3(e). |
| $L$ | Lipschitz constant of $\nabla F$ on the invariant set $\mathbb{B}$. |
| $\mu$ | Strong convexity parameter of $F$ on the invariant set $\mathbb{B}$. |

**Algorithmic parameters and distributed notation**

| Symbol | Description |
|---|---|
| $\gamma$ | Proximal step size in Algorithms 1–2. |
| $\theta$ | Gradient-tracking parameter in the decentralized algorithm. |
| $\mathbf{Y}_i^{(t)}$ | Local gradient-tracking variable at agent $i$ and iteration $t$. |
| $\bar{\mathbf{Y}}^{(t)}$ | Network-averaged gradient tracker, $\bar{\mathbf{Y}}^{(t)} = \frac{1}{m}\sum_{i=1}^m \mathbf{Y}_i^{(t)}$. |
| $\boldsymbol{\Sigma}_\diamond^{(t)}$ | Stacked local covariances, $\boldsymbol{\Sigma}_\diamond^{(t)} = [\boldsymbol{\Sigma}_1^{(t)};\ldots;\boldsymbol{\Sigma}_m^{(t)}]$. |
| $\mathbf{Y}_\diamond^{(t)}$ | Stacked gradient-tracking variables, $\mathbf{Y}_\diamond^{(t)} = [\mathbf{Y}_1^{(t)};\ldots;\mathbf{Y}_m^{(t)}]$. |
| $\mathbf{E}_{\boldsymbol{\Sigma}}^{(t)}$ | Consensus error in covariances, $\mathbf{E}_{\boldsymbol{\Sigma}}^{(t)} = \boldsymbol{\Sigma}_\diamond^{(t)} - \mathbf{1}\otimes\bar{\boldsymbol{\Sigma}}^{(t)}$. |
| $\mathbf{E}_{\mathbf{Y}}^{(t)}$ | Consensus error in gradient trackers, $\mathbf{E}_{\mathbf{Y}}^{(t)} = \mathbf{Y}_\diamond^{(t)} - \mathbf{1}\otimes\bar{\mathbf{Y}}^{(t)}$. |
| $\mathbf{D}_i^{(t)}$ | Local update increment, $\mathbf{D}_i^{(t)} = \boldsymbol{\Sigma}_i^{(t+\frac{1}{2})} - \boldsymbol{\Sigma}_i^{(t)}$. |
| $\mathbf{D}_\diamond^{(t)}$ | Stacked update increments, $\mathbf{D}_\diamond^{(t)} = [\mathbf{D}_1^{(t)};\ldots;\mathbf{D}_m^{(t)}]$. |
| $\underline{r},\bar{r}$ | Eigenvalue bounds defining invariant regions for the iterates. |
| $\mathbb{A}$ | Compact invariant set $\mathbb{A} = \{\boldsymbol{\Sigma}\in\mathbb{S}_{++}^d : \underline{r}\,\mathbf{I}\preceq\boldsymbol{\Sigma}\preceq\bar{r}\,\mathbf{I}\}$. |
| $\mathbb{B}$ | Enlarged invariant set on which $F$ is strongly convex and Lipschitz smooth. |
| $\kappa$ | Condition number $\kappa = L/\mu$ of $F$ over $\mathbb{B}$. |
| $\kappa_r$ | Refined condition number appearing in the linear-rate bounds (Corollaries 1–2). |
| $\mathcal{V}(\cdot)$ | Potential (Lyapunov) function used in the analysis of the decentralized algorithm. |

**Statistical quantities**

| Symbol | Description |
|---|---|
| $\sigma > 0$ | Uniform moment bound in Assumption 2. |
| $\nu > 0$ | Order parameter of the finite moment in Assumption 2. |
| $K$ | Moment bound $K = \max\{\sigma^2, (2\sigma)^{2(1+\nu)}\}$. |
| $\mu_0$ | Local strong convexity parameter in Theorem 3. |
| $c_a, c_\tau, c_\lambda, c_q, c_N$ | Positive universal constants in Theorem 3. |
| $C_s$ | Constant in the Frobenius error bound of $\boldsymbol{\Sigma}^b$. |
| $c, C, C_1, C_2, \ldots$ | Generic positive constants (values may change from line to line). |

**Linear algebra and probability notation**

| Symbol | Description |
|---|---|
| $A_{ij}$ or $[A]_{ij}$ | $(i,j)$-th entry of a matrix $A$. |
| $\mathbf{1}$ | All-ones vector; $\mathbf{0}$: all-zero matrix; $\mathbf{I}$: identity matrix. |
| $\lambda_{\min}(A), \lambda_{\max}(A)$ | Smallest / largest eigenvalues of a symmetric matrix $A$. |
| $\lambda_k(A)$ | $k$-th largest eigenvalue of $A$. |
| $\|A\|_F, \|A\|_1, \|A\|_{\max}$ | Frobenius, entrywise $\ell_1$, and elementwise max norms of $A$. |
| $\|A\|_2, \|A\|_\infty$ | Spectral norm and induced $\ell_\infty$ (row-sum) norm of $A$. |
| $A \preceq B$ | Loewner order: $B - A$ is positive semidefinite. |
| $A \succ 0$ / $A \succeq 0$ | Positive definite / positive semidefinite matrix. |

**Table 2 (continued)**

| Symbol | Description |
|---|---|
| $\langle A, B \rangle$ | Matrix inner product $\mathrm{trace}(A^\top B)$. |
| $f(x) = O(g(x))$ | Big-O notation: $\|f(x)\| \leq Cg(x)$ for some $C > 0$. |
| $\nabla f(x), \nabla^2 f(x)$ | Gradient and Hessian of $f$ at $x$. |
| $\partial f(x)$ | Subdifferential of a convex function $f$ at $x$. |
| $\otimes$ | Kronecker product. |
| $\mathrm{P}(\cdot), \mathrm{E}[\cdot]$ | Probability and expectation operators. |

# C  PROOFS OF MAIN THEORETICAL RESULTS

## C.1  PROOF OF THEOREM 1

We prove Theorem 1 in the following steps: 1) we show that the objective of problem 2 is coercive, and hence problem 2 admits a solution; 2) we show that there exists a set $\mathbb{A} = \{\boldsymbol{\Sigma} \mid \underline{r}\mathbf{I} \preceq \boldsymbol{\Sigma} \preceq \overline{r}\mathbf{I}\}$, with constants $\underline{r} \leq \overline{r}$, such that the sequence $\{\boldsymbol{\Sigma}^{(t)}\}$ generated by Algorithm 1 remains within $\mathbb{A}$, and that the objective function is strong convex and Lipschitz smooth over this set; 3) we establish the linear convergence of Algorithm 1 based on the aforementioned properties.

### C.1.1  EXISTENCE OF SOLUTIONS

We show that the objective function $\mathcal{L}$ is coercive over $\boldsymbol{\Sigma} \succ \mathbf{0}$; therefore, problem 2 admits a solution $\widehat{\boldsymbol{\Sigma}}$.

**Proposition 1.** *The objective function $\mathcal{L}$ is coercive over $\boldsymbol{\Sigma} \succ \mathbf{0}$; that is,*

$$\lim_{\lambda_{\min}(\boldsymbol{\Sigma}) \to 0} \mathcal{L}(\boldsymbol{\Sigma}) = +\infty \quad \text{and} \quad \lim_{\lambda_{\max}(\boldsymbol{\Sigma}) \to +\infty} \mathcal{L}(\boldsymbol{\Sigma}) = +\infty.$$

*Proof.* For $x \geq 0$ and some $x_0 \in [0, x]$, the Lagrange mean-value theorem gives $q(x) - q(0) = q'(x_0)x$. By (c) and (d) in Assumption 3, it follows that

$$q(x) \leq \left|q'(x_0)x\right| \leq \lambda|x|,$$

and hence

$$p(x) = \lambda|x| - q(x) \geq \lambda|x| - \lambda|x| = 0.$$

Therefore, based on Definition 1, we have

$$\mathcal{L}(\boldsymbol{\Sigma}) \geq \frac{1}{m} \sum_{i=1}^{m} \mathcal{H}_i(\boldsymbol{\Sigma}) - \tau \log \det(\boldsymbol{\Sigma})$$

$$= \frac{1}{N} \sum_{j=1}^{N} \sum_{k=1}^{d} \sum_{l=1}^{d} h(\Sigma_{kl} - x_{jk}x_{jl}) - \tau \log \det(\boldsymbol{\Sigma})$$

$$\geq \frac{1}{N} \sum_{j=1}^{N} \sum_{k=1}^{d} \sum_{l=1}^{d} a\left|\Sigma_{kl} - x_{jk}x_{jl}\right| - \frac{1}{2}d^2 a^2 - \tau \log \det(\boldsymbol{\Sigma})$$

$$\geq \frac{1}{N} \sum_{j=1}^{N} \sum_{k=1}^{d} \sum_{l=1}^{d} a\left|\Sigma_{kl}\right| - a\frac{1}{N} \sum_{j=1}^{N} \sum_{k=1}^{d} \sum_{l=1}^{d} \left|x_{jk}x_{jl}\right| - \frac{1}{2}d^2 a^2 - \tau \log \det(\boldsymbol{\Sigma})$$

$$= a\left\|\boldsymbol{\Sigma}\right\|_1 - a\frac{1}{N} \sum_{j=1}^{N} \left\|\mathbf{x}_j \mathbf{x}_j^\top\right\|_1 - \frac{1}{2}d^2 a^2 - \tau \log \det(\boldsymbol{\Sigma}).$$

Since $\left\|\boldsymbol{\Sigma}\right\|_1 \geq \sum_{k=1}^{d} \lambda_k(\boldsymbol{\Sigma})$, we have

$$\mathcal{L}(\boldsymbol{\Sigma}) \geq \sum_{k=1}^{d} \left(a\lambda_k(\boldsymbol{\Sigma}) - \tau \log \lambda_k(\boldsymbol{\Sigma})\right) - a\frac{1}{N} \sum_{j=1}^{N} \left\|\mathbf{x}_j \mathbf{x}_j^\top\right\|_1 - \frac{1}{2}d^2 a^2.$$

When $x = \frac{\tau}{a}$, $ax - \tau \log x$ reaches its minimum $\tau - \tau \log \frac{\tau}{a}$. Therefore, we have

$$\mathcal{L}\left(\boldsymbol{\Sigma}\right) \geq a\lambda_k\left(\boldsymbol{\Sigma}\right) - \tau \log \lambda_k\left(\boldsymbol{\Sigma}\right) + (d-1)\left(\tau - \tau \log \frac{\tau}{a}\right) - a\frac{1}{N}\sum_{j=1}^{N}\left\|\mathbf{x}_j\mathbf{x}_j^\top\right\|_1 - \frac{1}{2}d^2a^2. \quad (11)$$

The right side of equation 11 is a convex function with respect to $\lambda_k\left(\boldsymbol{\Sigma}\right)$. As $\lambda_k\left(\boldsymbol{\Sigma}\right) \to 0$ or $\lambda_k\left(\boldsymbol{\Sigma}\right) \to +\infty$, the right side of equation 11 diverges to infinity, and consequently, $\mathcal{L} \to +\infty$. Since $\mathcal{L}$ is continuous on $\boldsymbol{\Sigma} \succ \mathbf{0}$, by the Weierstrass extreme value theorem, the function attains its minimum $\widehat{\boldsymbol{\Sigma}}$. $\qquad \square$

### C.1.2 INVARIANT SET

According to Proposition 1, for any initialization $\boldsymbol{\Sigma}^{(0)}$, there exist constants $\overline{r} \geq \underline{r} > 0$ such that the sublevel set $\left\{\boldsymbol{\Sigma} \mid \mathcal{L}\left(\boldsymbol{\Sigma}\right) \leq \mathcal{L}\left(\boldsymbol{\Sigma}^{(0)}\right)\right\} \subseteq \mathbb{A} = \left\{\boldsymbol{\Sigma} \mid \underline{r}\mathbf{I} \preceq \boldsymbol{\Sigma} \preceq \overline{r}\mathbf{I}\right\}$. We then aim to find sufficient conditions under which, if $\boldsymbol{\Sigma}^{(t)} \in \mathbb{A}$, the next iterate $\boldsymbol{\Sigma}^{(t+1)} \in \mathbb{A}$.

**Proposition 2.** *For $\boldsymbol{\Sigma}^{(t)} \in \mathbb{A}$, if*

$$\gamma \geq \frac{2}{\underline{r}}\left(ad + \tau\frac{\sqrt{d}}{\underline{r}} + 2\lambda d\right),$$

*we have $\boldsymbol{\Sigma}^{(t+1)} \in \mathbb{B}$, where $\mathbb{B} = \left\{\boldsymbol{\Sigma} \mid \left(\underline{r}/2\right)\mathbf{I} \preceq \boldsymbol{\Sigma} \preceq \left(\overline{r} + \underline{r}/2\right)\mathbf{I}\right\}$.*

*Proof.* Define $\mathcal{U}^{(t)}\left(\boldsymbol{\Sigma}\right) = \mathcal{F}\left(\boldsymbol{\Sigma}^{(t)}\right) + \left\langle\nabla\mathcal{F}\left(\boldsymbol{\Sigma}^{(t)}\right), \boldsymbol{\Sigma} - \boldsymbol{\Sigma}^{(t)}\right\rangle + \frac{\gamma}{2}\left\|\boldsymbol{\Sigma} - \boldsymbol{\Sigma}^{(t)}\right\|_F^2$. Since $\mathcal{U}^{(t)}$ is $\gamma$-strongly convex and $\lambda\left\|\boldsymbol{\Sigma}\right\|_{1,\text{off}}$ is convex, for any $\boldsymbol{\Phi} \in \partial\lambda\left\|\boldsymbol{\Sigma}^{(t)}\right\|_{1,\text{off}}$ we have

$$\mathcal{U}^{(t)}\left(\boldsymbol{\Sigma}^{(t+1)}\right) + \lambda\left\|\boldsymbol{\Sigma}^{(t+1)}\right\|_{1,\text{off}}$$

$$\geq \mathcal{U}^{(t)}\left(\boldsymbol{\Sigma}^{(t)}\right) + \lambda\left\|\boldsymbol{\Sigma}^{(t)}\right\|_{1,\text{off}} + \left\langle\nabla\mathcal{U}^{(t)}\left(\boldsymbol{\Sigma}^{(t)}\right) + \boldsymbol{\Phi}, \boldsymbol{\Sigma}^{(t+1)} - \boldsymbol{\Sigma}^{(t)}\right\rangle + \frac{\gamma}{2}\left\|\boldsymbol{\Sigma}^{(t+1)} - \boldsymbol{\Sigma}^{(t)}\right\|_F^2$$

$$= \mathcal{U}^{(t)}\left(\boldsymbol{\Sigma}^{(t)}\right) + \lambda\left\|\boldsymbol{\Sigma}^{(t)}\right\|_{1,\text{off}} - \frac{1}{2\gamma}\left\|\nabla\mathcal{U}^{(t)}\left(\boldsymbol{\Sigma}^{(t)}\right) + \boldsymbol{\Phi}\right\|_F^2$$

$$+ \frac{\gamma}{2}\left\|\frac{1}{\gamma}\left(\nabla\mathcal{U}^{(t)}\left(\boldsymbol{\Sigma}^{(t)}\right) + \boldsymbol{\Phi}\right) + \boldsymbol{\Sigma}^{(t+1)} - \boldsymbol{\Sigma}^{(t)}\right\|_F^2$$

$$\geq \mathcal{U}^{(t)}\left(\boldsymbol{\Sigma}^{(t)}\right) + \lambda\left\|\boldsymbol{\Sigma}^{(t)}\right\|_{1,\text{off}} - \frac{1}{2\gamma}\left\|\nabla\mathcal{U}^{(t)}\left(\boldsymbol{\Sigma}^{(t)}\right) + \boldsymbol{\Phi}\right\|_F^2. \quad (12)$$

Meanwhile, according to the first-order optimality condition of equation 4, there exists a $\boldsymbol{\Psi} \in \partial\lambda\left\|\boldsymbol{\Sigma}^{(t+1)}\right\|_{1,\text{off}}$ such that

$$\left\langle\nabla\mathcal{U}^{(t)}\left(\boldsymbol{\Sigma}^{(t+1)}\right) + \boldsymbol{\Psi}, \boldsymbol{\Sigma}^{(t)} - \boldsymbol{\Sigma}^{(t+1)}\right\rangle \geq 0.$$

Then similar to equation 12, we have

$$\mathcal{U}^{(t)}\left(\boldsymbol{\Sigma}^{(t)}\right) + \lambda\left\|\boldsymbol{\Sigma}^{(t)}\right\|_{1,\text{off}}$$

$$\geq \mathcal{U}^{(t)}\left(\boldsymbol{\Sigma}^{(t+1)}\right) + \lambda\left\|\boldsymbol{\Sigma}^{(t+1)}\right\|_{1,\text{off}} + \left\langle\nabla\mathcal{U}^{(t)}\left(\boldsymbol{\Sigma}^{(t+1)}\right) + \boldsymbol{\Psi}, \boldsymbol{\Sigma}^{(t)} - \boldsymbol{\Sigma}^{(t+1)}\right\rangle$$

$$+ \frac{\gamma}{2}\left\|\boldsymbol{\Sigma}^{(t)} - \boldsymbol{\Sigma}^{(t+1)}\right\|_F^2$$

$$\geq \mathcal{U}^{(t)}\left(\boldsymbol{\Sigma}^{(t+1)}\right) + \lambda\left\|\boldsymbol{\Sigma}^{(t+1)}\right\|_{1,\text{off}} + \frac{\gamma}{2}\left\|\boldsymbol{\Sigma}^{(t)} - \boldsymbol{\Sigma}^{(t+1)}\right\|_F^2. \quad (13)$$

Combining equation 12 and equation 13, we have

$$\left\| \boldsymbol{\Sigma}^{(t)} - \boldsymbol{\Sigma}^{(t+1)} \right\|_F$$

$$\leq \frac{1}{\gamma} \left\| \nabla \mathcal{U}^{(t)} \left( \boldsymbol{\Sigma}^{(t)} \right) + \boldsymbol{\Phi} \right\|_F \leq \frac{1}{\gamma} \left\| \nabla \mathcal{F} \left( \boldsymbol{\Sigma}^{(t)} \right) \right\|_F + \frac{1}{\gamma} \left\| \boldsymbol{\Phi} \right\|_F$$

$$\leq \frac{1}{\gamma} \left\| \frac{1}{m} \sum_{i=1}^m \nabla \mathcal{H}_i \left( \boldsymbol{\Sigma}^{(t)} \right) \right\|_F + \frac{\tau}{\gamma} \left\| \left( \boldsymbol{\Sigma}^{(t)} \right)^{-1} \right\|_F + \frac{1}{\gamma} \left\| \nabla \mathcal{Q} \left( \boldsymbol{\Sigma}^{(t)} \right) \right\|_F + \frac{1}{\gamma} \left\| \boldsymbol{\Phi} \right\|_F.$$

According to Definition 1, we have $[\nabla \mathcal{H}_i (\boldsymbol{\Sigma})]_{kl} \leq a$ and hence $\left\| \frac{1}{m} \sum_{i=1}^m \nabla \mathcal{H}_i (\boldsymbol{\Sigma}) \right\|_F \leq ad$. Since $\boldsymbol{\Sigma}^{(t)} \in \mathbb{A}$, we have $\left\| \left( \boldsymbol{\Sigma}^{(t)} \right)^{-1} \right\|_F \leq \sqrt{d}/\underline{r}$. Due to Assumption 3 (d), we have $\left\| \nabla \mathcal{Q} \left( \boldsymbol{\Sigma}^{(t)} \right) \right\|_F \leq \lambda d$. As $\boldsymbol{\Phi} \in \partial \lambda \left\| \boldsymbol{\Sigma}^{(t)} \right\|_{1,\mathrm{off}}$, we have $\| \boldsymbol{\Phi} \|_F \leq \lambda d$. Therefore, we have

$$\left\| \boldsymbol{\Sigma}^{(t)} - \boldsymbol{\Sigma}^{(t+1)} \right\|_F \leq \frac{1}{\gamma} \left( ad + \frac{\tau \sqrt{d}}{r} + 2\lambda d \right).$$

Since $\gamma \geq 2 \left( ad + \tau \sqrt{d}/\underline{r} + 2\lambda d \right) / \underline{r}$, we have $\left\| \boldsymbol{\Sigma}^{(t)} - \boldsymbol{\Sigma}^{(t+1)} \right\|_F \leq \underline{r}/2$ and hence $\boldsymbol{\Sigma}^{(t+1)} \in \mathbb{B}$. $\square$

Based on Proposition 2, we have $\boldsymbol{\Sigma}^{(t)}, \boldsymbol{\Sigma}^{(t+1)} \in \mathbb{B}$, and hence $\mathcal{F}$ is strongly convex and Lipschitz smooth on $\mathbb{B}$.

**Proposition 3.** *Suppose that $\bar{r} + \underline{r}/2 < \sqrt{\tau/L_q}$. Then for any $\boldsymbol{\Sigma}_1, \boldsymbol{\Sigma}_2 \in \mathbb{B}$, we have*

$$\| \nabla \mathcal{F} (\boldsymbol{\Sigma}_1) - \nabla \mathcal{F} (\boldsymbol{\Sigma}_2) \|_F \leq L \| \boldsymbol{\Sigma}_1 - \boldsymbol{\Sigma}_2 \|_F,$$
$$\| \nabla \mathcal{F} (\boldsymbol{\Sigma}_1) - \nabla \mathcal{F} (\boldsymbol{\Sigma}_2) \|_F \geq \mu \| \boldsymbol{\Sigma}_1 - \boldsymbol{\Sigma}_2 \|_F,$$

*where $L = 1 + 4\tau \underline{r}^{-2}$ and $\mu = \tau \left( \bar{r} + \underline{r}/2 \right)^{-2} - L_q$.*

*Proof.* Since $\boldsymbol{\Sigma}_1 \in \mathbb{B}$, we have

$$\nabla^2 \left( -\tau \log \det (\boldsymbol{\Sigma}_1) \right) = \tau \boldsymbol{\Sigma}_1^{-1} \otimes \boldsymbol{\Sigma}_1^{-1} \succeq \frac{\tau}{\left( \bar{r} + \underline{r}/2 \right)^2} \mathbf{I},$$

and hence $-\tau \log \det (\boldsymbol{\Sigma}_1)$ is $\tau \left( \bar{r} + \underline{r}/2 \right)^{-2}$-strongly convex. Then due to the convexity of Huber loss, Assumption 3 (e), and $\bar{r} + \underline{r}/2 < \sqrt{\tau/L_q}$, we have

$$\| \nabla \mathcal{F} (\boldsymbol{\Sigma}_1) - \nabla \mathcal{F} (\boldsymbol{\Sigma}_2) \|_F$$

$$\geq \| \nabla \left( -\tau \log \det (\boldsymbol{\Sigma}_1) \right) - \nabla \left( -\tau \log \det (\boldsymbol{\Sigma}_2) \right) \|_F - \left\| \frac{1}{m} \sum_{i=1}^m \nabla \mathcal{H}_i (\boldsymbol{\Sigma}_1) - \frac{1}{m} \sum_{i=1}^m \nabla \mathcal{H}_i (\boldsymbol{\Sigma}_2) \right\|_F$$

$$- \| \nabla \mathcal{Q} (\boldsymbol{\Sigma}_1) - \nabla \mathcal{Q} (\boldsymbol{\Sigma}_2) \|_F$$

$$\geq \left( \frac{\tau}{\left( \bar{r} + \underline{r}/2 \right)^2} - L_q \right) \| \boldsymbol{\Sigma}_1 - \boldsymbol{\Sigma}_2 \|_F > 0.$$

For smoothness, according to Definition 1, we have

$$h'(x) = \begin{cases} x, & |x| \leq a, \\ a\,\mathrm{sign}(x), & |x| > a, \end{cases}$$

and it is easy to verify that $|h'(x) - h'(y)| \leq |x - y|$ for any $x$ and $y$. Moreover, we have

$$\nabla^2 \left( -\tau \log \det (\boldsymbol{\Sigma}_1) \right) = \tau \boldsymbol{\Sigma}_1^{-1} \otimes \boldsymbol{\Sigma}_1^{-1} \preceq \frac{4\tau}{\underline{r}^2} \mathbf{I}.$$

Due to the convexity of $\mathcal{Q}$ and $\bar{r} + \underline{r}/2 < \sqrt{\tau/L_q}$, we have

$$\mathbf{0} \prec \nabla^2 \left( -\tau \log \det (\boldsymbol{\Sigma}_1) - \mathcal{Q} (\boldsymbol{\Sigma}_1) \right) = \nabla^2 \left( -\tau \log \det (\boldsymbol{\Sigma}_1) \right) - \nabla^2 \mathcal{Q} (\boldsymbol{\Sigma}_1) \preceq \frac{4\tau}{\underline{r}^2} \mathbf{I},$$

and hence,

$$\|\nabla \mathcal{F}\left(\mathbf{\Sigma}_1\right) - \nabla \mathcal{F}\left(\mathbf{\Sigma}_2\right)\|_F$$

$$\leq \frac{1}{m} \sum_{i=1}^{m} \|\nabla \mathcal{H}_i\left(\mathbf{\Sigma}_1\right) - \nabla \mathcal{H}_i\left(\mathbf{\Sigma}_2\right)\|_F$$

$$+ \|\nabla\left(-\tau \log\det\left(\mathbf{\Sigma}_1\right) - \mathcal{Q}\left(\mathbf{\Sigma}_1\right)\right) - \nabla\left(-\tau \log\det\left(\mathbf{\Sigma}_2\right) - \mathcal{Q}\left(\mathbf{\Sigma}_2\right)\right)\|_F$$

$$\leq \left(1 + \frac{4\tau}{\underline{r}^2}\right) \|\mathbf{\Phi} - \mathbf{\Sigma}_2\|_F.$$

$\square$

Then based on Propositions 2 and 3, we can prove that for any $\mathbf{\Sigma}^{(t)} \in \mathbb{A}$, we have $\mathbf{\Sigma}^{(t+1)} \in \mathbb{A}$.

**Proposition 4.** *Based on Propositions 2 and 3, we have*

$$\mathcal{L}\left(\mathbf{\Sigma}^{(t)}\right) - \mathcal{L}\left(\mathbf{\Sigma}^{(t+1)}\right) \geq \left(\gamma - \frac{L}{2}\right) \left\|\mathbf{\Sigma}^{(t)} - \mathbf{\Sigma}^{(t+1)}\right\|_F^2.$$

*Proof.* According to Proposition 3, we have

$$\mathcal{F}\left(\mathbf{\Sigma}^{(t)}\right) - \mathcal{F}\left(\mathbf{\Sigma}^{(t+1)}\right) \geq \left\langle\nabla\mathcal{F}\left(\mathbf{\Sigma}^{(t)}\right), \mathbf{\Sigma}^{(t)} - \mathbf{\Sigma}^{(t+1)}\right\rangle - \frac{L}{2}\left\|\mathbf{\Sigma}^{(t)} - \mathbf{\Sigma}^{(t+1)}\right\|_F^2. \quad (14)$$

Due to the first-order optimality condition of equation 4, we have

$$\gamma\mathbf{\Sigma}^{(t)} - \nabla\mathcal{F}\left(\mathbf{\Sigma}^{(t)}\right) - \gamma\mathbf{\Sigma}^{(t+1)} \in \partial\lambda\left\|\mathbf{\Sigma}^{(t+1)}\right\|_{1,\text{off}}.$$

Then according to the definition of subgradient, we have

$$\lambda\left\|\mathbf{\Sigma}^{(t)}\right\|_{1,\text{off}} - \lambda\left\|\mathbf{\Sigma}^{(t+1)}\right\|_{1,\text{off}} \geq \left\langle\gamma\mathbf{\Sigma}^{(t)} - \nabla\mathcal{F}\left(\mathbf{\Sigma}^{(t)}\right) - \gamma\mathbf{\Sigma}^{(t+1)}, \mathbf{\Sigma}^{(t)} - \mathbf{\Sigma}^{(t+1)}\right\rangle$$

$$= \left\langle\nabla\mathcal{F}\left(\mathbf{\Sigma}^{(t)}\right), \mathbf{\Sigma}^{(t+1)} - \mathbf{\Sigma}^{(t)}\right\rangle + \gamma\left\|\mathbf{\Sigma}^{(t)} - \mathbf{\Sigma}^{(t+1)}\right\|_F^2. \quad (15)$$

Combining equation 14 and equation 15, we have the desired result. $\square$

Proposition 4 indicates that $\mathcal{L}\left(\mathbf{\Sigma}^{(t)}\right) - \mathcal{L}\left(\mathbf{\Sigma}^{(t+1)}\right) \geq 0$ given $\mathbf{\Sigma}^{(t)} \in \mathbb{A}$ and $\gamma \geq \max\left\{\frac{L}{2}, \frac{2}{\underline{r}}\left(ad + \tau\frac{\sqrt{d}}{\underline{r}} + 2\lambda d\right)\right\}$. Then inducting from $\mathbf{\Sigma}^{(0)}$ gives $\left\{\mathbf{\Sigma}^{(t)}\right\} \subset \mathbb{A}$.

### C.1.3 LINEAR CONVERGENCE

Finally, we prove the linear convergence result in Theorem 1. According to Proposition 3, the strong convexity of $\mathcal{F}$ with parameter $\mu$ implies that

$$\mathcal{F}\left(\widehat{\mathbf{\Sigma}}\right) - \mathcal{F}\left(\mathbf{\Sigma}^{(t)}\right) - \left\langle\nabla\mathcal{F}\left(\mathbf{\Sigma}^{(t)}\right), \widehat{\mathbf{\Sigma}} - \mathbf{\Sigma}^{(t)}\right\rangle \geq \frac{\mu}{2}\left\|\widehat{\mathbf{\Sigma}} - \mathbf{\Sigma}^{(t)}\right\|_F^2.$$

Using the Lipschitz smoothness of $\mathcal{F}$ with constant $L$, it follows that

$$\mathcal{F}\left(\mathbf{\Sigma}^{(t+1)}\right) - \mathcal{F}\left(\mathbf{\Sigma}^{(t)}\right) - \left\langle\nabla\mathcal{F}\left(\mathbf{\Sigma}^{(t)}\right), \mathbf{\Sigma}^{(t+1)} - \mathbf{\Sigma}^{(t)}\right\rangle \leq \frac{L}{2}\left\|\mathbf{\Sigma}^{(t+1)} - \mathbf{\Sigma}^{(t)}\right\|_F^2.$$

Combining the above two inequalities, we obtain

$$0 \leq \mathcal{F}\left(\widehat{\mathbf{\Sigma}}\right) - \mathcal{F}\left(\mathbf{\Sigma}^{(t+1)}\right) + \frac{L}{2}\left\|\mathbf{\Sigma}^{(t+1)} - \mathbf{\Sigma}^{(t)}\right\|_F^2 - \frac{\mu}{2}\left\|\widehat{\mathbf{\Sigma}} - \mathbf{\Sigma}^{(t)}\right\|_F^2$$

$$- \left\langle\nabla\mathcal{F}\left(\mathbf{\Sigma}^{(t)}\right), \widehat{\mathbf{\Sigma}} - \mathbf{\Sigma}^{(t+1)}\right\rangle. \quad (16)$$

Applying the first-order optimality condition for equation 4, there exists a $\mathbf{\Phi} \in \partial\lambda\left\|\mathbf{\Sigma}^{(t+1)}\right\|_{1,\text{off}}$ such that

$$\left\langle\nabla\mathcal{F}\left(\mathbf{\Sigma}^{(t)}\right) + \gamma\left(\mathbf{\Sigma}^{(t+1)} - \mathbf{\Sigma}^{(t)}\right) + \mathbf{\Phi}, \widehat{\mathbf{\Sigma}} - \mathbf{\Sigma}^{(t+1)}\right\rangle \geq 0.$$

Due to the convexity of $\lambda \left\| \cdot \right\|_{1,\mathrm{off}}$, we have

$$\lambda \left\| \widehat{\boldsymbol{\Sigma}} \right\|_{1,\mathrm{off}} - \lambda \left\| \boldsymbol{\Sigma}^{(t+1)} \right\|_{1,\mathrm{off}} - \left\langle \boldsymbol{\Phi}, \widehat{\boldsymbol{\Sigma}} - \boldsymbol{\Sigma}^{(t+1)} \right\rangle \geq 0.$$

Summing the two inequalities above, we obtain:

$$\lambda \left\| \widehat{\boldsymbol{\Sigma}} \right\|_{1,\mathrm{off}} - \lambda \left\| \boldsymbol{\Sigma}^{(t+1)} \right\|_{1,\mathrm{off}} + \left\langle \nabla \mathcal{F}\left(\boldsymbol{\Sigma}^{(t)}\right) + \gamma \left(\boldsymbol{\Sigma}^{(t+1)} - \boldsymbol{\Sigma}^{(t)}\right), \widehat{\boldsymbol{\Sigma}} - \boldsymbol{\Sigma}^{(t+1)} \right\rangle \geq 0. \quad (17)$$

Combining equation 16 and equation 17 leads to

$$0 \leq \mathcal{L}\left(\widehat{\boldsymbol{\Sigma}}\right) - \mathcal{L}\left(\boldsymbol{\Sigma}^{(t+1)}\right) + \frac{L}{2} \left\| \boldsymbol{\Sigma}^{(t+1)} - \boldsymbol{\Sigma}^{(t)} \right\|_F^2 - \frac{\mu}{2} \left\| \widehat{\boldsymbol{\Sigma}} - \boldsymbol{\Sigma}^{(t)} \right\|_F^2$$
$$+ \gamma \left\langle \boldsymbol{\Sigma}^{(t+1)} - \boldsymbol{\Sigma}^{(t)}, \widehat{\boldsymbol{\Sigma}} - \boldsymbol{\Sigma}^{(t+1)} \right\rangle.$$

Since $\widehat{\boldsymbol{\Sigma}}$ is the minimizer of $\mathcal{L}$, we have

$$0 \leq \frac{L - 2\gamma}{2} \left\| \boldsymbol{\Sigma}^{(t+1)} - \boldsymbol{\Sigma}^{(t)} \right\|_F^2 - \frac{\mu}{2} \left\| \widehat{\boldsymbol{\Sigma}} - \boldsymbol{\Sigma}^{(t)} \right\|_F^2 + \gamma \left\langle \boldsymbol{\Sigma}^{(t+1)} - \boldsymbol{\Sigma}^{(t)}, \widehat{\boldsymbol{\Sigma}} - \boldsymbol{\Sigma}^{(t)} \right\rangle.$$

Rearranging the above inequality, we get

$$2 \left\langle \boldsymbol{\Sigma}^{(t+1)} - \boldsymbol{\Sigma}^{(t)}, \boldsymbol{\Sigma}^{(t)} - \widehat{\boldsymbol{\Sigma}} \right\rangle \leq \frac{L - 2\gamma}{\gamma} \left\| \boldsymbol{\Sigma}^{(t+1)} - \boldsymbol{\Sigma}^{(t)} \right\|_F^2 - \frac{\mu}{\gamma} \left\| \widehat{\boldsymbol{\Sigma}} - \boldsymbol{\Sigma}^{(t)} \right\|_F^2,$$

and hence

$$\left\| \boldsymbol{\Sigma}^{(t+1)} - \widehat{\boldsymbol{\Sigma}} \right\|_F^2 \leq \left\| \boldsymbol{\Sigma}^{(t+1)} - \boldsymbol{\Sigma}^{(t)} \right\|_F^2 + \left\| \boldsymbol{\Sigma}^{(t)} - \widehat{\boldsymbol{\Sigma}} \right\|_F^2 + 2 \left\langle \boldsymbol{\Sigma}^{(t+1)} - \boldsymbol{\Sigma}^{(t)}, \boldsymbol{\Sigma}^{(t)} - \widehat{\boldsymbol{\Sigma}} \right\rangle$$
$$\leq \left(1 - \frac{\mu}{\gamma}\right) \left\| \boldsymbol{\Sigma}^{(t)} - \widehat{\boldsymbol{\Sigma}} \right\|_F^2.$$

Backtracking to $\boldsymbol{\Sigma}^{(0)}$ and using $\gamma = c_\gamma L$, we can obtain the linear convergence result in Theorem 1.

### C.1.4 On the computation of $\overline{r}$ and $\underline{r}$

In the previous proof, we only established the existence of $\overline{r}$ and $\underline{r}$. One way to compute them is, as stated in Theorem 1, to use the inequality equation 11 obtained based on Proposition 1. Since the right hand side of the inequality equation 11 is a convex lower bound of the objective function $\mathcal{L}$, equation equation 1 necessarily admits two positive solutions, and choosing these solutions as $\overline{r}$ and $\underline{r}$ guarantees $\left\{ \boldsymbol{\Sigma} \mid \mathcal{L}\left(\boldsymbol{\Sigma}\right) \leq \mathcal{L}\left(\boldsymbol{\Sigma}^{(0)}\right) \right\} \subseteq \mathbb{A}$.

### C.2 Proof of Theorem 2

For notational simplicity, we introduce the following compact forms:

$$\boldsymbol{\Sigma}_{\diamond}^{(t)} = \left[\boldsymbol{\Sigma}_1^{(t)}; \boldsymbol{\Sigma}_2^{(t)}; \ldots; \boldsymbol{\Sigma}_m^{(t)}\right], \qquad \mathbf{E}_{\boldsymbol{\Sigma}}^{(t)} = \boldsymbol{\Sigma}^{(t)} - \mathbf{1} \otimes \bar{\boldsymbol{\Sigma}}^{(t)},$$

$$\mathbf{Y}_{\diamond}^{(t)} = \left[\mathbf{Y}_1^{(t)}; \mathbf{Y}_2^{(t)}; \ldots; \mathbf{Y}_m^{(t)}\right], \qquad \mathbf{E}_{\mathbf{Y}}^{(t)} = \mathbf{Y}^{(t)} - \mathbf{1} \otimes \bar{\mathbf{Y}}^{(t)},$$

$$\mathbf{D}_i^{(t)} = \boldsymbol{\Sigma}_i^{\left(t+\frac{1}{2}\right)} - \boldsymbol{\Sigma}_i^{(t)}, \qquad \mathbf{D}_{\diamond}^{(t)} = \left[\mathbf{D}_1^{(t)}; \mathbf{D}_2^{(t)}; \ldots; \mathbf{D}_m^{(t)}\right].$$

The overall proof strategy for Theorem 2 parallels that of Theorem 1. Specifically, the analysis hinges on a potential function $\mathcal{V}$ that combines the objective function with the consensus error as follows:

$$\mathcal{V}\left(\boldsymbol{\Sigma}_{\diamond}, \mathbf{Y}_{\diamond}\right) = \sum_{i=1}^{m} \mathcal{L}\left(\boldsymbol{\Sigma}_i\right) + c_1 \left\| \mathbf{E}_{\boldsymbol{\Sigma}} \right\|_F^2 + c_2 \left\| \mathbf{E}_{\mathbf{Y}} \right\|_F^2, \quad (18)$$

where $\boldsymbol{\Sigma}_{\diamond} = \left[\boldsymbol{\Sigma}_1; \ldots; \boldsymbol{\Sigma}_m\right]$ and $\mathbf{Y}_{\diamond} = \left[\mathbf{Y}_1; \ldots; \mathbf{Y}_m\right]$ are the matrices that stack all the local variables, $\mathbf{E}_{\boldsymbol{\Sigma}} = \boldsymbol{\Sigma}_{\diamond} - \mathbf{1} \otimes \bar{\boldsymbol{\Sigma}}$ and $\mathbf{E}_{\mathbf{Y}} = \mathbf{Y}_{\diamond} - \mathbf{1} \otimes \bar{\mathbf{Y}}$ is consensus error, $\bar{\boldsymbol{\Sigma}} = \frac{1}{m} \sum_{i=1}^{m} \boldsymbol{\Sigma}_i$, $\bar{\mathbf{Y}} = \frac{1}{m} \sum_{i=1}^{m} \mathbf{Y}_i$, and $c_1, c_2 > 0$. Compared to Theorem 1, we first prove the local properties based on $\mathcal{V}$.

### C.2.1 PRELIMINARIES

**Proposition 5.** *The potential function $\mathcal{V}$ is coercive for $\{\boldsymbol{\Sigma}_i\}_{i=1}^m$, i.e., $\lim_{\min\{\lambda_{\min}(\boldsymbol{\Sigma}_i)\}_{i=1}^m \to 0} \mathcal{V}(\boldsymbol{\Sigma}_\diamond, \mathbf{Y}_\diamond) = +\infty$ and $\lim_{\max\{\lambda_{\max}(\boldsymbol{\Sigma}_i)\}_{i=1}^m \to +\infty} \mathcal{V}(\boldsymbol{\Sigma}_\diamond, \mathbf{Y}_\diamond) = +\infty$. Meanwhile, we have $\lim_{\|\mathbf{E}_\mathbf{Y}\|_F \to +\infty} \mathcal{V}(\boldsymbol{\Sigma}_\diamond, \mathbf{Y}_\diamond) = +\infty$.*

*Proof.* For $\boldsymbol{\Sigma}_\diamond$, following similar steps of the proof of Proposition 1 and utilizing the non-negative nature of the consensus error, we have

$$\mathcal{V}(\boldsymbol{\Sigma}_\diamond, \mathbf{Y}_\diamond)$$
$$\geq \frac{1}{m} \sum_{i=1}^m \sum_{k=1}^d \left( a\lambda_k(\boldsymbol{\Sigma}_i) - \tau \log \lambda_k(\boldsymbol{\Sigma}_i) \right) - a\frac{1}{N} \sum_{j=1}^N \left\| \mathbf{x}_j \mathbf{x}_j^\top \right\|_1 - \frac{1}{2} d^2 a^2 \tag{19}$$

$$\geq \frac{1}{m} \left( a\lambda_k(\boldsymbol{\Sigma}_i) - \tau \log \lambda_k(\boldsymbol{\Sigma}_i) \right) + \left( d - \frac{1}{m} \right) \left( \tau - \tau \log \frac{\tau}{a} \right) - \frac{a}{N} \sum_{j=1}^N \left\| \mathbf{x}_j \mathbf{x}_j^\top \right\|_1 - \frac{1}{2} d^2 a^2$$

for any $i = 1, \ldots, m$, and hence $\mathcal{V}$ is coercive. For $\mathbf{Y}_\diamond$, we have

$$\mathcal{V}(\boldsymbol{\Sigma}_\diamond, \mathbf{Y}_\diamond) \geq b \|\mathbf{E}_\mathbf{Y}\|_F^2 + d\left( \tau - \tau \log \frac{\tau}{a} \right) - \frac{a}{N} \sum_{j=1}^N \left\| \mathbf{x}_j \mathbf{x}_j^\top \right\|_1 - \frac{1}{2} d^2 a^2, \tag{20}$$

and hence $\mathcal{V}$ is coercive with respect to $\mathbf{E}_\mathbf{Y}$. $\qquad\square$

Based on Proposition 5, for any initialization $\left( \boldsymbol{\Sigma}_\diamond^{(0)}, \mathbf{Y}_\diamond^{(0)} \right)$, there exist parameters $\overline{r} \geq \underline{r} > 0$ and $e > 0$ such that for any $(\boldsymbol{\Sigma}_\diamond, \mathbf{Y}_\diamond) \in \left\{ (\boldsymbol{\Sigma}_\diamond, \mathbf{Y}_\diamond) \mid \mathcal{V}(\boldsymbol{\Sigma}_\diamond, \mathbf{Y}_\diamond) \leq \mathcal{V}\left( \boldsymbol{\Sigma}_\diamond^{(0)}, \mathbf{Y}_\diamond^{(0)} \right) \right\}$, we have $\boldsymbol{\Sigma}_\diamond \in \mathbb{A} = \{ \boldsymbol{\Sigma}_\diamond \mid \underline{r}\mathbf{I} \preceq \boldsymbol{\Sigma}_i \preceq \overline{r}\mathbf{I}, i = 1, \ldots, m \}$ and $\|\mathbf{E}_\mathbf{Y}\|_F \leq \sqrt{e}$. Then we first introduce the following lemma.

**Lemma 1** (Xia et al. (2025)). *For a $L$-smooth function $f$, we have*

$$f\left( \sum_{i=1}^m a_i \mathbf{Y}_i \right) \geq \sum_{i=1}^m a_i f(\mathbf{Y}_i) - \frac{L}{2} \sum_{i=1}^{m-1} \sum_{j=i+1}^m a_i a_j \|\mathbf{Y}_i - \mathbf{Y}_j\|_F^2,$$

*where $\sum_{i=1}^m a_i = 1$ and $a_i \geq 0$ for all $i$.*

Based on Lemma 1, we can bound the gradient tracking variables $\mathbf{Y}_i$ for each agent.

**Proposition 6.** *For every $\left( \boldsymbol{\Sigma}_\diamond^{(t)}, \mathbf{Y}_\diamond^{(t)} \right)$ generated by Algorithm 2 such that $\left( \boldsymbol{\Sigma}_\diamond^{(t)}, \mathbf{Y}_\diamond^{(t)} \right) \in \left\{ (\boldsymbol{\Sigma}_\diamond, \mathbf{Y}_\diamond) \mid \mathcal{V}(\boldsymbol{\Sigma}_\diamond, \mathbf{Y}_\diamond) \leq \mathcal{V}\left( \boldsymbol{\Sigma}_\diamond^{(0)}, \mathbf{Y}_\diamond^{(0)} \right) \right\}$, we have*

$$\left\| \mathbf{Y}_i^{(t)} \right\|_F \leq \sqrt{m \left( ad + \frac{\sqrt{d}\tau}{\underline{r}} + d\lambda \right)^2 + e}.$$

*Proof.* Since $\mathbf{Y}_i^{(0)} = \nabla \mathcal{F}_i\left( \boldsymbol{\Sigma}_i^{(0)} \right)$ for all $i = 1, \ldots, m$, the update rule for $\bar{\mathbf{Y}}$ can be expressed as

$$\bar{\mathbf{Y}}^{(t+1)} = \frac{1}{m} \sum_{i=1}^m \nabla \mathcal{F}_i\left( \boldsymbol{\Sigma}_i^{(t+1)} \right). \tag{21}$$

Applying Lemma 1 and the 2-smooth nature of $\|\cdot\|_F^2$ to the above equation leads to

$$
\frac{1}{m} \sum_{i=1}^m \left\| \mathbf{Y}_i^{(t)} \right\|_F^2 \leq \left\| \frac{1}{m} \sum_{i=1}^m \mathbf{Y}_i^{(t)} \right\|_F^2 + \frac{1}{m^2} \sum_{i=1}^{m-1} \sum_{j=i+1}^m \left\| \mathbf{Y}_i^{(t)} - \mathbf{Y}_j^{(t)} \right\|_F^2
$$

$$
= \left\| \frac{1}{m} \sum_{i=1}^m \nabla \mathcal{F}_i \left( \mathbf{\Sigma}_i^{(t)} \right) \right\|_F^2 + \frac{1}{m^2} \sum_{i=1}^{m-1} \sum_{j=i+1}^m \left\| \mathbf{Y}_i^{(t)} - \mathbf{Y}_j^{(t)} \right\|_F^2
$$

$$
\leq \frac{1}{m} \sum_{i=1}^m \left\| \nabla \mathcal{F}_i \left( \mathbf{\Sigma}_i^{(t)} \right) \right\|_F^2 + \frac{1}{m} \left\| \mathbf{E}_{\mathbf{Y}}^{(t)} \right\|_F^2
$$

$$
\leq \frac{1}{m} \sum_{i=1}^m \left( \left\| \nabla \mathcal{L}_i \left( \mathbf{\Sigma}_i^{(t)} \right) \right\|_F + \tau \left\| \left( \mathbf{\Sigma}_i^{(t)} \right)^{-1} \right\|_F + \left\| \nabla \mathcal{Q} \left( \mathbf{\Sigma}_i^{(t)} \right) \right\|_F \right)^2 + \frac{1}{m} \left\| \mathbf{E}_{\mathbf{Y}}^{(t)} \right\|_F^2 .
$$

According to Definition 1, we have $\left[ \nabla \mathcal{L}_i \left( \mathbf{\Sigma}_i^{(t)} \right) \right]_{kl} \leq a$ and hence $\left\| \frac{1}{m} \sum_{i=1}^m \nabla \mathcal{L}_i \left( \mathbf{\Sigma}_i^{(t)} \right) \right\|_F \leq ad$. Since $\mathbf{\Sigma}_\diamond^{(t)} \in \mathbb{A}$, we have $\left\| \left( \mathbf{\Sigma}_i^{(t)} \right)^{-1} \right\|_F \leq \sqrt{d}/\underline{r}$. Due to Assumption 3 (d), we have $\left\| \nabla \mathcal{Q} \left( \mathbf{\Sigma}_i^{(t)} \right) \right\|_F \leq \lambda d$. Then since $\|\mathbf{E}_{\mathbf{Y}}\|_F \leq \sqrt{e}$, we have

$$
\left\| \mathbf{Y}_i^{(t)} \right\|_F \leq \sqrt{\sum_{i=1}^m \left\| \mathbf{Y}_i^{(t)} \right\|_F^2} \leq \sqrt{m \left( ad + \frac{\sqrt{d}\tau}{\underline{r}} + d\lambda \right)^2 + e}.
$$

$\square$

Based on Proposition 6, we can bound $\mathbf{\Sigma}_\diamond^{\left(t+\frac{1}{2}\right)}$ and $\mathbf{\Sigma}_\diamond^{(t+1)}$ for $\left( \mathbf{\Sigma}_\diamond^{(t)}, \mathbf{Y}_\diamond^{(t)} \right) \in \left\{ (\mathbf{\Sigma}_\diamond, \mathbf{Y}_\diamond) \mid \mathcal{V} (\mathbf{\Sigma}_\diamond, \mathbf{Y}_\diamond) \leq \mathcal{V} \left( \mathbf{\Sigma}_\diamond^{(0)}, \mathbf{Y}_\diamond^{(0)} \right) \right\}$.

**Proposition 7.** *Suppose that* $\left( \mathbf{\Sigma}_\diamond^{(t)}, \mathbf{Y}_\diamond^{(t)} \right) \in \left\{ (\mathbf{\Sigma}_\diamond, \mathbf{Y}_\diamond) \mid \mathcal{V} (\mathbf{\Sigma}_\diamond, \mathbf{Y}_\diamond) \leq \mathcal{V} \left( \mathbf{\Sigma}_\diamond^{(0)}, \mathbf{Y}_\diamond^{(0)} \right) \right\}$ *and*

$$
\gamma \geq \frac{2}{\underline{r}} \left( \sqrt{m \left( ad + \frac{\sqrt{d}\tau}{\underline{r}} + d\lambda \right)^2 + e} + \lambda d \right), \tag{22}
$$

*we have* $\mathbf{\Sigma}_\diamond^{\left(t+\frac{1}{2}\right)}, \mathbf{\Sigma}_\diamond^{(t+1)} \in \mathbb{B}$, *where* $\mathbb{B} = \{ \mathbf{\Sigma}_\diamond \mid \underline{r}\mathbf{I}/2 \preceq \mathbf{\Sigma}_i \preceq (\bar{r} + \underline{r}/2) \mathbf{I}, i = 1, \ldots, m \}$.

*Proof.* Following similar steps as the proof of Proposition 2, for any $i = 1, \ldots, m$, we have

$$
\left\| \mathbf{\Sigma}_i^{\left(t+\frac{1}{2}\right)} - \mathbf{\Sigma}_i^{(t)} \right\|_F \leq \frac{1}{\gamma} \left\| \mathbf{Y}_i^{(t)} \right\|_F + \frac{1}{\gamma} \lambda d.
$$

Then according to Proposition 6 and equation 22, we have

$$
\left\| \mathbf{\Sigma}_i^{\left(t+\frac{1}{2}\right)} - \mathbf{\Sigma}_i^{(t)} \right\|_F \leq \frac{1}{\gamma} \sqrt{m \left( ad + \frac{\sqrt{d}\tau}{\underline{r}} + d\lambda \right)^2 + e} + \frac{1}{\gamma} \lambda d \leq \frac{\underline{r}}{2},
$$

and hence $\mathbf{\Sigma}_\diamond^{\left(t+\frac{1}{2}\right)}, \mathbf{\Sigma}_\diamond^{(t+1)} \in \mathbb{B}$ according to the update rule equation 6. $\square$

Based on Proposition 7 and following similar steps in the proof of Proposition 3, we have the following proposition.

**Proposition 8.** *Suppose that* $\bar{r} + \underline{r}/2 < \sqrt{\tau/L_q}$. *Then for any* $\mathbf{\Sigma}_1, \mathbf{\Sigma}_2 \in \mathbb{B}$, *we have*

$$
\|\nabla \mathcal{F} (\mathbf{\Sigma}_1) - \nabla \mathcal{F} (\mathbf{\Sigma}_2)\|_F \leq L \|\mathbf{\Sigma}_1 - \mathbf{\Sigma}_2\|_F,
$$
$$
\|\nabla \mathcal{F} (\mathbf{\Sigma}_1) - \nabla \mathcal{F} (\mathbf{\Sigma}_2)\|_F \geq \mu \|\mathbf{\Sigma} - \mathbf{\Sigma}_2\|_F,
$$

*where* $L = 1 + 4\tau \underline{r}^{-2}$ *and* $\mu = \tau (\bar{r} + \underline{r}/2)^{-2} - L_q$. *Moreover, the local loss function of each agent* $i$ *is also L-smooth and $\mu$-strongly convex.*

Proposition 8 is obtained from Proposition 3 with minor modifications and is therefore omitted. Then based on Proposition 8, we have the following upper bound of the optimality gap.

**Proposition 9.** *Based on Propositions 7 and 8, there holds*

$$\sum_{i=1}^{m} \mathcal{L}\left(\mathbf{\Sigma}_i^{(t+1)}\right) \leq \sum_{i=1}^{m} \mathcal{L}\left(\mathbf{\Sigma}_i^{(t)}\right) + \frac{\theta}{2}\left(\gamma - \frac{L}{2}\theta\right)^{-1}\left(4L^2 \left\|\mathbf{E}_{\mathbf{\Sigma}}^{(t)}\right\|_F^2 + 2\left\|\mathbf{E}_{\mathbf{Y}}^{(t)}\right\|_F^2\right)$$

$$- \theta\left(\frac{\gamma}{2} - \frac{L}{4}\theta\right)\left\|\mathbf{D}_\diamond^{(t)}\right\|_F^2, \tag{23}$$

*where $\gamma \geq L$.*

*Proof.* Define $\tilde{\mathbf{\Sigma}}_i^{(t)} = \mathbf{\Sigma}_i^{(t)} + \theta\mathbf{D}_i^{(t)}$. Consider the Taylor expansion of $\mathcal{F}$

$$\mathcal{F}\left(\tilde{\mathbf{\Sigma}}_i^{(t)}\right) = \mathcal{F}\left(\mathbf{\Sigma}_i^{(t)}\right) + \left\langle \nabla\mathcal{F}\left(\mathbf{\Sigma}_i^{(t)}\right), \theta\mathbf{D}_i^{(t)}\right\rangle + \frac{1}{2}\mathrm{vec}\left(\theta\mathbf{D}_i^{(t)}\right)^\top \nabla^2\mathcal{F}\left(\tilde{\mathbf{\Sigma}}_i^{(t)}\right)\mathrm{vec}\left(\theta\mathbf{D}_i^{(t)}\right)$$

$$= \mathcal{F}\left(\mathbf{\Sigma}_i^{(t)}\right) + \left\langle \nabla\mathcal{F}\left(\mathbf{\Sigma}_i^{(t)}\right) - \mathbf{Y}_i^{(t)}, \theta\mathbf{D}_i^{(t)}\right\rangle + \left\langle \mathbf{Y}_i^{(t)}, \theta\mathbf{D}_i^{(t)}\right\rangle$$

$$+ \frac{1}{2}\mathrm{vec}\left(\theta\mathbf{D}_i^{(t)}\right)^\top \nabla^2\mathcal{F}\left(\tilde{\mathbf{\Sigma}}_i^{(t)}\right)\mathrm{vec}\left(\theta\mathbf{D}_i^{(t)}\right). \tag{24}$$

Because $\mathcal{U}$ is strongly convex with $\gamma$ and $\lambda\left\|\cdot\right\|_{1,\mathrm{off}}$ is convex, according to the first-order optimality condition, we have

$$\lambda\left\|\mathbf{\Sigma}_i^{(t)}\right\|_{1,\mathrm{off}} - \lambda\left\|\mathbf{\Sigma}_i^{\left(t+\frac{1}{2}\right)}\right\|_{1,\mathrm{off}} \geq \left\langle \mathbf{Y}_i^{(t)}, \mathbf{D}_i^{(t)}\right\rangle + \gamma\left\|\mathbf{D}_i^{(t)}\right\|_F^2. \tag{25}$$

Using the convexity of $\lambda\left\|\cdot\right\|_{1,\mathrm{off}}$, we have

$$\lambda\left\|\tilde{\mathbf{\Sigma}}_i^{(t)}\right\|_{1,\mathrm{off}} = \lambda\left\|\theta\mathbf{\Sigma}_i^{\left(t+\frac{1}{2}\right)} + (1-\theta)\mathbf{\Sigma}_i^{(t)}\right\|_{1,\mathrm{off}} \leq \theta\lambda\left\|\mathbf{\Sigma}_i^{\left(t+\frac{1}{2}\right)}\right\|_{1,\mathrm{off}} + (1-\theta)\lambda\left\|\mathbf{\Sigma}_i^{(t)}\right\|_{1,\mathrm{off}}. \tag{26}$$

Substituting equation 25 and equation 26 into equation 24, we have

$$\mathcal{F}\left(\tilde{\mathbf{\Sigma}}_i^{(t)}\right) \leq \mathcal{F}\left(\mathbf{\Sigma}_i^{(t)}\right) + \left\langle \nabla\mathcal{F}\left(\mathbf{\Sigma}_i^{(t)}\right) - \mathbf{Y}_i^{(t)}, \theta\mathbf{D}_i^{(t)}\right\rangle + \lambda\left\|\mathbf{\Sigma}_i^{(t)}\right\|_{1,\mathrm{off}} - \lambda\left\|\tilde{\mathbf{\Sigma}}_i^{(t)}\right\|_{1,\mathrm{off}}$$

$$- \theta\gamma\left\|\mathbf{D}_i^{(t)}\right\|_F^2 + \frac{1}{2}\mathrm{vec}\left(\theta\mathbf{D}_i^{(t)}\right)^\top \nabla^2\mathcal{F}\left(\tilde{\mathbf{\Sigma}}_i^{(t)}\right)\mathrm{vec}\left(\theta\mathbf{D}_i^{(t)}\right). \tag{27}$$

According to Proposition 8, we have

$$\nabla^2\mathcal{F}\left(\tilde{\mathbf{\Sigma}}_i^{(t)}\right) \preceq L\mathbf{I}. \tag{28}$$

Substituting equation 28 into equation 27, we have

$$\mathcal{F}\left(\tilde{\mathbf{\Sigma}}_i^{(t)}\right) \leq \mathcal{F}\left(\mathbf{\Sigma}_i^{(t)}\right) + \left\langle \nabla\mathcal{F}\left(\mathbf{\Sigma}_i^{(t)}\right) - \mathbf{Y}_i^{(t)}, \theta\mathbf{D}_i^{(t)}\right\rangle + \lambda\left\|\mathbf{\Sigma}_i^{(t)}\right\|_{1,\mathrm{off}} - \lambda\left\|\tilde{\mathbf{\Sigma}}_i^{(t)}\right\|_{1,\mathrm{off}}$$

$$- \theta\gamma\left\|\mathbf{D}_i^{(t)}\right\|_F^2 + \frac{L}{2}\theta^2\left\|\mathbf{D}_i^{(t)}\right\|_F^2$$

$$\leq \mathcal{F}\left(\mathbf{\Sigma}_i^{(t)}\right) + \lambda\left\|\mathbf{\Sigma}_i^{(t)}\right\|_{1,\mathrm{off}} - \lambda\left\|\tilde{\mathbf{\Sigma}}_i^{(t)}\right\|_{1,\mathrm{off}} - \theta\left(\gamma - \frac{L}{2}\theta\right)\left\|\mathbf{D}_i^{(t)}\right\|_F^2$$

$$+ \theta\left\|\mathbf{D}_i^{(t)}\right\|_F \left\|\nabla\mathcal{F}\left(\mathbf{\Sigma}_i^{(t)}\right) - \mathbf{Y}_i^{(t)}\right\|_F,$$

which equals to

$$\mathcal{L}\left(\tilde{\mathbf{\Sigma}}_i^{(t)}\right) \leq \mathcal{L}\left(\mathbf{\Sigma}_i^{(t)}\right) - \theta\left(\gamma - \frac{L}{2}\theta\right)\left\|\mathbf{D}_i^{(t)}\right\|_F^2 + \theta\left\|\mathbf{D}_i^{(t)}\right\|_F \left\|\nabla\mathcal{F}\left(\mathbf{\Sigma}_i^{(t)}\right) - \mathbf{Y}_i^{(t)}\right\|_F. \tag{29}$$

Invoking the convexity of $\mathcal{L}$ and the doubly stochasticity of $\mathbf{W}$, we can bound $\sum_{i=1}^m \mathcal{L}\left(\mathbf{\Sigma}_i^{(t)}\right)$ as

$$\sum_{i=1}^{m} \mathcal{L}\left(\mathbf{\Sigma}_i^{(t+1)}\right) = \sum_{i=1}^{m} \mathcal{L}\left(\sum_{j=1}^{m} W_{ij}\tilde{\mathbf{\Sigma}}_j^{(t)}\right) \leq \sum_{i=1}^{m}\sum_{j=1}^{m} W_{ij}\mathcal{L}\left(\tilde{\mathbf{\Sigma}}_j^{(t)}\right) = \sum_{i=1}^{m} \mathcal{L}\left(\tilde{\mathbf{\Sigma}}_i^{(t)}\right). \tag{30}$$

We can now substitute equation 29 into equation 30 and get

$$\sum_{i=1}^{m} \mathcal{L}\left(\mathbf{\Sigma}_i^{(t+1)}\right) \leq \sum_{i=1}^{m} \mathcal{L}\left(\mathbf{\Sigma}_i^{(t)}\right)$$
$$+ \sum_{i=1}^{m}\left(\theta\left\|\mathbf{D}_i^{(t)}\right\|_F \left\|\nabla\mathcal{F}\left(\mathbf{\Sigma}_i^{(t)}\right) - \mathbf{Y}_i^{(t)}\right\|_F - \theta\left(\gamma - \frac{L}{2}\theta\right)\left\|\mathbf{D}_i^{(t)}\right\|_F^2\right).$$

Using Young's inequality, we have

$$\theta\left\|\mathbf{D}_i^{(t)}\right\|_F\left\|\nabla\mathcal{F}\left(\mathbf{\Sigma}_i^{(t)}\right) - \mathbf{Y}_i^{(t)}\right\|_F \leq \frac{\theta}{2}\epsilon_p\left\|\mathbf{D}_i^{(t)}\right\|_F^2 + \frac{\theta}{2}\epsilon_p^{-1}\left\|\nabla\mathcal{F}\left(\mathbf{\Sigma}_i^{(t)}\right) - \mathbf{Y}_i^{(t)}\right\|_F^2, \quad (31)$$

where $\epsilon_p > 0$. Therefore, we have

$$\sum_{i=1}^{m}\mathcal{L}\left(\mathbf{\Sigma}_i^{(t+1)}\right)$$
$$\leq \sum_{i=1}^{m}\mathcal{L}\left(\mathbf{\Sigma}_i^{(t)}\right) + \sum_{i=1}^{m}\left(\frac{\theta}{2}\epsilon_p^{-1}\left\|\nabla\mathcal{F}\left(\mathbf{\Sigma}_i^{(t)}\right) - \mathbf{Y}_i^{(t)}\right\|_F^2 - \theta\left(\gamma - \frac{L}{2}\theta - \frac{1}{2}\epsilon_p\right)\left\|\mathbf{D}_i^{(t)}\right\|_F^2\right) \quad (32)$$
$$= \sum_{i=1}^{m}U\left(\mathbf{\Sigma}_i^{(t)}\right) + \frac{\theta}{2}\epsilon_p^{-1}\sum_{i=1}^{m}\left\|\nabla\mathcal{F}\left(\mathbf{\Sigma}_i^{(t)}\right) - \mathbf{Y}_i^{(t)}\right\|_F^2 - \theta\left(\gamma - \frac{L}{2}\theta - \frac{1}{2}\epsilon_p\right)\left\|\mathbf{D}_\diamond^{(t)}\right\|_F^2, \quad (33)$$

and we choose $\epsilon_p = \left(\gamma - \frac{L}{2}\theta\right)$. Then, we bound $\sum_{i=1}^{m}\left\|\nabla\mathcal{F}\left(\mathbf{\Sigma}_i^{(t)}\right) - \mathbf{Y}_i^{(t)}\right\|_F^2$ in terms of the consensus errors $\left\|\mathbf{E}_{\mathbf{\Sigma}}^{(t)}\right\|_F^2$ and $\left\|\mathbf{E}_{\mathbf{Y}}^{(t)}\right\|_F^2$. Recall equation 21, we have

$$\sum_{i=1}^{m}\left\|\nabla\mathcal{F}\left(\mathbf{\Sigma}_i^{(t)}\right) - \mathbf{Y}_i^{(t)}\right\|_F^2 = \sum_{i=1}^{m}\left\|\nabla\mathcal{F}\left(\mathbf{\Sigma}_i^{(t)}\right) - \bar{\mathbf{Y}}^{(t)} + \bar{\mathbf{Y}}^{(t)} - \mathbf{Y}_i^{(t)}\right\|_F^2$$
$$= \sum_{i=1}^{m}\left\|\frac{1}{m}\sum_{j=1}^{m}\nabla\mathcal{F}_j\left(\mathbf{\Sigma}_i^{(t)}\right) - \frac{1}{m}\sum_{j=1}^{m}\nabla\mathcal{F}_j\left(\mathbf{\Sigma}_j^{(t)}\right) + \bar{\mathbf{Y}}^{(t)} - \mathbf{Y}_i^{(t)}\right\|_F^2$$
$$\leq \frac{2}{m}\sum_{i=1}^{m}\sum_{j=1}^{m}\left\|\nabla\mathcal{F}_j\left(\mathbf{\Sigma}_i^{(t)}\right) - \nabla\mathcal{F}_j\left(\mathbf{\Sigma}_j^{(t)}\right)\right\|_F^2 + 2\left\|\mathbf{E}_{\mathbf{Y}}^{(t)}\right\|_F^2.$$

Recall the Lipschitz smoothness of $\mathcal{F}_i$, $i = 1, 2, \ldots, m$, we have

$$\sum_{i=1}^{m}\left\|\nabla\mathcal{F}\left(\mathbf{\Sigma}_i^{(t)}\right) - \mathbf{Y}_i^{(t)}\right\|_F^2 \leq \frac{2}{m}L^2\sum_{i=1}^{m}\sum_{j=1}^{m}\left\|\mathbf{\Sigma}_i^{(t)} - \mathbf{\Sigma}_j^{(t)}\right\|_F^2 + 2\left\|\mathbf{E}_{\mathbf{Y}}^{(t)}\right\|_F^2$$
$$= 4L^2\left\|\mathbf{E}_{\mathbf{\Sigma}}^{(t)}\right\|_F^2 + 2\left\|\mathbf{E}_{\mathbf{Y}}^{(t)}\right\|_F^2. \quad (34)$$

Substituting equation 34 into equation 33, we obtain the desired result equation 23. $\qquad\square$

Subsequently, we bound the consensus errors with the following lemma from (Sun et al., 2022b).

**Lemma 2.** *(Sun et al., 2022b) The disagreements $\left\|\mathbf{E}_{\mathbf{\Sigma}}^{(t)}\right\|_F$ and $\left\|\mathbf{E}_{\mathbf{Y}}^{(t)}\right\|_F$ are bounded by*

$$\left\|\mathbf{E}_{\mathbf{\Sigma}}^{(t+1)}\right\|_F \leq \rho\left\|\mathbf{E}_{\mathbf{\Sigma}}^{(t)}\right\|_F + \theta\rho\left\|\mathbf{D}_\diamond^{(t)}\right\|_F, \quad (35)$$

*and*

$$\left\|\mathbf{E}_{\mathbf{Y}}^{(t+1)}\right\|_F \leq \rho\left\|\mathbf{E}_{\mathbf{Y}}^{(t)}\right\|_F + 2L\rho\left\|\mathbf{E}_{\mathbf{\Sigma}}^{(t)}\right\|_F + \theta L\rho\left\|\mathbf{D}_\diamond^{(t)}\right\|_F. \quad (36)$$

Based on Proposition 9 and Lemma 2, we can prove that the potential function $\mathcal{V}$ equation 18 is non-increasing and hence $\mathbf{\Sigma}_\diamond^{(t)} \in \mathbb{A}$ for all $t$.

**Theorem 4.** *Assume that Assumptions 1 and 3 are satisfied. Based on Proposition 7, when*

$$\gamma \geq \max \left\{ 2\underline{r}^{-1} \left( \sqrt{m \left( ad + \frac{\sqrt{d}\tau}{\underline{r}} + \lambda d \right)^2 + e + \lambda d} \right), L \right\},$$

*and*

$$\theta \leq \min \left\{ \left( \sqrt{\frac{L^2}{16} + 32L\gamma \left( \frac{\rho^2(1+\rho^2)}{(1-\rho^2)^2} + 4\frac{\rho^4(1+\rho^2)^2}{(1-\rho^2)^4} \right)} + \frac{L}{4} \right)^{-1} \gamma, 1 \right\}, \qquad (37)$$

*for* $\left\{ \mathbf{\Sigma}_\diamond^{(t)}, \mathbf{Y}_\diamond^{(t)} \right\}_k$ *obtained by Algorithm 2, we have*

$$\mathcal{V} \left( \mathbf{\Sigma}_\diamond^{(t+1)}, \mathbf{Y}_\diamond^{(t+1)} \right) \leq \mathcal{V} \left( \mathbf{\Sigma}_\diamond^{(t)}, \mathbf{Y}_\diamond^{(t)} \right) - c_3 \sum_{i=1}^m \left\| \mathbf{D}_i^{(t)} \right\|_F^2, \qquad (38)$$

*where*

$$\mathcal{V} \left( \mathbf{\Theta}^{(t)}, \mathbf{E_Y}^{(t)} \right) = \sum_{i=1}^m \mathcal{L} \left( \mathbf{\Sigma}_i^{(t)} \right) + \frac{2\theta}{\left( \gamma - \frac{L}{2}\theta \right)(1-\rho^2)} \left\| \mathbf{E_Y}^{(t)} \right\|_F^2$$
$$+ \frac{4\theta L^2 (1+3\rho^2)^2}{\left( \gamma - \frac{L}{2}\theta \right)(1-\rho^2)^3} \left\| \mathbf{E_\Sigma}^{(t)} \right\|_F^2,$$

*and*

$$c_3 = \theta \left( \frac{\gamma}{2} - \frac{L}{4}\theta - \frac{8\theta^2 L^2 \rho^2 (1+\rho^2)}{\left( \gamma - \frac{L}{2}\theta \right)(1-\rho^2)^2} - \frac{16\theta^2 L^2 \rho^4 (1+\rho^2)^2}{\left( \gamma - \frac{L}{2}\theta \right)(1-\rho^2)^4} \right) \geq 0.$$

*Proof.* Squaring both sides of inequality equation 35 and utilizing Young's inequality provides

$$\left\| \mathbf{E_\Sigma}^{(t+1)} \right\|_F^2 \leq \rho^2 \left\| \mathbf{E_\Sigma}^{(t)} \right\|_F^2 + \theta^2 \rho^2 \left\| \mathbf{D}_\diamond^{(t)} \right\|_F^2 + 2\theta\rho^2 \left\| \mathbf{E_\Sigma}^{(t)} \right\|_F \left\| \mathbf{D}_\diamond^{(t)} \right\|_F$$
$$\leq \rho^2 (1+\epsilon_d) \left\| \mathbf{E_\Sigma}^{(t)} \right\|_F^2 + \theta^2 \rho^2 \left( 1+\epsilon_d^{-1} \right) \left\| \mathbf{D}_\diamond^{(t)} \right\|_F^2. \qquad (39)$$

Similarly, we have

$$\left\| \mathbf{E_Y}^{(t+1)} \right\|_F^2 \leq \rho^2 \left\| \mathbf{E_Y}^{(t)} \right\|_F^2 + \left( 2L\rho \left\| \mathbf{E_\Sigma}^{(t)} \right\|_F + \theta L\rho \left\| \mathbf{D}_\diamond^{(t)} \right\|_F \right)^2$$
$$+ 2\rho \left\| \mathbf{E_Y}^{(t)} \right\|_F \left( 2L\rho \left\| \mathbf{E_\Sigma}^{(t)} \right\|_F + \theta L\rho \left\| \mathbf{D}_\diamond^{(t)} \right\|_F \right)$$
$$\leq \rho^2 (1+\epsilon_d) \left\| \mathbf{E_Y}^{(t)} \right\|_F^2 + L^2 \rho^2 \left( 1+\epsilon_d^{-1} \right) \left( 2 \left\| \mathbf{E_\Sigma}^{(t)} \right\|_F + \theta \left\| \mathbf{D}_\diamond^{(t)} \right\|_F \right)^2$$
$$\leq \rho^2 (1+\epsilon_d) \left\| \mathbf{E_Y}^{(t)} \right\|_F^2 + 2L^2 \rho^2 \left( 1+\epsilon_d^{-1} \right) \left( 4 \left\| \mathbf{E_\Sigma}^{(t)} \right\|_F^2 + \theta^2 \left\| \mathbf{D}_\diamond^{(t)} \right\|_F^2 \right)$$
$$= \rho^2 (1+\epsilon_d) \left\| \mathbf{E_Y}^{(t)} \right\|_F^2 + 8L^2 \rho^2 \left( 1+\epsilon_d^{-1} \right) \left\| \mathbf{E_\Sigma}^{(t)} \right\|_F^2 + 2L^2 \rho^2 \left( 1+\epsilon_d^{-1} \right) \theta^2 \left\| \mathbf{D}_\diamond^{(t)} \right\|_F^2. \qquad (40)$$

Multiplying $\frac{2\theta L^2\left(4\rho^2\left(1+\epsilon_d^{-1}\right)+1-\rho^2(1+\epsilon_d)\right)}{\left(\gamma-\frac{L}{2}\theta\right)\left(1-\rho^2(1+\epsilon_d)\right)^2}$ and add $\frac{2\theta L^2\left(4\rho^2\left(1+\epsilon_d^{-1}\right)+1-\rho^2(1+\epsilon_d)\right)}{\left(\gamma-\frac{L}{2}\theta\right)\left(1-\rho^2(1+\epsilon_d)\right)}\left\|\mathbf{E}_{\mathbf{\Sigma}}^{(t)}\right\|_F^2$ on both sides of equation 39 leads to

$$
\begin{aligned}
&\frac{2\theta L^2\left(4\rho^2\left(1+\epsilon_d^{-1}\right)+1-\rho^2\left(1+\epsilon_d\right)\right)}{\left(\gamma-\frac{L}{2}\theta\right)\left(1-\rho^2\left(1+\epsilon_d\right)\right)^2}\left\|\mathbf{E}_{\mathbf{\Sigma}}^{(t+1)}\right\|_F^2 \\
&+\frac{2\theta L^2\left(4\rho^2\left(1+\epsilon_d^{-1}\right)+1-\rho^2\left(1+\epsilon_d\right)\right)}{\left(\gamma-\frac{L}{2}\theta\right)\left(1-\rho^2\left(1+\epsilon_d\right)\right)}\left\|\mathbf{E}_{\mathbf{\Sigma}}^{(t)}\right\|_F^2 \\
&\leq\frac{2\theta L^2\left(4\rho^2\left(1+\epsilon_d^{-1}\right)+1-\rho^2\left(1+\epsilon_d\right)\right)}{\left(\gamma-\frac{L}{2}\theta\right)\left(1-\rho^2\left(1+\epsilon_d\right)\right)^2}\left\|\mathbf{E}_{\mathbf{\Sigma}}^{(t)}\right\|_F^2 \\
&+\frac{2\theta^3 L^2\left(4\rho^2\left(1+\epsilon_d^{-1}\right)+1-\rho^2\left(1+\epsilon_d\right)\right)\rho^2\left(1+\epsilon_d^{-1}\right)}{\left(\gamma-\frac{L}{2}\theta\right)\left(1-\rho^2\left(1+\epsilon_d\right)\right)^2}\left\|\mathbf{D}_{\diamond}^{(t)}\right\|_F^2,
\end{aligned}
\tag{41}
$$

while multiplying $\frac{\theta\left(\gamma-\frac{L}{2}\theta\right)^{-1}}{1-\rho^2(1+\epsilon_d)}$ and add $\theta\left(\gamma-\frac{L}{2}\theta\right)^{-1}\left\|\mathbf{E}_{\mathbf{Y}}^{(t)}\right\|_F^2$ on both sides of equation 40 leads to

$$
\begin{aligned}
&\frac{\theta\left(\gamma-\frac{L}{2}\theta\right)^{-1}}{1-\rho^2\left(1+\epsilon_d\right)}\left\|\mathbf{E}_{\mathbf{Y}}^{(t+1)}\right\|_F^2+\theta\left(\gamma-\frac{L}{2}\theta\right)^{-1}\left\|\mathbf{E}_{\mathbf{Y}}^{(t)}\right\|_F^2 \\
&\leq\frac{\theta\left(\gamma-\frac{L}{2}\theta\right)^{-1}}{1-\rho^2\left(1+\epsilon_d\right)}\left\|\mathbf{E}_{\mathbf{Y}}^{(t)}\right\|_F^2+\frac{8\theta\left(\gamma-\frac{L}{2}\theta\right)^{-1}L^2\rho^2\left(1+\epsilon_d^{-1}\right)}{1-\rho^2\left(1+\epsilon_d\right)}\left\|\mathbf{E}_{\mathbf{\Sigma}}^{(t)}\right\|_F^2 \\
&+\frac{2\theta\left(\gamma-\frac{L}{2}\theta\right)^{-1}L^2\rho^2\left(1+\epsilon_d^{-1}\right)\theta^2}{1-\rho^2\left(1+\epsilon_d\right)}\left\|\mathbf{D}_{\diamond}^{(t)}\right\|_F^2.
\end{aligned}
\tag{42}
$$

By summing equation 23, equation 41, and equation 42, using equation 37, and choosing $\epsilon_d=\frac{1-\rho^2}{2\rho^2}$, we obtain the desired result equation 38. $\qquad\square$

Theorem 4 shows that, inducting from $\mathbf{\Sigma}_{\diamond}^{(0)}$, $\mathbf{\Sigma}_{\diamond}^{(t)}\in\mathbb{A}$ for all $t$, and hence the local variable of each agent is positive definite.

### C.2.2 LINEAR CONVERGENCE

Based on the local properties, we can prove the linear convergence of Algorithm 2. According to Proposition 8 and Theorem 4, we have

$$
\mathcal{F}\left(\widehat{\mathbf{\Sigma}}\right)-\mathcal{F}\left(\mathbf{\Sigma}_i^{(t)}\right)-\left\langle\nabla\mathcal{F}\left(\mathbf{\Sigma}_i^{(t)}\right),\widehat{\mathbf{\Sigma}}-\mathbf{\Sigma}_i^{(t)}\right\rangle\geq\frac{\mu}{2}\left\|\widehat{\mathbf{\Sigma}}-\mathbf{\Sigma}_i^{(t)}\right\|_F^2,
$$

and

$$
\mathcal{F}\left(\mathbf{\Sigma}_i^{\left(t+\frac{1}{2}\right)}\right)-\mathcal{F}\left(\mathbf{\Sigma}_i^{(t)}\right)-\left\langle\nabla\mathcal{F}\left(\mathbf{\Sigma}_i^{(t)}\right),\mathbf{D}_i^{(t)}\right\rangle\leq\frac{L}{2}\left\|\mathbf{D}_i^{(t)}\right\|_F^2.
$$

Adding the two inequalities above leads to

$$
\begin{aligned}
0\leq&\mathcal{F}\left(\widehat{\mathbf{\Sigma}}\right)-\mathcal{F}\left(\mathbf{\Sigma}_i^{\left(t+\frac{1}{2}\right)}\right)+\frac{L}{2}\left\|\mathbf{D}_i^{(t)}\right\|_F^2-\frac{\mu}{2}\left\|\widehat{\mathbf{\Sigma}}-\mathbf{\Sigma}_i^{(t)}\right\|_F^2 \\
&-\left\langle\nabla\mathcal{F}\left(\mathbf{\Sigma}_i^{(t)}\right),\widehat{\mathbf{\Sigma}}-\mathbf{\Sigma}_i^{\left(t+\frac{1}{2}\right)}\right\rangle.
\end{aligned}
\tag{43}
$$

Due to the first-order optimality condition of equation 5, $\exists\mathbf{\Phi}\in\partial\lambda\left\|\mathbf{\Sigma}_i^{\left(t+\frac{1}{2}\right)}\right\|_{1,\text{off}}$ such that

$$
\left\langle\mathbf{Y}_i^{(t)}+\gamma\left(\mathbf{D}_i^{(t)}\right)+\mathbf{\Phi},\widehat{\mathbf{\Sigma}}-\mathbf{\Sigma}_i^{\left(t+\frac{1}{2}\right)}\right\rangle\geq0.
$$

Since $\lambda\left\|\cdot\right\|_{1,\text{off}}$ is convex, we have

$$
\lambda\left\|\widehat{\mathbf{\Sigma}}\right\|_{1,\text{off}}-\lambda\left\|\mathbf{\Sigma}_i^{\left(t+\frac{1}{2}\right)}\right\|_{1,\text{off}}-\left\langle\mathbf{\Phi},\widehat{\mathbf{\Sigma}}-\mathbf{\Sigma}_i^{\left(t+\frac{1}{2}\right)}\right\rangle\geq0.
$$

Adding the above two inequalities leads to

$$\lambda \left\| \widehat{\boldsymbol{\Sigma}} \right\|_{1,\text{off}} - \lambda \left\| \boldsymbol{\Sigma}_i^{\left(t+\frac{1}{2}\right)} \right\|_{1,\text{off}} + \left\langle \mathbf{Y}_i^{(t)} + \gamma \left( \mathbf{D}_i^{(t)} \right), \widehat{\boldsymbol{\Sigma}} - \boldsymbol{\Sigma}_i^{\left(t+\frac{1}{2}\right)} \right\rangle \geq 0. \qquad (44)$$

Combining equation 43 and equation 44 gives

$$0 \leq \mathcal{L}\left( \widehat{\boldsymbol{\Sigma}} \right) - \mathcal{L}\left( \boldsymbol{\Sigma}_i^{\left(t+\frac{1}{2}\right)} \right) + \frac{L}{2} \left\| \mathbf{D}_i^{(t)} \right\|_F^2 - \frac{\mu}{2} \left\| \widehat{\boldsymbol{\Sigma}} - \boldsymbol{\Sigma}_i^{(t)} \right\|_F^2$$

$$+ \gamma \left\langle \mathbf{D}_i^{(t)}, \widehat{\boldsymbol{\Sigma}} - \boldsymbol{\Sigma}_i^{\left(t+\frac{1}{2}\right)} \right\rangle + \left\langle \mathbf{Y}_i^{(t)} - \nabla \mathcal{F}\left( \boldsymbol{\Sigma}_i^{(t)} \right), \widehat{\boldsymbol{\Sigma}} - \boldsymbol{\Sigma}_i^{\left(t+\frac{1}{2}\right)} \right\rangle.$$

Since $\widehat{\boldsymbol{\Sigma}}$ is the minimum of $\mathcal{L}$, we have $\mathcal{L}\left( \widehat{\boldsymbol{\Sigma}} \right) - \mathcal{L}\left( \boldsymbol{\Sigma}_i^{\left(t+\frac{1}{2}\right)} \right) \leq 0$, and hence

$$2 \left\langle \mathbf{D}_i^{(t)}, \boldsymbol{\Sigma}_i^{(t)} - \widehat{\boldsymbol{\Sigma}} \right\rangle \leq \frac{L - 2\gamma}{\gamma} \left\| \mathbf{D}_i^{(t)} \right\|_F^2 - \frac{\mu}{\gamma} \left\| \widehat{\boldsymbol{\Sigma}} - \boldsymbol{\Sigma}_i^{(t)} \right\|_F^2$$

$$+ \frac{2}{\gamma} \left\langle \mathbf{Y}_i^{(t)} - \nabla \mathcal{F}\left( \boldsymbol{\Sigma}_i^{(t)} \right), \widehat{\boldsymbol{\Sigma}} - \boldsymbol{\Sigma}_i^{\left(t+\frac{1}{2}\right)} \right\rangle,$$

and hence

$$\left\| \boldsymbol{\Sigma}_i^{\left(t+\frac{1}{2}\right)} - \widehat{\boldsymbol{\Sigma}} \right\|_F^2 \leq \left\| \mathbf{D}_i^{(t)} \right\|_F^2 + \left\| \boldsymbol{\Sigma}_i^{(t)} - \widehat{\boldsymbol{\Sigma}} \right\|_F^2 + 2 \left\langle \mathbf{D}_i^{(t)}, \boldsymbol{\Sigma}_i^{(t)} - \widehat{\boldsymbol{\Sigma}} \right\rangle$$

$$\leq - \left( 1 - \frac{L}{\gamma} \right) \left\| \mathbf{D}_i^{(t)} \right\|_F^2 + \left( 1 - \frac{\mu}{\gamma} \right) \left\| \boldsymbol{\Sigma}_i^{(t)} - \widehat{\boldsymbol{\Sigma}} \right\|_F^2$$

$$+ \frac{2}{\gamma} \left\langle \mathbf{Y}_i^{(t)} - \nabla \mathcal{F}\left( \boldsymbol{\Sigma}_i^{(t)} \right), \widehat{\boldsymbol{\Sigma}} - \boldsymbol{\Sigma}_i^{\left(t+\frac{1}{2}\right)} \right\rangle. \qquad (45)$$

Then we bound $\left\langle \mathbf{Y}_i^{(t)} - \nabla \mathcal{F}\left( \boldsymbol{\Sigma}_i^{(t)} \right), \widehat{\boldsymbol{\Sigma}} - \boldsymbol{\Sigma}_i^{\left(t+\frac{1}{2}\right)} \right\rangle$ by

$$\left\langle \mathbf{Y}_i^{(t)} - \nabla \mathcal{F}\left( \boldsymbol{\Sigma}_i^{(t)} \right), \widehat{\boldsymbol{\Sigma}} - \boldsymbol{\Sigma}_i^{\left(t+\frac{1}{2}\right)} \right\rangle$$

$$\leq \left\langle \mathbf{Y}_i^{(t)} - \bar{\mathbf{Y}}^{(t)}, \widehat{\boldsymbol{\Sigma}} - \boldsymbol{\Sigma}_i^{\left(t+\frac{1}{2}\right)} \right\rangle$$

$$+ \left\langle \frac{1}{m} \sum_{j=1}^m \nabla \mathcal{F}_j \left( \boldsymbol{\Sigma}_j^{(t)} \right) - \frac{1}{m} \sum_{j=1}^m \nabla \mathcal{F}_j \left( \bar{\mathbf{S}}^{(t)} \right), \widehat{\boldsymbol{\Sigma}} - \boldsymbol{\Sigma}_i^{\left(t+\frac{1}{2}\right)} \right\rangle$$

$$+ \left\langle \frac{1}{m} \sum_{j=1}^m \nabla \mathcal{F}_j \left( \bar{\mathbf{S}}^{(t)} \right) - \frac{1}{m} \sum_{j=1}^m \nabla \mathcal{F}_j \left( \boldsymbol{\Sigma}_i^{(t)} \right), \widehat{\boldsymbol{\Sigma}} - \boldsymbol{\Sigma}_i^{\left(t+\frac{1}{2}\right)} \right\rangle. \qquad (46)$$

According to the update rule equation 7, we have

$$\mathbf{Y}_i^{(t)} - \bar{\mathbf{Y}}^{(t)} = \sum_{j=1}^m \ell_{ij}^{(t)} \mathbf{E}_{\mathbf{Y}_j}^{(0)} + \sum_{s=0}^{t-1} \sum_{j=1}^m \ell_{ij}^{(t-s)} \left( \nabla \mathcal{F}_j \left( \boldsymbol{\Sigma}_j^{(s+1)} \right) - \nabla \mathcal{F}_j \left( \boldsymbol{\Sigma}_j^{(s)} \right) \right),$$

where $\ell_{ij}^{(t)}$ denotes the $(i,j)$-th element of matrix $(\mathbf{W} - \mathbf{J})^t$. Therefore, we have

$$\sum_{i=1}^m \left\langle \mathbf{Y}_i^{(t)} - \bar{\mathbf{Y}}^{(t)}, \widehat{\boldsymbol{\Sigma}} - \boldsymbol{\Sigma}_i^{\left(t+\frac{1}{2}\right)} \right\rangle$$

$$= \sum_{i=1}^m \left\langle \sum_{j=1}^m \ell_{ij}^{(t)} \mathbf{E}_{\mathbf{Y}_j}^{(0)} + \sum_{s=0}^{t-1} \sum_{j=1}^m \ell_{ij}^{(t-s)} \left( \nabla \mathcal{F}_j \left( \boldsymbol{\Sigma}_j^{(s+1)} \right) - \nabla \mathcal{F}_j \left( \boldsymbol{\Sigma}_j^{(s)} \right) \right), \widehat{\boldsymbol{\Sigma}} - \boldsymbol{\Sigma}_i^{\left(t+\frac{1}{2}\right)} \right\rangle$$

$$\leq \sum_{i=1}^m \sum_{j=1}^m \ell_{ij}^{(t)} \left\| \mathbf{E}_{\mathbf{Y}_j}^{(0)} \right\|_F \left\| \widehat{\boldsymbol{\Sigma}} - \boldsymbol{\Sigma}_i^{\left(t+\frac{1}{2}\right)} \right\|_F$$

$$+ \sum_{i=1}^m \sum_{s=0}^{t-1} \sum_{j=1}^m \ell_{ij}^{(t-s)} \left\| \nabla \mathcal{F}_j \left( \boldsymbol{\Sigma}_j^{(s+1)} \right) - \nabla \mathcal{F}_j \left( \boldsymbol{\Sigma}_j^{(s)} \right) \right\|_F \left\| \widehat{\boldsymbol{\Sigma}} - \boldsymbol{\Sigma}_i^{\left(t+\frac{1}{2}\right)} \right\|_F.$$

Moreover, under Assumption 1, we have $\left\|(\mathbf{W}-\mathbf{J})^t\right\|_2 = \rho^t$, and hence $\left\|(\mathbf{W}-\mathbf{J})^t\right\|_\infty \leq \sqrt{m}\left\|(\mathbf{W}-\mathbf{J})^t\right\|_2 = \sqrt{m}\rho^t$. Then based on Young's inequality, Proposition 8, and Theorem 4, we have

$$\sum_{i=1}^m \left\langle \mathbf{Y}_i^{(t)} - \bar{\mathbf{Y}}^{(t)}, \widehat{\boldsymbol{\Sigma}} - \boldsymbol{\Sigma}_i^{(t+\frac{1}{2})} \right\rangle$$

$$\leq \frac{\sqrt{m}\rho^t}{2\epsilon}\left\|\mathbf{E}_{\mathbf{Y}}^{(0)}\right\|_F^2 + \frac{\epsilon\sqrt{m}\rho^t}{2}\sum_{i=1}^m\left\|\widehat{\boldsymbol{\Sigma}} - \boldsymbol{\Sigma}_i^{(t+\frac{1}{2})}\right\|_F^2 + \frac{L\sqrt{m}}{2\epsilon}\sum_{s=0}^{t-1}\rho^{t-s}\left\|\boldsymbol{\Sigma}_\diamond^{(s+1)} - \boldsymbol{\Sigma}_\diamond^{(s)}\right\|_F^2$$

$$+ \frac{L\epsilon\sqrt{m}}{2}\sum_{s=0}^{t-1}\rho^{t-s}\sum_{i=1}^m\left\|\widehat{\boldsymbol{\Sigma}} - \boldsymbol{\Sigma}_i^{(t+\frac{1}{2})}\right\|_F^2$$

$$\leq \frac{\sqrt{m}\rho^t}{2\epsilon}\left\|\mathbf{E}_{\mathbf{Y}}^{(0)}\right\|_F^2 + \frac{L\epsilon\sqrt{m}\rho}{1-\rho}\sum_{i=1}^m\left\|\widehat{\boldsymbol{\Sigma}} - \boldsymbol{\Sigma}_i^{(t+\frac{1}{2})}\right\|_F^2 + \frac{L\sqrt{m}}{2\epsilon}\sum_{s=0}^{t-1}\rho^{t-s}\left\|\boldsymbol{\Sigma}_\diamond^{(s+1)} - \boldsymbol{\Sigma}_\diamond^{(s)}\right\|_F^2,$$

where $\epsilon > 0$. Since

$$\left\|\boldsymbol{\Sigma}_\diamond^{(s+1)} - \boldsymbol{\Sigma}_\diamond^{(s)}\right\|_F^2 = \left\|(\mathbf{W}\otimes\mathbf{I})\left(\boldsymbol{\Sigma}_\diamond^{(s)} + \theta\mathbf{D}_\diamond^{(s)}\right) - \boldsymbol{\Sigma}_\diamond^{(s)}\right\|_F^2 \leq 8\left\|\mathbf{E}_{\boldsymbol{\Sigma}}^{(s)}\right\|_F^2 + 2\left\|\mathbf{D}_\diamond^{(s)}\right\|_F^2,$$

we have

$$\sum_{i=1}^m \left\langle \mathbf{Y}_i^{(t)} - \bar{\mathbf{Y}}^{(t)}, \widehat{\boldsymbol{\Sigma}} - \boldsymbol{\Sigma}_i^{(t+\frac{1}{2})} \right\rangle$$

$$\leq \frac{\sqrt{m}\rho^t}{2\epsilon}\left\|\mathbf{E}_{\mathbf{Y}}^{(0)}\right\|_F^2 + \frac{L\epsilon\sqrt{m}\rho}{1-\rho}\sum_{i=1}^m\left\|\widehat{\boldsymbol{\Sigma}} - \boldsymbol{\Sigma}_i^{(t+\frac{1}{2})}\right\|_F^2 + \frac{L\sqrt{m}}{2\epsilon}\sum_{s=0}^{t-1}\rho^{t-s}\left(8\left\|\mathbf{E}_{\boldsymbol{\Sigma}}^{(s)}\right\|_F^2 + 2\left\|\mathbf{D}_\diamond^{(s)}\right\|_F^2\right).$$

$$\tag{47}$$

Similarly, according to the update rule equation 6, we have

$$\boldsymbol{\Sigma}_i^{(t)} - \bar{\boldsymbol{\Sigma}}^{(t)} = \sum_{j=1}^m \ell_{ij}^{(t)}\mathbf{E}_{\boldsymbol{\Sigma}_j}^{(0)} + \theta\sum_{j=1}^m\sum_{s=0}^{t-1}\ell_{ij}^{(t-s)}\left(\boldsymbol{\Sigma}_j^{(s+\frac{1}{2})} - \boldsymbol{\Sigma}_j^{(s)}\right), \tag{48}$$

and thus

$$\sum_{i=1}^m \left\langle \frac{1}{m}\sum_{j=1}^m\nabla\mathcal{F}_j\left(\boldsymbol{\Sigma}_j^{(t)}\right) - \frac{1}{m}\sum_{j=1}^m\nabla\mathcal{F}_j\left(\bar{\boldsymbol{\Sigma}}^{(t)}\right), \widehat{\boldsymbol{\Sigma}} - \boldsymbol{\Sigma}_i^{(t+\frac{1}{2})}\right\rangle$$

$$\leq \sum_{i=1}^m\frac{L}{m}\sum_{j=1}^m\left\|\bar{\boldsymbol{\Sigma}}^{(t)} - \boldsymbol{\Sigma}_j^{(t)}\right\|_F\left\|\widehat{\boldsymbol{\Sigma}} - \boldsymbol{\Sigma}_i^{(t+\frac{1}{2})}\right\|_F$$

$$\leq \sum_{i=1}^m\frac{L}{m}\sum_{j=1}^m\left\|\sum_{l=1}^m\ell_{jl}^{(t)}\mathbf{E}_{\boldsymbol{\Sigma}_l}^{(0)} + \theta\sum_{l=1}^m\sum_{s=0}^{t-1}\ell_{jl}^{(t-s)}\left(\boldsymbol{\Sigma}_l^{(s+\frac{1}{2})} - \boldsymbol{\Sigma}_l^{(s)}\right)\right\|_F\left\|\widehat{\boldsymbol{\Sigma}} - \boldsymbol{\Sigma}_i^{(t+\frac{1}{2})}\right\|_F$$

$$\leq \frac{L}{m}\sum_{i=1}^m\sum_{j=1}^m\sum_{l=1}^m\ell_{jl}^{(t)}\left\|\mathbf{E}_{\boldsymbol{\Sigma}_l}^{(0)}\right\|_F\left\|\widehat{\boldsymbol{\Sigma}} - \boldsymbol{\Sigma}_i^{(t+\frac{1}{2})}\right\|_F$$

$$+ \frac{L\theta}{m}\sum_{i=1}^m\sum_{j=1}^m\sum_{l=1}^m\sum_{s=0}^{t-1}\ell_{jl}^{(t-s)}\left\|\boldsymbol{\Sigma}_l^{(s+\frac{1}{2})} - \boldsymbol{\Sigma}_l^{(s)}\right\|_F\left\|\widehat{\boldsymbol{\Sigma}} - \boldsymbol{\Sigma}_i^{(t+\frac{1}{2})}\right\|_F$$

$$\leq \frac{L\sqrt{m}\rho^t}{2\epsilon}\left\|\mathbf{E}_{\boldsymbol{\Sigma}}^{(0)}\right\|_F^2 + \frac{L\epsilon\sqrt{m}\rho^t}{2}\sum_{i=1}^m\left\|\widehat{\boldsymbol{\Sigma}} - \boldsymbol{\Sigma}_i^{(t+\frac{1}{2})}\right\|_F^2 + \frac{L\theta\sqrt{m}}{2\epsilon}\sum_{s=0}^{t-1}\rho^{t-s}\left\|\mathbf{D}_\diamond^{(s)}\right\|_F^2$$

$$+ \frac{L\theta\epsilon\sqrt{m}}{2}\sum_{s=0}^{t-1}\rho^{t-s}\sum_{i=1}^m\left\|\widehat{\boldsymbol{\Sigma}} - \boldsymbol{\Sigma}_i^{(t+\frac{1}{2})}\right\|_F^2$$

$$\leq \frac{L\sqrt{m}\rho^t}{2\epsilon}\left\|\mathbf{E}_{\boldsymbol{\Sigma}}^{(0)}\right\|_F^2 + \frac{L\theta\epsilon\sqrt{m}\rho}{1-\rho}\sum_{i=1}^m\left\|\widehat{\boldsymbol{\Sigma}} - \boldsymbol{\Sigma}_i^{(t+\frac{1}{2})}\right\|_F^2 + \frac{L\theta\sqrt{m}}{2\epsilon}\sum_{s=0}^{t-1}\rho^{t-s}\left\|\mathbf{D}_\diamond^{(s)}\right\|_F^2, \tag{49}$$

and

$$\sum_{i=1}^{m} \left\langle \frac{1}{m} \sum_{j=1}^{m} \nabla \mathcal{F}_j \left( \bar{\mathbf{\Sigma}}^{(t)} \right) - \frac{1}{m} \sum_{j=1}^{m} \nabla \mathcal{F}_j \left( \mathbf{\Sigma}_i^{(t)} \right), \widehat{\mathbf{\Sigma}} - \mathbf{\Sigma}_i^{\left(t+\frac{1}{2}\right)} \right\rangle$$

$$\leq \sum_{i=1}^{m} L \left\| \bar{\mathbf{\Sigma}}^{(t)} - \mathbf{\Sigma}_i^{(t)} \right\|_F \left\| \widehat{\mathbf{\Sigma}} - \mathbf{\Sigma}_i^{\left(t+\frac{1}{2}\right)} \right\|_F$$

$$\leq \sum_{i=1}^{m} L \left\| \sum_{j=1}^{m} \ell_{ij}^{(t)} \mathbf{E}_{\mathbf{\Sigma}_j}^{(0)} + \theta \sum_{j=1}^{m} \sum_{s=0}^{t-1} \ell_{ij}^{(t-s)} \left( \mathbf{\Sigma}_j^{\left(s+\frac{1}{2}\right)} - \mathbf{\Sigma}_j^{(s)} \right) \right\|_F \left\| \widehat{\mathbf{\Sigma}} - \mathbf{\Sigma}_i^{\left(t+\frac{1}{2}\right)} \right\|_F$$

$$\leq L \sum_{i=1}^{m} \sum_{j=1}^{m} \ell_{ij}^{(t)} \left\| \mathbf{E}_{\mathbf{\Sigma}_j}^{(0)} \right\|_F \left\| \widehat{\mathbf{\Sigma}} - \mathbf{\Sigma}_i^{\left(t+\frac{1}{2}\right)} \right\|_F$$

$$+ L\theta \sum_{i=1}^{m} \sum_{j=1}^{m} \sum_{s=0}^{t-1} \ell_{ij}^{(t-s)} \left\| \mathbf{\Sigma}_j^{\left(s+\frac{1}{2}\right)} - \mathbf{\Sigma}_j^{(s)} \right\|_F \left\| \widehat{\mathbf{\Sigma}} - \mathbf{\Sigma}_i^{\left(t+\frac{1}{2}\right)} \right\|_F \tag{50}$$

$$\leq \frac{L\sqrt{m}\rho^t}{2\epsilon} \left\| \mathbf{E}_{\mathbf{\Sigma}}^{(0)} \right\|_F^2 + \frac{L\epsilon\sqrt{m}\rho^t}{2} \sum_{i=1}^{m} \left\| \widehat{\mathbf{\Sigma}} - \mathbf{\Sigma}_i^{\left(t+\frac{1}{2}\right)} \right\|_F^2 + \frac{L\theta\sqrt{m}}{2\epsilon} \sum_{s=0}^{t-1} \rho^{t-s} \left\| \mathbf{D}_{\diamond}^{(s)} \right\|_F^2$$

$$+ \frac{L\theta\epsilon\sqrt{m}}{2} \sum_{s=0}^{t-1} \rho^{t-s} \sum_{i=1}^{m} \left\| \widehat{\mathbf{\Sigma}} - \mathbf{\Sigma}_i^{\left(t+\frac{1}{2}\right)} \right\|_F^2$$

$$\leq \frac{L\sqrt{m}\rho^t}{2\epsilon} \left\| \mathbf{E}_{\mathbf{\Sigma}}^{(0)} \right\|_F^2 + \frac{L\theta\epsilon\sqrt{m}\rho}{1-\rho} \sum_{i=1}^{m} \left\| \widehat{\mathbf{\Sigma}} - \mathbf{\Sigma}_i^{\left(t+\frac{1}{2}\right)} \right\|_F^2 + \frac{L\theta\sqrt{m}}{2\epsilon} \sum_{s=0}^{t-1} \rho^{t-s} \left\| \mathbf{D}_{\diamond}^{(s)} \right\|_F^2. \tag{51}$$

Plugging equation 47, equation 49, and equation 51 into equation 46 leads to

$$\sum_{i=1}^{m} \left\langle \mathbf{Y}_i^{(t)} - \nabla \mathcal{F} \left( \mathbf{\Sigma}_i^{(t)} \right), \widehat{\mathbf{\Sigma}} - \mathbf{\Sigma}_i^{\left(t+\frac{1}{2}\right)} \right\rangle$$

$$\leq \frac{\sqrt{m}\rho^t}{2\epsilon} \left\| \mathbf{E}_{\mathbf{Y}}^{(0)} \right\|_F^2 + \frac{4L\sqrt{m}}{\epsilon} \sum_{s=0}^{t-1} \rho^{t-s} \left\| \mathbf{E}_{\mathbf{\Sigma}}^{(s)} \right\|_F^2 \tag{52}$$

$$+ \frac{L\sqrt{m}\rho^t}{\epsilon} \left\| \mathbf{E}_{\mathbf{\Sigma}}^{(0)} \right\|_F^2 + \frac{3L\epsilon\sqrt{m}\rho}{1-\rho} \sum_{i=1}^{m} \left\| \widehat{\mathbf{\Sigma}} - \mathbf{\Sigma}_i^{\left(t+\frac{1}{2}\right)} \right\|_F^2 + \frac{2L\sqrt{m}}{\epsilon} \sum_{s=0}^{t-1} \rho^{t-s} \left\| \mathbf{D}_{\diamond}^{(s)} \right\|_F^2.$$

Plugging equation 52 into equation 45, we have

$$\left( 1 - \frac{6L\epsilon\sqrt{m}\rho}{\gamma(1-\rho)} \right) \sum_{i=1}^{m} \left\| \mathbf{\Sigma}_i^{\left(t+\frac{1}{2}\right)} - \widehat{\mathbf{\Sigma}} \right\|_F^2$$

$$\leq \left( 1 - \frac{\mu}{\gamma} \right) \sum_{i=1}^{m} \left\| \mathbf{\Sigma}_i^{(t)} - \widehat{\mathbf{\Sigma}} \right\|_F^2 - \left( 1 - \frac{L}{\gamma} \right) \left\| \mathbf{D}_{\diamond}^{(t)} \right\|_F^2$$

$$+ \frac{4L\sqrt{m}}{\gamma\epsilon} \sum_{s=0}^{t-1} \rho^{t-s} \left\| \mathbf{D}_{\diamond}^{(s)} \right\|_F^2 + \frac{8L\sqrt{m}}{\gamma\epsilon} \sum_{s=0}^{t-1} \rho^{t-s} \left\| \mathbf{E}_{\mathbf{\Sigma}}^{(s)} \right\|_F^2$$

$$+ \frac{\sqrt{m}\rho^t}{\gamma\epsilon} \left\| \mathbf{E}_{\mathbf{Y}}^{(0)} \right\|_F^2 + \frac{2L\sqrt{m}\rho^t}{\gamma\epsilon} \left\| \mathbf{E}_{\mathbf{\Sigma}}^{(0)} \right\|_F^2. \tag{53}$$

Due to the update rule equation 6, we have

$$\sum_{i=1}^{m} \left\| \tilde{\mathbf{\Sigma}}_i^{(t)} - \widehat{\mathbf{\Sigma}} \right\|_F^2 = \sum_{i=1}^{m} \left\| \mathbf{\Sigma}_i^{(t)} + \theta \left( \mathbf{D}_i^{(t)} \right) - \widehat{\mathbf{\Sigma}} \right\|_F^2$$

$$\leq \theta \sum_{i=1}^{m} \left\| \mathbf{\Sigma}_i^{\left(t+\frac{1}{2}\right)} - \widehat{\mathbf{\Sigma}} \right\|^2 + (1-\theta) \sum_{i=1}^{m} \left\| \mathbf{\Sigma}_i^{(t)} - \widehat{\mathbf{\Sigma}} \right\|^2. \tag{54}$$

Plugging equation 54 into equation 53 and the fact that $\left\|\mathbf{\Sigma}_i^{(t)} - \widehat{\mathbf{\Sigma}}\right\|_F^2 \leq \left\|\tilde{\mathbf{\Sigma}}_i^{(t-1)} - \widehat{\mathbf{\Sigma}}\right\|_F^2$, we have

$$
\frac{1}{\theta}\left(1 - \frac{6L\epsilon\sqrt{m}\rho}{\gamma\left(1-\rho\right)}\right)\sum_{i=1}^m \left\|\tilde{\mathbf{\Sigma}}_i^{(t)} - \widehat{\mathbf{\Sigma}}\right\|_F^2
$$

$$
\leq \left(1 - \frac{\mu}{\gamma} + \frac{1-\theta}{\theta}\left(1 - \frac{6L\epsilon\sqrt{m}\rho}{\gamma\left(1-\rho\right)}\right)\right)\sum_{i=1}^m \left\|\mathbf{\Sigma}_i^{(t-1)} - \widehat{\mathbf{\Sigma}}\right\|_F^2 - \left(1 - \frac{L}{\gamma}\right)\left\|\mathbf{D}_\diamond^{(t)}\right\|_F^2
$$

$$
+ \frac{4L\sqrt{m}}{\gamma\epsilon}\sum_{s=0}^{t-1}\rho^{t-s}\left\|\mathbf{D}_\diamond^{(s)}\right\|_F^2 + \frac{8L\sqrt{m}}{\gamma\epsilon}\sum_{s=0}^{t-1}\rho^{t-s}\left\|\mathbf{E}_{\mathbf{\Sigma}}^{(s)}\right\|_F^2
$$

$$
+ \frac{\sqrt{m}\rho^t}{\gamma\epsilon}\left\|\mathbf{E}_{\mathbf{Y}}^{(0)}\right\|_F^2 + \frac{2L\sqrt{m}\rho^t}{\gamma\epsilon}\left\|\mathbf{E}_{\mathbf{\Sigma}}^{(0)}\right\|_F^2. \tag{55}
$$

We then prove the linear convergence based on the following lemmas.

**Lemma 3** (Sun et al. (2022a)). *For $T \geq 1$, $\rho \in (0,1)$, $z \in (\rho, 1)$, and a nonnegative sequence $\{a(t)\}$, define*

$$
A^{(T)}(z) = \sum_{t=1}^T a(t)z^{-t}.
$$

*Then we have:*

- $\sum_{t=1}^T a(t+1)z^{-t} \geq zA^{(T)}(z) - a(1)$;
- $\sum_{t=1}^T \left(\sum_{\eta=0}^{t-1}\rho^{t-\eta}a(\eta)\right)z^{-t} \leq \frac{\rho}{z-\rho}\left(A^{(T)}(z) + a(0)\right)$;
- $\sum_{t=1}^T \left(\sum_{\eta=0}^{t-1}\rho^{t-\eta}a(\eta+1)\right)z^{-t} \leq \frac{z\rho}{z-\rho}A^{(T)}(z)$.

**Lemma 4** (Sun et al. (2022a)). *If for all $T \geq 1$ and $z \in (0,1)$, a nonnegative sequence $\{a(t)\}$ satisfies*

$$
\sum_{t=1}^T a(t)z^{-t} \leq B + c\sum_{t=1}^T z^{-t},
$$

*where $B, c > 0$, then we have*

$$
a(t) \leq Bz^t + \frac{c}{1-z}.
$$

Following Lemma 3, for $z \in (\rho, 1)$, we define

$$
V^{(T)}(z) = \sum_{t=1}^T \sum_{i=1}^m \left\|\tilde{\mathbf{\Sigma}}_i^{(t-1)} - \widehat{\mathbf{\Sigma}}\right\|_F^2 z^{-t},
$$

$$
D^{(T)}(z) = \sum_{t=1}^T \left\|\mathbf{D}_\diamond^{(t)}\right\|_F^2 z^{-t},
$$

$$
E^{(T)}(z) = \sum_{t=1}^T \left\|\mathbf{E}_{\mathbf{\Sigma}}^{(t)}\right\|_F^2 z^{-t}.
$$

Then multiplying $z^{-t}$ on both sides and summing from 0 to $T$, equation 55 becomes

$$
\frac{1}{\theta}\left(1 - \frac{6L\epsilon\sqrt{m}\rho}{\gamma\left(1-\rho\right)}\right)\left(zV^{(T)}(z) - \sum_{i=1}^m \left\|\tilde{\mathbf{\Sigma}}_i^{(0)} - \widehat{\mathbf{\Sigma}}\right\|_F^2\right)
$$

$$
\leq \left(1 - \frac{\mu}{\gamma} + \frac{1-\theta}{\theta}\left(1 - \frac{6L\epsilon\sqrt{m}\rho}{\gamma\left(1-\rho\right)}\right)\right)V^{(T)}(z) - \left(1 - \frac{L}{\gamma}\right)D^{(T)}(z)
$$

$$
+ \frac{4L\sqrt{m}}{\gamma\epsilon}\frac{\rho}{z-\rho}D^{(T)}(z) + \frac{8L\sqrt{m}}{\gamma\epsilon}\frac{\rho}{z-\rho}E^{(T)}(z)
$$

$$
+ \frac{\sqrt{m}}{\gamma\epsilon}\frac{\rho}{z-\rho}\left\|\mathbf{E}_{\mathbf{Y}}^{(0)}\right\|_F^2 + \frac{2L\sqrt{m}}{\gamma\epsilon}\frac{\rho}{z-\rho}\left\|\mathbf{E}_{\mathbf{\Sigma}}^{(0)}\right\|_F^2. \tag{56}
$$

Meanwhile, recall that equation 48, we have

$$\left\| \mathbf{E}_{\boldsymbol{\Sigma}}^{(t)} \right\|_F^2 \leq 2\rho^{2t} \left\| \mathbf{E}_{\boldsymbol{\Sigma}}^{(0)} \right\|_F^2 + \frac{2\rho}{1-\rho} \sum_{s=0}^{t-1} \rho^{t-s} \left\| \mathbf{D}_{\diamond}^{(s)} \right\|_F^2. \tag{57}$$

Multiplying $z^{-t}$ on both sides and summing from $0$ to $T$, equation 57 becomes

$$E^{(T)}(z) \leq 2\frac{\rho^2}{z-\rho^2} \left\| \mathbf{E}_{\boldsymbol{\Sigma}}^{(0)} \right\|^2 + \frac{2\rho}{1-\rho}\frac{\rho}{z-\rho} D^{(T)}(z).$$

Therefore, we have

$$\left( \frac{z}{\theta}\left(1 - \frac{6L\epsilon\sqrt{m}\rho}{\gamma(1-\rho)}\right) - \left(1 - \frac{\mu}{\gamma} + \frac{1-\theta}{\theta}\left(1 - \frac{6L\epsilon\sqrt{m}\rho}{\gamma(1-\rho)}\right)\right) \right) \left( zV^{(T)}(z) - \sum_{i=1}^{m} \left\| \tilde{\boldsymbol{\Sigma}}_i^{(0)} - \widehat{\boldsymbol{\Sigma}} \right\|_F^2 \right)$$

$$\leq -\left(1 - \frac{L}{\gamma} - \frac{4L\sqrt{m}}{\gamma\epsilon}\frac{\rho}{z-\rho} - \frac{16L\sqrt{m}}{\gamma\epsilon}\frac{2\rho}{1-\rho}\left(\frac{\rho}{z-\rho}\right)^2\right) D^{(T)}(z)$$

$$+ \left( \frac{16L\sqrt{m}}{\gamma\epsilon}\frac{\rho}{z-\rho}\frac{\rho^2}{z-\rho^2} + \frac{2L\sqrt{m}}{\gamma\epsilon}\frac{\rho}{z-\rho}\right) \left\| \mathbf{E}_{\boldsymbol{\Sigma}}^{(0)} \right\|^2 + \frac{\sqrt{m}}{\gamma\epsilon}\frac{\rho}{z-\rho}\left\| \mathbf{E}_{\mathbf{Y}}^{(0)} \right\|_F^2.$$

Then we prove that there exists some $z \in (\rho, 1)$ such that

$$\frac{z}{\theta}\left(1 - \frac{6L\epsilon\sqrt{m}\rho}{\gamma(1-\rho)}\right) - \left(1 - \frac{\mu}{\gamma} + \frac{1-\theta}{\theta}\left(1 - \frac{6L\epsilon\sqrt{m}\rho}{\gamma(1-\rho)}\right)\right) > 0, \tag{58}$$

and

$$1 - \frac{L}{\gamma} - \frac{4L\sqrt{m}}{\gamma\epsilon}\frac{\rho}{z-\rho} - \frac{16L\sqrt{m}}{\gamma\epsilon}\frac{2\rho}{1-\rho}\left(\frac{\rho}{z-\rho}\right)^2 > 0, \tag{59}$$

so that the linear convergence can be obtained using Lemma 4. Here we choose $\epsilon = \frac{\mu}{L}\frac{1-\rho}{12\sqrt{m}}$. Then equation 58 holds when

$$z > 1 - \theta\frac{2-\rho}{2\frac{\gamma}{\mu}-\rho}, \tag{60}$$

and equation 59 holds when

$$z > \frac{8\rho^2}{1-\rho}\left(\sqrt{1 + \frac{\mu\rho(\gamma-L)}{3L^2m}} - 1\right)^{-1} + \rho.$$

Note that equation 60 is satisfied when

$$z > 1 - \frac{(2-\rho)\theta L}{2\gamma\kappa}.$$

Since $\gamma \geq L + \frac{48L^2m\sqrt{\rho}}{\mu(1-\rho)^2}$, we have

$$\frac{8\rho^2}{1-\rho}\left(\sqrt{1 + \frac{\mu\rho(\gamma-L)}{3L^2m}} - 1\right)^{-1} + \rho$$

$$\leq \frac{8\rho^2}{1-\rho}\left(\sqrt{1 + \frac{16\rho\sqrt{\rho}}{(1-\rho)^2}} - 1\right)^{-1} + \rho$$

$$= \frac{\sqrt{\rho}(1-\rho)}{2\sqrt{\rho}}\left(\frac{4\sqrt{\rho\sqrt{\rho}}}{1-\rho} + 2\right) + \rho$$

$$\leq 3\rho + \sqrt{\rho}.$$

Since

$$\rho \leq \left(\frac{\sqrt{\kappa^2 + (12\kappa-2)(\kappa-1)} - \kappa}{6\kappa-1}\right)^2,$$

we have

$$3\rho + \sqrt{\rho} \le 1 - \frac{2-\rho}{2\kappa} \le 1 - \frac{(2-\rho)\theta L}{2\gamma\kappa}.$$

Finally, we have

$$\sum_{i=1}^{m} \left\| \mathbf{\Sigma}_i^{(t)} - \widehat{\mathbf{\Sigma}} \right\|_F^2 \le C_1' \left( 1 - \frac{1}{C_2'\kappa} \right)^t, \tag{61}$$

where

$$C_1' = \frac{\left( \frac{192mL^2}{\gamma\mu(1-\rho)} \frac{\rho}{z-\rho} \frac{\rho^2}{z-\rho^2} + \frac{24mL^2}{\gamma\mu(1-\rho)} \frac{\rho}{z-\rho} \right) \left\| \mathbf{E}_{\mathbf{\Sigma}}^{(0)} \right\|^2 + \frac{12mL}{\gamma\mu(1-\rho)} \frac{\rho}{z-\rho} \left\| \mathbf{E}_{\mathbf{Y}}^{(0)} \right\|_F^2}{\frac{z}{\theta}\left(1 - \frac{\rho\mu}{2\gamma}\right) - \left(1 - \frac{\mu}{\gamma} + \frac{1-\theta}{\theta}\left(1 - \frac{\rho\mu}{2\gamma}\right)\right)} + \sum_{i=1}^{m} \left\| \tilde{\mathbf{\Sigma}}_i^{(0)} - \widehat{\mathbf{\Sigma}} \right\|_F^2,$$

and

$$C_2' = \frac{2\gamma}{(2-\rho)\theta L},$$

which complete the proof.

### C.2.3 On the computation of $\overline{r}$, $\underline{r}$, and $e$

In the previous proof, we only established the existence of $\overline{r}$, $\underline{r}$, and $e$. Similar to the proof of Theorem 1, one way to compute them is to use the inequalities equation 19 and equation 20 obtained in Proposition 5 to derive a bound. Specificaly, we can choose $\overline{r}$ and $\underline{r}$ as the solutions of

$$\frac{1}{m}\left(ay - \tau \log y\right) = \left(d - \frac{1}{m}\right)\left(\tau - \tau \log \frac{\tau}{a}\right) - \frac{a}{N}\sum_{j=1}^{N} \left\| \mathbf{x}_j \mathbf{x}_j^\top \right\|_1 - \frac{1}{2}d^2 a^2 - \mathcal{V}\left(\mathbf{\Sigma}_\diamond^{(0)}, \mathbf{Y}_\diamond^{(0)}\right),$$

and $e$ as the solution of

$$c_2 y = d\left(\tau - \tau \log \frac{\tau}{a}\right) - \frac{a}{N}\sum_{j=1}^{N} \left\| \mathbf{x}_j \mathbf{x}_j^\top \right\|_1 - \frac{1}{2}d^2 a^2 - \mathcal{V}\left(\mathbf{\Sigma}_\diamond^{(0)}, \mathbf{Y}_\diamond^{(0)}\right),$$

with respect to the variables $y$.

### C.3 Proof of Theorem 3

We first characterize the local strong convexity within a neighborhood of $\mathbf{\Sigma}^\star$ with the following proposition.

**Proposition 10.** *(Local strong convexity) Suppose Assumption 2 holds. Assume $a = c_a\sqrt{\frac{KN}{\log d}}$ where $c_a \ge \frac{4}{\sqrt{2-\mu}}\sqrt{\frac{\log d}{N}}$ and $K = \max\{\sigma^2, (2\sigma)^{2(1+\nu)}\}$, and $N \ge \frac{16c_N \log d}{(2-\mu)^2}$. Let $\mathcal{H} = \frac{1}{m}\sum_{i=1}^{m}\mathcal{H}_i$. Then for any $\mathbf{\Sigma}_1, \mathbf{\Sigma}_2 \in \left\{\mathbf{\Sigma} \mid \|\mathbf{\Sigma} - \mathbf{\Sigma}^\star\|_{\max} \le \frac{a}{2}\right\}$, we have*

$$\langle \nabla\mathcal{H}(\mathbf{\Sigma}_1) - \nabla\mathcal{H}(\mathbf{\Sigma}_2), \mathbf{\Sigma}_1 - \mathbf{\Sigma}_2 \rangle \ge \frac{\mu_0}{2}\|\mathbf{\Sigma}_1 - \mathbf{\Sigma}_2\|_F^2$$

*with probability at least $1 - 2/d^{2(c_N-1)}$, where $\mu_0 \in (0, 2)$.*

*Proof.* Define $D_{kl} = \frac{1}{N}\sum_{j=1}^{N} I\left(|\Sigma_{kl}^\star - x_{jk}x_{jl}| \le \frac{a}{2}\right)$ for $k, l = 1, \ldots, d$. Since $a = c_a\sqrt{\frac{KN}{\log d}}$ and $c_a \ge \frac{4}{\sqrt{2-\mu_0}}\sqrt{\frac{\log d}{N}}$, we have $a \ge 4\sqrt{\frac{K}{2-\mu_0}}$. By Chebyshev's inequality, we have

$$\mathrm{E}(D_{kl}) = \mathrm{P}\left(|\Sigma_{kl}^\star - x_{jk}x_{jl}| \le \frac{a}{2}\right) \ge 1 - \frac{4K}{a^2} \ge \frac{2+\mu_0}{4}.$$

Then utilizing the fact that $I\left(|\Sigma_{kl}^\star - x_{jk}x_{jl}| \le \frac{a}{2}\right) \in \{0, 1\}$ and Hoeffding's inequality, we have

$$\mathrm{P}\left(D_{kl} \le \frac{\mu_0}{2}\right) \le \mathrm{P}\left(|D_{kl} - \mathrm{E}(D_{kl})| \ge \frac{2-\mu_0}{4}\right) \le 2\exp\left(-\frac{N(2-\mu_0)^2}{8}\right).$$

According to union bound and $N \geq \frac{16 c_N \log d}{(2-\mu_0)^2}$, we have

$$\mathrm{P}\left(\min_{k,l} D_{kl} \leq \frac{\mu_0}{2}\right) \leq 2d^2 \exp\left(-\frac{N\left(2-\mu_0\right)^2}{8}\right) \leq 2\exp\left(2\left(1-c_N\right)\log d\right).$$

Due to the non-decreasing nature of $h'$, for each $(k, l)$ we have

$$\left(h'\left((\Sigma_1)_{kl} - x_{jk}x_{jl}\right) - h'\left((\Sigma_2)_{kl} - x_{jk}x_{jl}\right)\right)\left((\Sigma_1)_{kl} - (\Sigma_2)_{kl}\right) \geq 0.$$

Conditioned on $\min_{k,l} D_{kl} \geq \frac{\mu_0}{2}$, we have

$$\frac{1}{N}\sum_{j=1}^{N}\left(h'\left((\Sigma_1)_{kl} - x_{jk}x_{jl}\right) - h'\left((\Sigma_2)_{kl} - x_{jk}x_{jl}\right)\right)\left((\Sigma_1)_{kl} - (\Sigma_2)_{kl}\right)$$

$$\geq \frac{1}{N}\sum_{j=1}^{N} I\left(|\Sigma_{kl}^{\star} - x_{jk}x_{jl}| \leq \frac{a}{2}\right)\left(h'\left((\Sigma_1)_{kl} - x_{jk}x_{jl}\right) - h'\left((\Sigma_2)_{kl} - x_{jk}x_{jl}\right)\right)\left((\Sigma_1)_{kl} - (\Sigma_2)_{kl}\right)$$

$$\geq \frac{1}{N}\sum_{j=1}^{N} I\left(|\Sigma_{kl}^{\star} - x_{jk}x_{jl}| \leq \frac{a}{2}\right)\left((\Sigma_1)_{kl} - (\Sigma_2)_{kl}\right)^2$$

$$\geq \frac{\mu_0}{2}\left((\Sigma_1)_{kl} - (\Sigma_2)_{kl}\right)^2.$$

Summing over all $(k, l)$ leads to the desired result. $\qquad\square$

Then we bound $\|\nabla\left(\mathcal{H}\left(\mathbf{\Sigma}^{\star}\right) - \tau\log\det\left(\mathbf{\Sigma}^{\star}\right)\right)\|_{\max}$ by the regularization parameter $\lambda$.

**Proposition 11.** *Suppose Assumptions 2 and 3 hold. Assume* $a = c_a\sqrt{\frac{KN}{\log d}}$, $\tau \leq c_\tau\left\|\left(\mathbf{\Sigma}^{\star}\right)^{-1}\right\|_{\max}^{-1}\sqrt{\frac{\log d}{N}}$, *and* $\lambda = c_\lambda\left(\left(\sqrt{6} + 2c_a + \frac{1}{c_a}\right)\sqrt{K} + c_\tau\right)\sqrt{\frac{\log d}{N}}$. *Then we have*

$$\|\nabla\left(\mathcal{H}\left(\mathbf{\Sigma}^{\star}\right) - \tau\log\det\left(\mathbf{\Sigma}^{\star}\right)\right)\|_{\max} \leq \frac{\lambda}{2}$$

*with probability at least* $1 - 2/d$.

*Proof.* For each $k, l = 1, \ldots, d$, we have

$$|\mathrm{E}\left(h'\left(\Sigma_{kl}^{\star} - x_{jk}x_{jl}\right)\right)|$$
$$= |\mathrm{E}\left(\left(\Sigma_{kl}^{\star} - x_{jk}x_{jl}\right) I\left(|\Sigma_{kl}^{\star} - x_{jk}x_{jl}| \leq a\right) + a\,\mathrm{sign}\left(\Sigma_{kl}^{\star} - x_{jk}x_{jl}\right) I\left(|\Sigma_{kl}^{\star} - x_{jk}x_{jl}| > a\right)\right)|$$
$$= |\mathrm{E}\left(\Sigma_{kl}^{\star} - x_{jk}x_{jl} + \left(a\,\mathrm{sign}\left(\Sigma_{kl}^{\star} - x_{jk}x_{jl}\right) - \left(\Sigma_{kl}^{\star} - x_{jk}x_{jl}\right)\right) I\left(|\Sigma_{kl}^{\star} - x_{jk}x_{jl}| > a\right)\right)|$$
$$= |\mathrm{E}\left(\Sigma_{kl}^{\star} - x_{jk}x_{jl}\right) + \mathrm{E}\left(\left(\Sigma_{kl}^{\star} - x_{jk}x_{jl} - a\,\mathrm{sign}\left(\Sigma_{kl}^{\star} - x_{jk}x_{jl}\right)\right) I\left(|\Sigma_{kl}^{\star} - x_{jk}x_{jl}| > a\right)\right)|$$
$$\leq \mathrm{E}\left(\left(|\Sigma_{kl}^{\star} - x_{jk}x_{jl}| - a\right) I\left(|\Sigma_{kl}^{\star} - x_{jk}x_{jl}| > a\right)\right)$$
$$\leq \frac{1}{|\Sigma_{kl}^{\star} - x_{jk}x_{jl}| + a}\mathrm{E}\left(\left(\left(\Sigma_{kl}^{\star} - x_{jk}x_{jl}\right)^2 - a^2\right) I\left(|\Sigma_{kl}^{\star} - x_{jk}x_{jl}| > a\right)\right)$$
$$\leq \frac{1}{a}\mathrm{E}\left(\left(\Sigma_{kl}^{\star} - x_{jk}x_{jl}\right)^2\right)$$
$$\leq \frac{K}{a}. \tag{62}$$

Then since the $a = c_a\sqrt{\frac{KN}{\log d}}$ and $N > \frac{16 c_N \log d}{(2-\mu_0)^2}$, we have

$$|h'\left(\Sigma_{kl}^{\star} - x_{jk}x_{jl}\right) - \mathrm{E}\left(h'\left(\Sigma_{kl}^{\star} - x_{jk}x_{jl}\right)\right)| \leq |h'\left(\Sigma_{kl}^{\star} - x_{jk}x_{jl}\right)| + |\mathrm{E}\left(h'\left(\Sigma_{kl}^{\star} - x_{jk}x_{jl}\right)\right)|$$
$$\leq a + \frac{K}{a} \leq 2a.$$

Then since $h$ is 1-Lipschitz smooth, for $n \geq 2$ we have

$$\mathrm{E}\left(\left(h'\left(\Sigma_{kl}^{\star} - x_{jk}x_{jl}\right) - \mathrm{E}\left(h'\left(\Sigma_{kl}^{\star} - x_{jk}x_{jl}\right)\right)\right)^n\right)$$

$$\leq (2a)^{n-2}\,\mathrm{E}\left(\left(h'\left(\Sigma_{kl}^{\star} - x_{jk}x_{jl}\right) - \mathrm{E}\left(h'\left(\Sigma_{kl}^{\star} - x_{jk}x_{jl}\right)\right)\right)^2\right)$$

$$= (2a)^{n-2}\,\mathrm{Var}\left(h'\left(\Sigma_{kl}^{\star} - x_{jk}x_{jl}\right)\right)$$

$$= (2a)^{n-2}\,\inf_u \mathrm{E}\left(\left(h'\left(\Sigma_{kl}^{\star} - x_{jk}x_{jl}\right) - u\right)^2\right)$$

$$\leq (2a)^{n-2}\,\mathrm{E}\left(\left(h'\left(\Sigma_{kl}^{\star} - x_{jk}x_{jl}\right) - h'\left(\mathrm{E}\left(\Sigma_{kl}^{\star} - x_{jk}x_{jl}\right)\right)\right)^2\right)$$

$$\leq (2a)^{n-2}\,\mathrm{E}\left(\left(\Sigma_{kl}^{\star} - x_{jk}x_{jl} - \mathrm{E}\left(\Sigma_{kl}^{\star} - x_{jk}x_{jl}\right)\right)^2\right)$$

$$= (2a)^{n-2}\,\mathrm{Var}\left(\Sigma_{kl}^{\star} - x_{jk}x_{jl}\right)$$

$$\leq (2a)^{n-2}\,K$$

$$\leq \frac{n!}{2}a^{n-2}K.$$

Then according to the Bernstein's inequality, we have

$$\mathrm{P}\left(\left|h'\left(\Sigma_{kl}^{\star} - x_{jk}x_{jl}\right) - \mathrm{E}\left(h'\left(\Sigma_{kl}^{\star} - x_{jk}x_{jl}\right)\right)\right| \geq t\right) \leq 2\exp\left(-\frac{Nt^2}{2\left(K + \frac{at}{3}\right)}\right).$$

Taking $t = \sqrt{\frac{6\sigma^2 \log d}{N}} + a\frac{2\log d}{N}$, we have

$$\mathrm{P}\left(\left|h'\left(\Sigma_{kl}^{\star} - x_{jk}x_{jl}\right) - \mathrm{E}\left(h'\left(\Sigma_{kl}^{\star} - x_{jk}x_{jl}\right)\right)\right| \geq \sqrt{\frac{6K\log d}{N}} + a\frac{2\log d}{N}\right)$$

$$\leq 2\exp\left(-\frac{N\left(\sqrt{\frac{6K\log d}{N}} + a\frac{2\log d}{N}\right)^2}{2\left(\sigma^2 + \frac{a}{3}\left(\sqrt{\frac{6K\log d}{N}} + a\frac{2\log d}{N}\right)\right)}\right)$$

$$\leq 2\exp\left(-\frac{\left(6\sigma^2 + 4a\sqrt{\frac{6K\log d}{N}} + 4a^2\frac{\log d}{N}\right)}{\left(6\sigma^2 + 2a\sqrt{\frac{6K\log d}{N}} + 4a^2\frac{\log d}{N}\right)}3\log d\right)$$

$$\leq \frac{2}{d^3}.$$

In conjunction with the union bound, we have

$$\mathrm{P}\left(\left\|\nabla\mathcal{H}\left(\Sigma^{\star}\right) - \mathrm{E}\left(\nabla\mathcal{H}\left(\Sigma^{\star}\right)\right)\right\|_{\max} \geq \sqrt{\frac{6K\log d}{N}} + a\frac{2\log d}{N}\right) \leq \frac{2}{d}.$$

Recall equation 62, we have

$$\mathrm{P}\left(\left\|\nabla\mathcal{H}\left(\Sigma^{\star}\right)\right\|_{\max} \geq \sqrt{\frac{6K\log d}{N}} + a\frac{2\log d}{N}\right)$$

$$\leq \mathrm{P}\left(\left\|\nabla\mathcal{H}\left(\Sigma^{\star}\right) - \mathrm{E}\left(\nabla\mathcal{H}\left(\Sigma^{\star}\right)\right)\right\|_{\max} + \left\|\mathrm{E}\left(\nabla\mathcal{H}\left(\Sigma^{\star}\right)\right)\right\|_{\max} \geq \sqrt{\frac{6K\log d}{N}} + a\frac{2\log d}{N} + \frac{K}{a}\right)$$

$$\leq \frac{2}{d}.$$

Since $\tau \leq c_\tau \left\|\left(\Sigma^{\star}\right)^{-1}\right\|_{\max}^{-1}\sqrt{\frac{\log d}{N}}$, we have

$$\left\|\nabla\left(-\tau\log\det\left(\Sigma^{\star}\right)\right)\right\|_{\max} = \tau\left\|\left(\Sigma^{\star}\right)^{-1}\right\|_{\max} \leq c_\tau\sqrt{\frac{\log d}{N}}.$$

Combining the two inequalities above, we have

$$\left\|\nabla\left(\mathcal{H}\left(\mathbf{\Sigma}^{\star}\right)-\tau\log\det\left(\mathbf{\Sigma}^{\star}\right)\right)\right\|_{\max} \leq \sqrt{\frac{6K\log d}{N}} + a\frac{2\log d}{N} + \frac{K}{a} + c_{\tau}\sqrt{\frac{\log d}{N}}$$

with probability at least $1 - 2/d$. Finally, since $a = c_a\sqrt{\frac{KN}{\log d}}$ and $\lambda = c_{\lambda}\left(\left(\sqrt{6}+2c_a+\frac{1}{c_a}\right)\sqrt{K}+c_{\tau}\right)\sqrt{\frac{\log d}{N}}$, we have the desired result. $\qquad\square$

Based on Proposition 11, we can further bound $\left\|\left(\widehat{\mathbf{\Sigma}}-\mathbf{\Sigma}^{\star}\right)_{\mathcal{S}^c}\right\|_1$.

**Proposition 12.** *Based on Proposition 11 and suppose that there exists a constant $r_q$ such that* $\left\|\widehat{\mathbf{\Sigma}}-\mathbf{\Sigma}^{\star}\right\|_{\max} \leq r_q$, *we have*

$$\left\|\left(\widehat{\mathbf{\Sigma}}-\mathbf{\Sigma}^{\star}\right)_{\mathcal{S}^c}\right\|_1 \leq \frac{2\lambda + \left\|\nabla\left(\frac{1}{m}\sum_{i=1}^{m}\mathcal{H}_i\left(\mathbf{\Sigma}^{\star}\right)-\tau\log\det\left(\mathbf{\Sigma}^{\star}\right)\right)\right\|_{\max} + L_q r_q}{\lambda - \left\|\nabla\left(\frac{1}{m}\sum_{i=1}^{m}\mathcal{H}_i\left(\mathbf{\Sigma}^{\star}\right)-\tau\log\det\left(\mathbf{\Sigma}^{\star}\right)\right)\right\|_{\max} - L_q r_q}\left\|\left(\widehat{\mathbf{\Sigma}}-\mathbf{\Sigma}^{\star}\right)_{\mathcal{S}}\right\|_1.$$

*Proof.* Due to the first-order optimization condition of equation 2, for $\mathbf{\Phi} \in \partial\lambda\left\|\widehat{\mathbf{\Sigma}}\right\|_{1,\text{off}}$, we have

$$0 \geq \left\langle\nabla\mathcal{F}\left(\widehat{\mathbf{\Sigma}}\right)+\mathbf{\Phi},\widehat{\mathbf{\Sigma}}-\mathbf{\Sigma}^{\star}\right\rangle$$

$$\geq \left\langle\nabla\left(\mathcal{H}\left(\widehat{\mathbf{\Sigma}}\right)-\tau\log\det\left(\widehat{\mathbf{\Sigma}}\right)\right)-\nabla\left(\mathcal{H}\left(\mathbf{\Sigma}^{\star}\right)-\tau\log\det\left(\mathbf{\Sigma}^{\star}\right)\right),\widehat{\mathbf{\Sigma}}-\mathbf{\Sigma}^{\star}\right\rangle$$

$$+\left\langle\nabla\left(\mathcal{H}\left(\mathbf{\Sigma}^{\star}\right)-\tau\log\det\left(\mathbf{\Sigma}^{\star}\right)\right),\widehat{\mathbf{\Sigma}}-\mathbf{\Sigma}^{\star}\right\rangle-\left\langle\nabla\mathcal{Q}\left(\widehat{\mathbf{\Sigma}}\right)-\nabla\mathcal{Q}\left(\mathbf{\Sigma}^{\star}\right),\widehat{\mathbf{\Sigma}}-\mathbf{\Sigma}^{\star}\right\rangle$$

$$-\left\langle\nabla\mathcal{Q}\left(\mathbf{\Sigma}^{\star}\right),\widehat{\mathbf{\Sigma}}-\mathbf{\Sigma}^{\star}\right\rangle+\left\langle\mathbf{\Phi},\widehat{\mathbf{\Sigma}}-\mathbf{\Sigma}^{\star}\right\rangle.$$

Due to the convexity of Huber loss, we have

$$\left\langle\nabla\left(\mathcal{H}\left(\widehat{\mathbf{\Sigma}}\right)-\tau\log\det\left(\widehat{\mathbf{\Sigma}}\right)\right)-\nabla\left(\mathcal{H}\left(\mathbf{\Sigma}^{\star}\right)-\tau\log\det\left(\mathbf{\Sigma}^{\star}\right)\right),\widehat{\mathbf{\Sigma}}-\mathbf{\Sigma}^{\star}\right\rangle \geq 0.$$

In addition, we have

$$\left\langle\mathbf{\Phi},\widehat{\mathbf{\Sigma}}-\mathbf{\Sigma}^{\star}\right\rangle = \left\langle\mathbf{\Phi}_{\mathcal{S}^c},\left(\widehat{\mathbf{\Sigma}}-\mathbf{\Sigma}^{\star}\right)_{\mathcal{S}^c}\right\rangle+\left\langle\mathbf{\Phi}_{\mathcal{S}},\left(\widehat{\mathbf{\Sigma}}-\mathbf{\Sigma}^{\star}\right)_{\mathcal{S}}\right\rangle$$

$$= \left\langle\mathbf{\Phi}_{\mathcal{S}^c},\widehat{\mathbf{\Sigma}}_{\mathcal{S}^c}\right\rangle+\left\langle\mathbf{\Phi}_{\mathcal{S}},\left(\widehat{\mathbf{\Sigma}}-\mathbf{\Sigma}^{\star}\right)_{\mathcal{S}}\right\rangle$$

$$= \lambda\left\|\widehat{\mathbf{\Sigma}}_{\mathcal{S}^c}\right\|_1+\left\langle\mathbf{\Phi}_{\mathcal{S}},\left(\widehat{\mathbf{\Sigma}}-\mathbf{\Sigma}^{\star}\right)_{\mathcal{S}}\right\rangle$$

$$\geq \lambda\left\|\left(\widehat{\mathbf{\Sigma}}-\mathbf{\Sigma}^{\star}\right)_{\mathcal{S}^c}\right\|_1-\lambda\left\|\left(\widehat{\mathbf{\Sigma}}-\mathbf{\Sigma}^{\star}\right)_{\mathcal{S}}\right\|_1. \qquad (63)$$

According to Assumption 3, we have

$$\left\langle\nabla\mathcal{Q}\left(\mathbf{\Sigma}^{\star}\right),\widehat{\mathbf{\Sigma}}-\mathbf{\Sigma}^{\star}\right\rangle = \left\langle\nabla\mathcal{Q}\left(\mathbf{\Sigma}^{\star}\right)_{\mathcal{S}},\left(\widehat{\mathbf{\Sigma}}-\mathbf{\Sigma}^{\star}\right)_{\mathcal{S}}\right\rangle$$

$$\leq \lambda\left\|\left(\widehat{\mathbf{\Sigma}}-\mathbf{\Sigma}^{\star}\right)_{\mathcal{S}}\right\|_1.$$

Combining the four inequalities above, we have

$$0 \geq -\left\|\nabla\left(\mathcal{H}\left(\mathbf{\Sigma}^{\star}\right)-\tau\log\det\left(\mathbf{\Sigma}^{\star}\right)\right)\right\|_{\max}\left\|\widehat{\mathbf{\Sigma}}-\mathbf{\Sigma}^{\star}\right\|_1-L_q\left\|\widehat{\mathbf{\Sigma}}-\mathbf{\Sigma}^{\star}\right\|_{\max}\left\|\widehat{\mathbf{\Sigma}}-\mathbf{\Sigma}^{\star}\right\|_1$$

$$+\lambda\left\|\left(\widehat{\mathbf{\Sigma}}-\mathbf{\Sigma}^{\star}\right)_{\mathcal{S}^c}\right\|_1-2\lambda\left\|\left(\widehat{\mathbf{\Sigma}}-\mathbf{\Sigma}^{\star}\right)_{\mathcal{S}}\right\|_1.$$

Decomposing $\left\|\widehat{\mathbf{\Sigma}}-\mathbf{\Sigma}^{\star}\right\|_1$ into $\mathcal{S}^c$ and $\mathcal{S}$, we have

$$\left\|\left(\widehat{\mathbf{\Sigma}}-\mathbf{\Sigma}^{\star}\right)_{\mathcal{S}^c}\right\|_1 \leq \frac{2\lambda + \left\|\nabla\left(\mathcal{H}\left(\mathbf{\Sigma}^{\star}\right)-\tau\log\det\left(\mathbf{\Sigma}^{\star}\right)\right)\right\|_{\max}+L_q\left\|\widehat{\mathbf{\Sigma}}-\mathbf{\Sigma}^{\star}\right\|_{\max}}{\lambda - \left\|\nabla\left(\mathcal{H}\left(\mathbf{\Sigma}^{\star}\right)-\tau\log\det\left(\mathbf{\Sigma}^{\star}\right)\right)\right\|_{\max}-L_q\left\|\widehat{\mathbf{\Sigma}}-\mathbf{\Sigma}^{\star}\right\|_{\max}}\left\|\left(\widehat{\mathbf{\Sigma}}-\mathbf{\Sigma}^{\star}\right)_{\mathcal{S}}\right\|_1.$$

Since $\left\|\widehat{\mathbf{\Sigma}}-\mathbf{\Sigma}^{\star}\right\|_{\max} \leq r_q$, we have the desired result. $\qquad\square$

Then we can bound the estimation error $\left\| \widehat{\boldsymbol{\Sigma}} - \boldsymbol{\Sigma}^\star \right\|_F$.

**Theorem 5.** *Suppose that* $N > \frac{38 c_\lambda}{\mu_0 c_a \sqrt{K}} \left( \left( \sqrt{6} + 2 c_a + \frac{1}{c_a} \right) \sqrt{K} + c_\tau \right) \sqrt{s} \log d,$ $\lambda =$ $c_\lambda \left( \left( \sqrt{6} + 2 c_a + \frac{1}{c_a} \right) \sqrt{K} + c_\tau \right) \sqrt{\frac{\log d}{N}},$ $a = c_a \sqrt{\frac{KN}{\log d}},$ *and* $L_q \leq \frac{c_q \mu_0}{\sqrt{s}},$ *where* $c_a, c_\lambda, c_\tau, c_q > 0,$ *and* $\mu_0 \in (0, 2).$ *Based on Propositions 10, 11, and 12, we have*

$$\left\| \widehat{\boldsymbol{\Sigma}} - \boldsymbol{\Sigma}^\star \right\|_F < \frac{a}{2},$$

*and*

$$\left\| \widehat{\boldsymbol{\Sigma}} - \boldsymbol{\Sigma}^\star \right\|_F \leq \frac{60}{\mu_0} c_\lambda \left( \left( \sqrt{6} + 2 c_a + \frac{1}{c_a} \right) \sqrt{K} + c_\tau \right) \sqrt{\frac{s \log d}{N}}.$$

*Proof.* Based on Propositions 11 and 12, when $L_q r_q \leq \frac{\lambda}{4}$, we have

$$\left\| \left( \widehat{\boldsymbol{\Sigma}} - \boldsymbol{\Sigma}^\star \right)_{\mathcal{S}^c} \right\|_1 \leq 11 \left\| \left( \widehat{\boldsymbol{\Sigma}} - \boldsymbol{\Sigma}^\star \right)_{\mathcal{S}} \right\|_1,$$

and hence

$$\left\| \widehat{\boldsymbol{\Sigma}} - \boldsymbol{\Sigma}^\star \right\|_1 \leq 12 \sqrt{s} \left\| \widehat{\boldsymbol{\Sigma}} - \boldsymbol{\Sigma}^\star \right\|_F. \tag{64}$$

Define $\eta = \sup \left\{ u \in (0, 1] \mid u \left\| \widehat{\boldsymbol{\Sigma}} - \boldsymbol{\Sigma}^\star \right\|_{\max} \leq \frac{a}{2} \right\}$, and $\widehat{\boldsymbol{\Sigma}}_\eta = (1 - \eta) \boldsymbol{\Sigma}^\star + \eta \widehat{\boldsymbol{\Sigma}}$. Note that $\eta = 1$ if $\left\| \widehat{\boldsymbol{\Sigma}} - \boldsymbol{\Sigma}^\star \right\|_{\max} \leq \frac{a}{2}$ and $\eta \in (0, 1)$ otherwise. Meanwhile, if $\left\| \widehat{\boldsymbol{\Sigma}}_\eta - \boldsymbol{\Sigma}^\star \right\|_{\max} < \frac{a}{2}$, then $\widehat{\boldsymbol{\Sigma}}_\eta = \widehat{\boldsymbol{\Sigma}}$. By convexity of Huber loss, we have

$$\left\langle \nabla \mathcal{H} \left( \widehat{\boldsymbol{\Sigma}}_\eta \right) - \nabla \mathcal{H} \left( \boldsymbol{\Sigma}^\star \right), \widehat{\boldsymbol{\Sigma}}_\eta - \boldsymbol{\Sigma}^\star \right\rangle \leq \eta \left\langle \nabla \mathcal{H} \left( \widehat{\boldsymbol{\Sigma}} \right) - \nabla \mathcal{H} \left( \boldsymbol{\Sigma}^\star \right), \widehat{\boldsymbol{\Sigma}} - \boldsymbol{\Sigma}^\star \right\rangle. \tag{65}$$

Since $\left\| \widehat{\boldsymbol{\Sigma}}_\eta - \boldsymbol{\Sigma}^\star \right\|_{\max} \leq \frac{a}{2}$, according to Proposition 10, we have

$$\left\langle \nabla \mathcal{H} \left( \widehat{\boldsymbol{\Sigma}}_\eta \right) - \nabla \mathcal{H} \left( \boldsymbol{\Sigma}^\star \right), \widehat{\boldsymbol{\Sigma}}_\eta - \boldsymbol{\Sigma}^\star \right\rangle \geq \frac{\mu_0}{2} \left\| \widehat{\boldsymbol{\Sigma}}_\eta - \boldsymbol{\Sigma}^\star \right\|_F^2.$$

Combining the two inequalities above, we have

$$\left\| \widehat{\boldsymbol{\Sigma}}_\eta - \boldsymbol{\Sigma}^\star \right\|_F^2 \leq \frac{2\eta}{\mu_0} \left\langle \nabla \mathcal{H} \left( \widehat{\boldsymbol{\Sigma}} \right) - \nabla \mathcal{H} \left( \boldsymbol{\Sigma}^\star \right), \widehat{\boldsymbol{\Sigma}} - \boldsymbol{\Sigma}^\star \right\rangle. \tag{66}$$

Then we bound the right side of equation 66. For $\boldsymbol{\Phi} \in \partial \lambda \left\| \widehat{\boldsymbol{\Sigma}} \right\|_{1, \text{off}}$, according to the first-order optimality condition of equation 2, we have

$$\left\langle \nabla \mathcal{H} \left( \widehat{\boldsymbol{\Sigma}} \right) - \nabla \mathcal{H} \left( \boldsymbol{\Sigma}^\star \right), \widehat{\boldsymbol{\Sigma}} - \boldsymbol{\Sigma}^\star \right\rangle$$

$$= \left\langle \nabla \mathcal{F} \left( \widehat{\boldsymbol{\Sigma}} \right) + \boldsymbol{\Phi}, \widehat{\boldsymbol{\Sigma}} - \boldsymbol{\Sigma}^\star \right\rangle - \left\langle \nabla \left( \mathcal{H} \left( \boldsymbol{\Sigma}^\star \right) - \tau \log \det \left( \boldsymbol{\Sigma}^\star \right) \right), \widehat{\boldsymbol{\Sigma}} - \boldsymbol{\Sigma}^\star \right\rangle$$

$$- \left\langle \boldsymbol{\Phi}, \widehat{\boldsymbol{\Sigma}} - \boldsymbol{\Sigma}^\star \right\rangle - \tau \left\langle \left( \boldsymbol{\Sigma}^\star \right)^{-1} - \widehat{\boldsymbol{\Sigma}}^{-1}, \widehat{\boldsymbol{\Sigma}} - \boldsymbol{\Sigma}^\star \right\rangle + \left\langle \nabla \mathcal{Q} \left( \hat{\boldsymbol{\Sigma}} \right), \hat{\boldsymbol{\Sigma}} - \boldsymbol{\Sigma}^\star \right\rangle$$

$$\leq - \left\langle \nabla \left( \mathcal{H} \left( \boldsymbol{\Sigma}^\star \right) - \tau \log \det \left( \boldsymbol{\Sigma}^\star \right) \right), \widehat{\boldsymbol{\Sigma}} - \boldsymbol{\Sigma}^\star \right\rangle - \left\langle \boldsymbol{\Phi}, \widehat{\boldsymbol{\Sigma}} - \boldsymbol{\Sigma}^\star \right\rangle + \left\langle \nabla \mathcal{Q} \left( \hat{\boldsymbol{\Sigma}} \right), \hat{\boldsymbol{\Sigma}} - \boldsymbol{\Sigma}^\star \right\rangle.$$

For the right side of the above inequality, we have

$$- \left\langle \nabla \left( \mathcal{H} \left( \boldsymbol{\Sigma}^\star \right) - \tau \log \det \left( \boldsymbol{\Sigma}^\star \right) \right), \widehat{\boldsymbol{\Sigma}} - \boldsymbol{\Sigma}^\star \right\rangle$$

$$\leq \left\| \nabla \left( \mathcal{H} \left( \boldsymbol{\Sigma}^\star \right) - \tau \log \det \left( \boldsymbol{\Sigma}^\star \right) \right) \right\|_{\max} \left\| \hat{\boldsymbol{\Sigma}} - \boldsymbol{\Sigma}^\star \right\|_1.$$

Then recall equation 63 and Assumption 3, we have

$$\left\langle \nabla \mathcal{H} \left( \widehat{\boldsymbol{\Sigma}} \right) - \nabla \mathcal{H} \left( \boldsymbol{\Sigma}^\star \right), \widehat{\boldsymbol{\Sigma}} - \boldsymbol{\Sigma}^\star \right\rangle$$

$$\leq \left\| \nabla \left( \mathcal{H} \left( \boldsymbol{\Sigma}^\star \right) - \tau \log \det \left( \boldsymbol{\Sigma}^\star \right) \right) \right\|_{\max} \left\| \hat{\boldsymbol{\Sigma}} - \boldsymbol{\Sigma}^\star \right\|_1$$

$$- \lambda \left\| \left( \widehat{\boldsymbol{\Sigma}} - \boldsymbol{\Sigma}^\star \right)_{\mathcal{S}^c} \right\|_1 + \lambda \left\| \left( \widehat{\boldsymbol{\Sigma}} - \boldsymbol{\Sigma}^\star \right)_{\mathcal{S}} \right\|_1 + \lambda \left( \left\| \left( \hat{\boldsymbol{\Sigma}} - \boldsymbol{\Sigma}^\star \right)_{\mathcal{S}} \right\|_1 + \left\| \left( \hat{\boldsymbol{\Sigma}} - \boldsymbol{\Sigma}^\star \right)_{\mathcal{S}^c} \right\|_1 \right)$$

$$= \left\| \nabla \left( \mathcal{H} \left( \boldsymbol{\Sigma}^\star \right) - \tau \log \det \left( \boldsymbol{\Sigma}^\star \right) \right) \right\|_{\max} \left\| \hat{\boldsymbol{\Sigma}} - \boldsymbol{\Sigma}^\star \right\|_1 + 2\lambda \left\| \left( \hat{\boldsymbol{\Sigma}} - \boldsymbol{\Sigma}^\star \right)_{\mathcal{S}} \right\|_1. \tag{67}$$

Combining Proposition 11, equation 64, equation 66, and equation 67, we have

$$\left\langle \nabla\mathcal{H}\left(\widehat{\boldsymbol{\Sigma}}\right) - \nabla\mathcal{H}\left(\boldsymbol{\Sigma}^\star\right), \widehat{\boldsymbol{\Sigma}} - \boldsymbol{\Sigma}^\star \right\rangle \leq 30\lambda\sqrt{s}\left\|\widehat{\boldsymbol{\Sigma}} - \boldsymbol{\Sigma}^\star\right\|_F. \tag{68}$$

Combining equation 66 and equation 68 leads to

$$\left\|\widehat{\boldsymbol{\Sigma}}_\eta - \boldsymbol{\Sigma}^\star\right\|_F^2 \leq \frac{60}{\mu_0}\lambda\sqrt{s}\left\|\widehat{\boldsymbol{\Sigma}}_\eta - \boldsymbol{\Sigma}^\star\right\|_F.$$

Since $a = c_a\sqrt{\frac{KN}{\log d}}$, $N > \frac{38c_\lambda}{\mu_0 c_a\sqrt{K}}\left(\left(\sqrt{6} + 2c_a + \frac{1}{c_a}\right)\sqrt{K} + c_\tau\right)\sqrt{s}\log d$, and $\lambda = c_\lambda\left(\left(\sqrt{6} + 2c_a + \frac{1}{c_a}\right)\sqrt{K} + c_\tau\right)\sqrt{\frac{\log d}{N}}$, we have $\frac{a}{2} > \frac{60}{\mu_0}\lambda\sqrt{s}$. Therefore, we have $\left\|\widehat{\boldsymbol{\Sigma}}_\eta - \boldsymbol{\Sigma}^\star\right\|_F < \frac{a}{2}$, and hence $\widehat{\boldsymbol{\Sigma}}_\eta = \widehat{\boldsymbol{\Sigma}}$, which means that $\left\|\widehat{\boldsymbol{\Sigma}} - \boldsymbol{\Sigma}^\star\right\|_F < \frac{a}{2}$ and

$$\left\|\widehat{\boldsymbol{\Sigma}} - \boldsymbol{\Sigma}^\star\right\|_F \leq \frac{60}{\mu_0}c_\lambda\left(\left(\sqrt{6} + 2c_a + \frac{1}{c_a}\right)\sqrt{K} + c_\tau\right)\sqrt{\frac{s\log d}{N}}.$$

Moreover, since $L_q \leq \frac{c_q\mu_0}{\sqrt{s}}$, we have $L_q\left\|\widehat{\boldsymbol{\Sigma}} - \boldsymbol{\Sigma}^\star\right\|_{\max} \leq L_q\left\|\widehat{\boldsymbol{\Sigma}} - \boldsymbol{\Sigma}^\star\right\|_F \leq \frac{60}{\mu_0}L_q\lambda\sqrt{s} \leq \frac{\lambda}{4}$ if $c_q \leq \frac{1}{240}$, which complete the proof. $\square$

Based on Propositions 10, 11, and 12, and Theorem 5, we can finally prove Theorem 3. According to Theorem 5, we have $\left\|\widehat{\boldsymbol{\Sigma}} - \boldsymbol{\Sigma}^\star\right\|_F < \frac{a}{2}$. Then recall Proposition 10 and $L_q \leq \frac{c_q\mu_0}{\sqrt{s}}$ where $c_q \in (0, \sqrt{s})$, we have

$$\mathcal{F}\left(\widehat{\boldsymbol{\Sigma}}\right) - \mathcal{F}\left(\boldsymbol{\Sigma}^\star\right) - \left\langle \nabla\mathcal{F}\left(\boldsymbol{\Sigma}^\star\right), \widehat{\boldsymbol{\Sigma}} - \boldsymbol{\Sigma}^\star \right\rangle \geq \frac{\mu_0 - L_q}{2}\left\|\widehat{\boldsymbol{\Sigma}} - \boldsymbol{\Sigma}^\star\right\|_F^2,$$

and

$$\mathcal{F}\left(\boldsymbol{\Sigma}^\star\right) - \mathcal{F}\left(\widehat{\boldsymbol{\Sigma}}\right) - \left\langle \nabla\mathcal{F}\left(\widehat{\boldsymbol{\Sigma}}\right), \boldsymbol{\Sigma}^\star - \widehat{\boldsymbol{\Sigma}} \right\rangle \geq \frac{\mu_0 - L_q}{2}\left\|\boldsymbol{\Sigma}^\star - \widehat{\boldsymbol{\Sigma}}\right\|_F^2.$$

Since $\lambda\|\cdot\|_{1,\text{off}}$ is convex, for $\boldsymbol{\Phi} \in \partial\lambda\left\|\widehat{\boldsymbol{\Sigma}}\right\|_{1,\text{off}}$ and $\boldsymbol{\Psi} \in \partial\lambda\|\boldsymbol{\Sigma}^\star\|_{1,\text{off}}$, we have

$$\lambda\left\|\widehat{\boldsymbol{\Sigma}}\right\|_{1,\text{off}} - \lambda\|\boldsymbol{\Sigma}^\star\|_{1,\text{off}} - \left\langle \boldsymbol{\Psi}, \widehat{\boldsymbol{\Sigma}} - \boldsymbol{\Sigma}^\star \right\rangle \geq 0,$$

and

$$\lambda\|\boldsymbol{\Sigma}^\star\|_{1,\text{off}} - \lambda\left\|\widehat{\boldsymbol{\Sigma}}\right\|_{1,\text{off}} - \left\langle \boldsymbol{\Phi}, \boldsymbol{\Sigma}^\star - \widehat{\boldsymbol{\Sigma}} \right\rangle \geq 0.$$

According to the first-order optimality condition of equation 2, we have

$$\left\langle \nabla\mathcal{F}\left(\widehat{\boldsymbol{\Sigma}}\right) + \boldsymbol{\Phi}, \boldsymbol{\Sigma}^\star - \widehat{\boldsymbol{\Sigma}} \right\rangle \geq 0.$$

Combining the above inequalities, we have

$$\begin{aligned}
(\mu_0 - L_q)\left\|\widehat{\boldsymbol{\Sigma}} - \boldsymbol{\Sigma}^\star\right\|_F^2 &\leq \left\langle \nabla\mathcal{F}\left(\boldsymbol{\Sigma}^\star\right) + \boldsymbol{\Psi}, \boldsymbol{\Sigma}^\star - \widehat{\boldsymbol{\Sigma}} \right\rangle \\
&\leq \sum_{k,l}\left|(\nabla\mathcal{F}\left(\boldsymbol{\Sigma}^\star\right) + \boldsymbol{\Psi})_{kl}\right|\left|\left(\boldsymbol{\Sigma}^\star - \widehat{\boldsymbol{\Sigma}}\right)_{kl}\right|.
\end{aligned} \tag{69}$$

Then for $(k,l) \in \mathcal{S}^c$, according to Assumption 3, we have $q'_\lambda(\boldsymbol{\Sigma}^\star_{kl}) = 0$. Moreover, according to Proposition 11, we have

$$\|\nabla(\mathcal{H}(\boldsymbol{\Sigma}^\star) - \tau\log\det(\boldsymbol{\Sigma}^\star))\|_{\max} \leq \frac{\lambda}{2},$$

and hence $|\nabla\mathcal{F}(\boldsymbol{\Sigma}^\star)_{kl}| \leq \lambda$. Then since $\boldsymbol{\Psi}_{kl} \in [-\lambda, \lambda]$, there exists a $\boldsymbol{\Psi}$ such that $|(\nabla\mathcal{F}(\boldsymbol{\Sigma}^\star) + \boldsymbol{\Psi})_{kl}| = 0$. Therefore, we have

$$\sum_{(k,l)\in\mathcal{S}^c}|(\nabla\mathcal{F}(\boldsymbol{\Sigma}^\star) + \boldsymbol{\Psi})_{kl}|\left|\left(\boldsymbol{\Sigma}^\star - \widehat{\boldsymbol{\Sigma}}\right)_{kl}\right| = 0. \tag{70}$$

For $(k, l) \in \mathcal{S}$, since $b\lambda \leq \left|\mathbf{\Sigma}_{ij}^\star\right|$, by Assumption 3, we have $p_\lambda'\left(\mathbf{\Sigma}_{kl}^\star\right) = 0$. Therefore, we have

$$\sum_{(k,l)\in\mathcal{S}} \left|(\nabla\mathcal{F}\left(\mathbf{\Sigma}^\star\right) + \mathbf{\Psi})_{kl}\right| \left|\left(\mathbf{\Sigma}^\star - \widehat{\mathbf{\Sigma}}\right)_{kl}\right|$$

$$\leq \sum_{(k,l)\in\mathcal{S}} \left|(\nabla\left(\mathcal{H}\left(\mathbf{\Sigma}^\star\right) - \tau\log\det\left(\mathbf{\Sigma}^\star\right)\right))_{kl}\right| \left|\left(\mathbf{\Sigma}^\star - \widehat{\mathbf{\Sigma}}\right)_{kl}\right|$$

$$\leq \left\|(\nabla\left(\mathcal{H}\left(\mathbf{\Sigma}^\star\right) - \tau\log\det\left(\mathbf{\Sigma}^\star\right)\right))_\mathcal{S}\right\|_F \left\|\mathbf{\Sigma}^\star - \widehat{\mathbf{\Sigma}}\right\|_F. \tag{71}$$

Then similar to the proof of Proposition 11, we have

$$\left|\mathrm{E}\left(h'\left(\Sigma_{kl}^\star - x_{jk}x_{jl}\right)\right)\right|$$
$$\leq \mathrm{E}\left(\left(|\Sigma_{kl}^\star - x_{jk}x_{jl}| - a\right) I\left(|\Sigma_{kl}^\star - x_{jk}x_{jl}| > a\right)\right)$$
$$\leq \frac{1}{|\Sigma_{kl}^\star - x_{jk}x_{jl}|^{1+\nu} + a^{1+\nu}} \mathrm{E}\left(\left(\left(\Sigma_{kl}^\star - x_{jk}x_{jl}\right)^{2(1+\nu)} - a^{2(1+\nu)}\right) I\left(|\Sigma_{kl}^\star - x_{jk}x_{jl}| > a\right)\right)$$
$$\leq \frac{1}{a^{1+\nu}} \mathrm{E}\left(\left(\Sigma_{kl}^\star - x_{jk}x_{jl}\right)^{2(1+\nu)}\right)$$
$$\leq \frac{K}{a^{1+\nu}}.$$

In addition, we have

$$\mathrm{E}\left(\left(h'\left(\Sigma_{kl}^\star - x_{jk}x_{jl}\right) - \mathrm{E}\left(h'\left(\Sigma_{kl}^\star - x_{jk}x_{jl}\right)\right)\right)^2\right) \leq \mathrm{Var}\left(\Sigma_{kl}^\star - x_{jk}x_{jl}\right) \leq K.$$

Therefore, recall that $a = c_a\sqrt{\frac{KN}{\log d}}$ and $N > \frac{(\log d)^{1+\frac{1}{\nu}}}{c_a^{2\left(1+\frac{1}{\nu}\right)}K}$ we have

$$\mathrm{E}\left(\left(\frac{1}{N}\sum_{j=1}^N h'\left(\Sigma_{kl}^\star - x_{jk}x_{jl}\right)\right)^2\right)$$

$$= \left(\frac{1}{N}\sum_{j=1}^N \mathrm{E}\left(h'\left(\Sigma_{kl}^\star - x_{jk}x_{jl}\right)\right)\right)^2$$

$$+ \mathrm{E}\left(\left(\frac{1}{N}\sum_{j=1}^N \left(h'\left(\Sigma_{kl}^\star - x_{jk}x_{jl}\right) - \mathrm{E}\left(h'\left(\Sigma_{kl}^\star - x_{jk}x_{jl}\right)\right)\right)\right)^2\right)$$

$$\leq \frac{K^2}{a^{2(1+\nu)}} + \frac{K}{N}$$

$$\leq \frac{2K}{N},$$

and hence

$$\mathrm{E}\left(\|\nabla\mathcal{H}\left(\mathbf{\Sigma}^\star\right)_\mathcal{S}\|_F\right) \leq \mathrm{E}\left(\sqrt{\sum_{(k,l)\in\mathcal{S}}\left(\frac{1}{N}\sum_{j=1}^N h'\left(\Sigma_{kl}^\star - x_{jk}x_{jl}\right)\right)^2}\right)$$

$$\leq \sqrt{\sum_{(k,l)\in\mathcal{S}} \mathrm{E}\left(\left(\frac{1}{N}\sum_{j=1}^N h'\left(\Sigma_{kl}^\star - x_{jk}x_{jl}\right)\right)^2\right)}$$

$$\leq \sqrt{\frac{2sK}{N}}.$$

According to Markov's inequality, we have

$$\mathrm{P}\left(\|\nabla\mathcal{H}\left(\boldsymbol{\Sigma}^{\star}\right)_{\mathcal{S}}\|_F \geq \beta\sqrt{\frac{sK}{N}}\right) \leq \frac{\mathrm{E}\left(\|\nabla\mathcal{H}\left(\boldsymbol{\Sigma}^{\star}\right)_{\mathcal{S}}\|_F\right)}{\beta\sqrt{\frac{sK}{N}}} \leq \frac{\sqrt{\frac{2sK}{N}}}{\beta\sqrt{\frac{sK}{N}}} \leq \frac{\sqrt{2}}{\beta}.$$

Then since $\tau \leq c_\tau \left\|\left((\boldsymbol{\Sigma}^{\star})^{-1}\right)_{\mathcal{S}}\right\|_F^{-1}\sqrt{\frac{s}{N}}$, we have

$$\|(\nabla\left(\mathcal{H}\left(\boldsymbol{\Sigma}^{\star}\right) - \tau\log\det\left(\boldsymbol{\Sigma}^{\star}\right)\right))_{\mathcal{S}}\|_F \leq \left(\beta\sqrt{K} + c_\tau\right)\sqrt{\frac{s}{N}}. \tag{72}$$

with probability at least $1 - \frac{\sqrt{2}}{\beta}$. Combining equation 71 and equation 72, we have

$$\sum_{(k,l)\in\mathcal{S}} |(\nabla\mathcal{F}\left(\boldsymbol{\Sigma}^{\star}\right) + \boldsymbol{\Psi})_{kl}|\left|\left(\boldsymbol{\Sigma}^{\star} - \widehat{\boldsymbol{\Sigma}}\right)_{kl}\right| \leq \left(\beta\sqrt{K} + c_\tau\right)\sqrt{\frac{s}{N}}\left\|\boldsymbol{\Sigma}^{\star} - \widehat{\boldsymbol{\Sigma}}\right\|_F. \tag{73}$$

Finally, combining equation 69, equation 70, and equation 73, we have the desired result

$$\left\|\widehat{\boldsymbol{\Sigma}} - \boldsymbol{\Sigma}^{\star}\right\|_F \leq \frac{\beta\sqrt{K} + c_\tau}{\mu_0 - L_q}\sqrt{\frac{s}{N}}$$

with high probability. Specifically, according to union bound of the probability of equation 72 and the probabilities in Propositions 10 and 11, the final probability is $1 - \left(\frac{2}{d} + \frac{2}{d^{2(c_N-1)}} + \frac{\sqrt{2}}{\beta}\right)$.

### C.4 PROOF OF COROLLARY 1

According to Theorem 5, with high probability we have

$$\left\|\widehat{\boldsymbol{\Sigma}} - \boldsymbol{\Sigma}^{\star}\right\|_{\max} \leq \left\|\widehat{\boldsymbol{\Sigma}} - \boldsymbol{\Sigma}^{\star}\right\|_F < \frac{a}{2}.$$

Then since $T = \max\left\{0, \left\lceil 2\log\left(c_h/\sqrt{C_1}\right)/\log\left(1 - 1/(C_2\kappa)\right)\right\rceil\right\}$ and Theorem 1, we have

$$\left\|\boldsymbol{\Sigma}^{(t)} - \widehat{\boldsymbol{\Sigma}}\right\|_{\max} \leq \left\|\boldsymbol{\Sigma}^{(t)} - \widehat{\boldsymbol{\Sigma}}\right\|_F \leq \sqrt{C_1\left(1 - \frac{1}{C_2\kappa}\right)^T} < \frac{a}{2} - \left\|\widehat{\boldsymbol{\Sigma}} - \boldsymbol{\Sigma}^{\star}\right\|_F \leq \frac{a}{2} - \left\|\widehat{\boldsymbol{\Sigma}} - \boldsymbol{\Sigma}^{\star}\right\|_{\max},$$

and hence with high probability

$$\left\|\boldsymbol{\Sigma}^{(t)} - \boldsymbol{\Sigma}^{\star}\right\|_{\max} \leq \left\|\boldsymbol{\Sigma}^{(t)} - \widehat{\boldsymbol{\Sigma}}\right\|_{\max} + \left\|\widehat{\boldsymbol{\Sigma}} - \boldsymbol{\Sigma}^{\star}\right\|_{\max} \leq \frac{a}{2}.$$

Then recall Proposition 10, we have

$$\left\langle\nabla\mathcal{H}\left(\boldsymbol{\Sigma}^{(t)}\right) - \nabla\mathcal{H}\left(\widehat{\boldsymbol{\Sigma}}\right), \boldsymbol{\Sigma}^{(t)} - \widehat{\boldsymbol{\Sigma}}\right\rangle \geq \frac{\mu_0}{2}\left\|\boldsymbol{\Sigma}^{(t)} - \widehat{\boldsymbol{\Sigma}}\right\|_F^2$$

with high probability, and hence

$$\mathcal{F}\left(\widehat{\boldsymbol{\Sigma}}\right) - \mathcal{F}\left(\boldsymbol{\Sigma}^{(t)}\right) - \left\langle\mathcal{F}\left(\boldsymbol{\Sigma}^{(t)}\right), \widehat{\boldsymbol{\Sigma}} - \boldsymbol{\Sigma}^{(t)}\right\rangle \geq \frac{\mu_0 - L_q}{2}\left\|\boldsymbol{\Sigma}^{(t)} - \widehat{\boldsymbol{\Sigma}}\right\|_F^2. \tag{74}$$

Due to Theorem 1, we have

$$\left\|\boldsymbol{\Sigma}^{(t+1)} - \widehat{\boldsymbol{\Sigma}}\right\|_F \leq \left\|\boldsymbol{\Sigma}^{(t)} - \widehat{\boldsymbol{\Sigma}}\right\|_F < \frac{a}{2} - \left\|\widehat{\boldsymbol{\Sigma}} - \boldsymbol{\Sigma}^{\star}\right\|_F,$$

and hence

$$\left\|\boldsymbol{\Sigma}^{(t+1)} - \boldsymbol{\Sigma}^{\star}\right\|_F \leq \frac{a}{2}.$$

According to Weyl's inequality, we have

$$\lambda_{\min}\left(\boldsymbol{\Sigma}^{(t+1)}\right) \geq \lambda_{\min}\left(\boldsymbol{\Sigma}^{\star}\right) - \left\|\boldsymbol{\Sigma}^{(t+1)} - \boldsymbol{\Sigma}^{\star}\right\|_2 \geq \lambda_{\min}\left(\boldsymbol{\Sigma}^{\star}\right) - \left\|\boldsymbol{\Sigma}^{(t+1)} - \boldsymbol{\Sigma}^{\star}\right\|_F \geq \lambda_{\min}\left(\boldsymbol{\Sigma}^{\star}\right) - \frac{a}{2}$$

and

$$\lambda_{\min}\left(\boldsymbol{\Sigma}^{(t)}\right) \geq \lambda_{\min}\left(\boldsymbol{\Sigma}^{\star}\right) - \frac{a}{2}.$$

Therefore, when $\lambda_{\min}\left(\boldsymbol{\Sigma}^{(t)}\right) - \frac{a}{2} > 0$, we have

$$\mathcal{F}\left(\boldsymbol{\Sigma}^{(t+1)}\right) - \mathcal{F}\left(\boldsymbol{\Sigma}^{(t)}\right) - \left\langle \mathcal{F}\left(\boldsymbol{\Sigma}^{(t)}\right), \boldsymbol{\Sigma}^{(t+1)} - \boldsymbol{\Sigma}^{(t)}\right\rangle \leq \frac{1 + \frac{\tau}{(\lambda_{\min}(\boldsymbol{\Sigma}^{\star}) - a/2)^2}}{2}\left\|\boldsymbol{\Sigma}^{(t)} - \boldsymbol{\Sigma}^{(t+1)}\right\|_F^2.$$
(75)

Substituting equation 74 and equation 75 into Section C.1.3, we obtain the desired result.

When $\lambda_{\min}\left(\boldsymbol{\Sigma}^{(t)}\right) - \frac{a}{2} \leq 0$, according to equation 64, we have

$$\left\|\widehat{\boldsymbol{\Sigma}} - \boldsymbol{\Sigma}^{\star}\right\|_2 \leq \left\|\widehat{\boldsymbol{\Sigma}} - \boldsymbol{\Sigma}^{\star}\right\|_1 \leq 12\left\|\widehat{\boldsymbol{\Sigma}} - \boldsymbol{\Sigma}^{\star}\right\|_F.$$

Then due to $C_s\sqrt{s/N} < \lambda_{\min}\left(\boldsymbol{\Sigma}^{\star}\right)/(12c_r)$ and Theorem 3, we have

$$\left\|\widehat{\boldsymbol{\Sigma}} - \boldsymbol{\Sigma}^{\star}\right\|_2 \leq 12C_s\sqrt{\frac{s}{N}} < \frac{\lambda_{\min}\left(\boldsymbol{\Sigma}^{\star}\right)}{12c_r}.$$

Then recall Weyl's inequality, we have

$$\lambda_{\min}\left(\widehat{\boldsymbol{\Sigma}}\right) \geq \lambda_{\min}\left(\boldsymbol{\Sigma}^{\star}\right) - \left\|\widehat{\boldsymbol{\Sigma}} - \boldsymbol{\Sigma}^{\star}\right\|_2 > \frac{12c_r - 1}{12c_r}\lambda_{\min}\left(\boldsymbol{\Sigma}^{\star}\right).$$

Theorem 3, $T = \max\left\{0, \left\lceil 2\log\left(c_h/\sqrt{C_1}\right)/\log\left(1 - 1/(C_2\kappa)\right)\right\rceil\right\}$, and $c_h = \lambda_{\min}\left(\widehat{\boldsymbol{\Sigma}}\right) - (12c_r - 1)\lambda_{\min}\left(\boldsymbol{\Sigma}^{\star}\right)/12c_r$, we have

$$\lambda_{\min}\left(\boldsymbol{\Sigma}^{(t)}\right) \geq \lambda_{\min}\left(\widehat{\boldsymbol{\Sigma}}\right) - \left\|\boldsymbol{\Sigma}^{(t)} - \widehat{\boldsymbol{\Sigma}}\right\|_2 \geq \lambda_{\min}\left(\widehat{\boldsymbol{\Sigma}}\right) - \left\|\boldsymbol{\Sigma}^{(t)} - \widehat{\boldsymbol{\Sigma}}\right\|_F \geq \frac{12c_r - 1}{12c_r}\lambda_{\min}\left(\boldsymbol{\Sigma}^{\star}\right).$$

Similarly, we have

$$\lambda_{\min}\left(\boldsymbol{\Sigma}^{(t+1)}\right) \geq \frac{12c_r - 1}{12c_r}\lambda_{\min}\left(\boldsymbol{\Sigma}^{\star}\right),$$

and hence

$$\mathcal{F}\left(\boldsymbol{\Sigma}^{(t+1)}\right) - \mathcal{F}\left(\boldsymbol{\Sigma}^{(t)}\right) - \left\langle \mathcal{F}\left(\boldsymbol{\Sigma}^{(t)}\right), \boldsymbol{\Sigma}^{(t+1)} - \boldsymbol{\Sigma}^{(t)}\right\rangle \leq \frac{1 + \frac{144c_r^2\tau}{(12c_r - 1)^2\lambda_{\min}^2(\boldsymbol{\Sigma}^{\star})}}{2}\left\|\boldsymbol{\Sigma}^{(t)} - \boldsymbol{\Sigma}^{(t+1)}\right\|_F^2.$$
(76)

Substituting equation 74 and equation 76 into Section C.1.3, we obtain the desired result.

## C.5 PROOF OF COROLLARY 2

We first show that $\sum_{i=1}^{m}\left\|\boldsymbol{\Sigma}_i^{\left(t+\frac{1}{2}\right)} - \widehat{\boldsymbol{\Sigma}}\right\|_F^2$ converges linearly as well.

**Corollary 3.** *Suppose Assumptions 1, 2, and 3 hold, and all conditions in Theorem 2 are satisfied. Then we have*

$$\sum_{i=1}^{m}\left\|\boldsymbol{\Sigma}_i^{\left(t+\frac{1}{2}\right)} - \widehat{\boldsymbol{\Sigma}}\right\|_F^2 \leq C_1''\left(1 - \frac{1}{C_2'\kappa}\right)^t,$$
(77)

*where*

$$C_1'' = \frac{\left(\frac{16L\sqrt{m}}{\gamma\epsilon}\frac{\rho}{\underline{z} - \rho}\frac{\rho^2}{\underline{z} - \rho^2} + \frac{2L\sqrt{m}}{\gamma\epsilon}\frac{\rho}{\underline{z} - \rho}\right)\left\|\mathbf{E}_{\boldsymbol{\Sigma}}^{(0)}\right\|^2 + \frac{\sqrt{m}}{\gamma\epsilon}\frac{\rho}{\underline{z} - \rho}\left\|\mathbf{E}_{\mathbf{Y}}^{(0)}\right\|_F^2}{1 - \frac{6L\epsilon\sqrt{m}\rho}{\gamma(1-\rho)} - \left(1 - \frac{\mu}{\gamma}\right)\frac{\theta}{\underline{z} - (1-\theta)}}$$

$$+ \frac{\left(1 - \frac{\mu}{\gamma}\right)\frac{1}{\underline{z} - (1-\theta)}\sum_{i=1}^{m}\left\|\tilde{\boldsymbol{\Sigma}}_i^{(0)} - \widehat{\boldsymbol{\Sigma}}\right\|_F^2}{1 - \frac{6L\epsilon\sqrt{m}\rho}{\gamma(1-\rho)} - \left(1 - \frac{\mu}{\gamma}\right)\frac{\theta}{\underline{z} - (1-\theta)}} + \sum_{i=1}^{m}\left\|\boldsymbol{\Sigma}_i^{\left(\frac{1}{2}\right)} - \widehat{\boldsymbol{\Sigma}}\right\|_F^2.$$

*Proof.* Since equation 54 and the fact that $\left\|\mathbf{\Sigma}_i^{(t)} - \widehat{\mathbf{\Sigma}}\right\|_F^2 \leq \left\|\tilde{\mathbf{\Sigma}}_i^{(t-1)} - \widehat{\mathbf{\Sigma}}\right\|_F^2$, we have

$$\sum_{i=1}^m \left\|\tilde{\mathbf{\Sigma}}_i^{(t)} - \widehat{\mathbf{\Sigma}}\right\|_F^2 \leq \theta \sum_{i=1}^m \left\|\mathbf{\Sigma}_i^{\left(t+\frac{1}{2}\right)} - \widehat{\mathbf{\Sigma}}\right\|^2 + (1-\theta) \sum_{i=1}^m \left\|\tilde{\mathbf{\Sigma}}_i^{(t-1)} - \widehat{\mathbf{\Sigma}}\right\|^2. \tag{78}$$

Multiplying $z^{-t}$ on both sides, summing from 1 to $T$, and utilizing Lemma 3, we have

$$V^{(T)}(z) \leq \frac{\theta}{z-(1-\theta)} \sum_{t=1}^T \sum_{i=1}^m \left\|\mathbf{\Sigma}_i^{\left(t+\frac{1}{2}\right)} - \widehat{\mathbf{\Sigma}}\right\|^2 z^{-t} + \frac{1}{z-(1-\theta)} \sum_{i=1}^m \left\|\tilde{\mathbf{\Sigma}}_i^{(0)} - \widehat{\mathbf{\Sigma}}\right\|_F^2. \tag{79}$$

Define

$$W^{(T)}(z) = \sum_{t=1}^T \sum_{i=1}^m \left\|\mathbf{\Sigma}_i^{\left(t-\frac{1}{2}\right)} - \widehat{\mathbf{\Sigma}}\right\|_F^2 z^{-t}.$$

Combining equation 53, equation 79, and the fact that $\left\|\mathbf{\Sigma}_i^{(t)} - \widehat{\mathbf{\Sigma}}\right\|_F^2 \leq \left\|\tilde{\mathbf{\Sigma}}_i^{(t-1)} - \widehat{\mathbf{\Sigma}}\right\|_F^2$, multiplying $z^{-t}$ on both sides, summing from 1 to $T$, and utilizing Lemma 3, we have

$$\left(1 - \frac{6L\epsilon\sqrt{m}\rho}{\gamma(1-\rho)} - \left(1-\frac{\mu}{\gamma}\right)\frac{\theta}{z-(1-\theta)}\right)\left(zW^{(T)}(z) - \sum_{i=1}^m \left\|\mathbf{\Sigma}_i^{\left(\frac{1}{2}\right)} - \widehat{\mathbf{\Sigma}}\right\|_F^2\right)$$

$$\leq \left(1-\frac{\mu}{\gamma}\right)\frac{1}{z-(1-\theta)} \sum_{i=1}^m \left\|\tilde{\mathbf{\Sigma}}_i^{(0)} - \widehat{\mathbf{\Sigma}}\right\|_F^2 - \left(1-\frac{L}{\gamma}\right)D^{(T)}(z)$$

$$+ \frac{4L\sqrt{m}}{\gamma\epsilon}\frac{\rho}{z-\rho}D^{(T)}(z) + \frac{8L\sqrt{m}}{\gamma\epsilon}\frac{\rho}{z-\rho}E^{(T)}(z)$$

$$+ \frac{\sqrt{m}}{\gamma\epsilon}\frac{\rho}{z-\rho}\left\|\mathbf{E}_\mathbf{Y}^{(0)}\right\|_F^2 + \frac{2L\sqrt{m}}{\gamma\epsilon}\frac{\rho}{z-\rho}\left\|\mathbf{E}_\mathbf{\Sigma}^{(0)}\right\|_F^2. \tag{80}$$

Then following the proof of Theorem 2, we have the desired result. $\qquad\square$

Corollary 3 shows that the convergence rate of $\sum_{i=1}^m \left\|\mathbf{\Sigma}_i^{\left(t+\frac{1}{2}\right)} - \widehat{\mathbf{\Sigma}}\right\|_F^2$ differs from that of $\sum_{i=1}^m \left\|\mathbf{\Sigma}_i^{(t)} - \widehat{\mathbf{\Sigma}}\right\|_F^2$ with only a constant factor. Therefore, defining $C_3'' = \max\{C_1', C_3''\}$ and following similar steps to Section C.4, we can prove Corollary 2.

# D DISCUSSION

**Zero-mean assumption of x** In the previous discussion, we assumed for simplicity that the random vector $\mathbf{x}$ is zero-mean. However, in many practical applications, the mean is unknown. In scenarios where the mean is unknown, its effect can be removed by constructing a new sample using the pairwise difference of observations within each local subsample $\mathcal{J}_i$ (Maronna et al., 2019). The proposed algorithms and theoretical guarantees in this paper can be readily extended to this setting with only minor modifications.

**Supporting recovery** Table 1 in Section 7 demonstrates that our method achieves both the lowest NMSE and the highest F1-score among all compared approaches. This suggests that it not only provides excellent estimation accuracy in terms of the Frobenius norm but also exhibits strong support recovery capability. Due to space constraints, we have focused on establishing statistical convergence guarantees under the Frobenius norm in Theorem 3. As part of future work, we plan to develop formal support recovery guarantees to further characterize the theoretical properties of our estimator.

**High-dimensional covariance matrix estimation when features are distributed** This paper focuses on the sample-partitioned setting. However, in many practical applications, feature variables are distributed across different agents (Hu et al., 2019; Liu et al., 2022). As a direction for future work, we plan to extend our method to accommodate such feature-distributed scenarios.

**Parameter selection** The proposed method involves six parameters, which may introduce tuning overhead in practical applications. Specifically, $a$ controls robustness to heavy tails and outliers, $\tau$ ensures positive definiteness, $b$ adjusts the bias correction of the non-convex penalty, $\lambda$ determines the sparsity level, and $\gamma$ and $\theta$ govern the convergence behavior. In our experiments, the performance is largely insensitive to $\tau$, and setting it to a small constant is typically sufficient. The parameters $\gamma$ and $\theta$ can be selected empirically, as described in the main text, or through simple increasing/decreasing schedules. For $b$, a commonly recommended choice is $b = 3.7$ when using SCAD (Fan & Li, 2001). Therefore, in practice, the most critical parameters to tune are $a$ and $\lambda$. In future work, we aim to develop a more user-friendly robust sparse covariance estimator with fewer tuning parameters.

