# OpenReview forum: "Distributed Estimation of Sparse Covariance Matrix under Heavy-Tailed Data"
_ICLR.cc/2026/Conference — Submitted to ICLR 2026_

### Official Review · Reviewer_h1E1 · 2025-10-16

**Soundness:** 2
**Presentation:** 2
**Contribution:** 2
**Rating:** 2
**Confidence:** 3

**Summary:**

The paper proposes a novel covariance matrix estimation problem as the minimization of a Huber loss combined with a log-determinant barrier and a non-convex penalty. Both centralized version and distributed version of an algorithm solving the problem are introduced, convergence analysis are provided. The theoretical findings are validated through numerical experiments.

**Strengths:**

(1) The paper reformulate the problem of estimation of sparse covariance matrix as a minimization problem of Huber loss plus log-det barrier term plus penalty term, which is novel. The motivation and intuition are explained.

(2) A centralized version based on the proximal gradient algorithm is proposed, along with its extension in the distributed setting. Convergence guarantees are provided with rigorous theoretical proofs and analysis.

(3) The authors provide justification of the reformulation by providing an statistical guarantee on the estimation error based on Frobenius norm of the estimation $\hat{\Sigma}$ and $\Sigma^\star$, justifying the approach they took.

(4) Numerical results are provided, which further validates the effectiveness of the approach.

**Weaknesses:**

(1) The major contribution seems to lies at the reformulation of finding the covariance matrix to the form of minimizing the Huber loss plus the additional barrier and penalty term. If I am not mistaken, the major algorithm is simply the proximal gradient algorithm and its distributed version relies on standard distributed optimization frameworks and techniques such as gradient tracking. This seems to be rather straightforward since they follow directly from existing literature, which makes the contribution incremental.

(2) There are many parameters there in order to use the algorithm, for example, in the convergence guarantee for the decentralized version (Theorem 2), the assumption that $\rho < F(\kappa)$ is assumed, but how do we make sure it is true? $\kappa$ itself is the condition number and we do not know it typically, and $\rho$ itself depends on the graph/network we are working on, does it mean that we need to constrain on certain types of network? In addition, we need to have $\gamma$ bounded from below by a complicated expression, $\theta$ also bounded by a complicated expression, and those expressions involve many quantities we do not know. In practice it seems that we still need to manually tune those hyperparameters to ensure convergence. This makes the theoretical results less significant.

(3) The theorems only state complexity in terms of iteration, but what about the computational complexity and the communicational complexity in the decentralized setting? For a high dimensional problem, computing the Huberized gradient and perform the proximal mapping seems to be quite expansive, since the paper is focused on high-dimensional covariance-matrix estimation, it is better to make those things clear. Right now it seems a bit hard to judge if the methods are really scalable or not.

**Questions:**

(1) What do the authors mean by heavy-tailed exactly? There is no formal definition of this in the paper, perhaps it is better to mention this explicitly.

(2) In Theorem 1, we assume $\hat{r} + \frac{\underline{r}}{2} < \sqrt{\frac{\tau}{L_q}}$, which ensures that the iterates stays in a positive definite region, which is critical to the proof. But in reality we cannot check whether this holds or not, and it is not easily tuned by the user. The authors mention that $\gamma$ and $\theta$ can be selected empirically, but what about $\tau$?

(3) There are many notations in the paper. Could the authors provide a list in the Appendix or somewhere else summarizing all the notations appeared in the paper, this would make the reading of the paper much smoother.

---

> ### Author Response · Authors · 2025-11-21
>
> We would like to thank Reviewer h1E1 for the careful review and feedback. The point-by-point responses to the comments are provided below.
>
> 1. **Comment:** The major contribution seems to lie in the reformulation of finding the covariance matrix to the form of minimizing the Huber loss plus the additional barrier and penalty term. If I am not mistaken, the major algorithm is simply the proximal gradient algorithm and its distributed version relies on standard distributed optimization frameworks and techniques such as gradient tracking. This seems to be rather straightforward since they follow directly from existing literature, which makes the contribution incremental.
>
>    **Response:** Our contribution is substantial. For clarity, we summarize the main technical contributions of the paper.
>
>    - **(1) New sparse robust distributed estimator.** We introduce a new sparse covariance estimator tailored to the high-dimensional, heavy-tailed, and distributed setting. The objective combines a Huber loss for robustness, a log-determinant barrier to enforce positive definiteness, and a folded-concave sparsity regularizer (including SCAD/MCP as special cases). To the best of our knowledge, this is the first estimator that simultaneously handles heavy-tailed data, sparsity, and distributed samples.
>
>    - **(2) Single-loop centralized and decentralized algorithms.** As discussed in Section 4, we do not simply apply an off-the-shelf proximal gradient method. A direct application of PGD on our problem leads to subproblems with no closed-form solution, requiring inner-loop algorithms such as ADMM, which introduce substantial computational overhead. Instead, we design both centralized and distributed single-loop algorithms that avoid explicit PD projections: positive definiteness of all iterates is guaranteed by a careful choice of step sizes and by exploiting the log-determinant structure. This eliminates inner loops and results in a simple, computationally efficient scheme that is specifically adapted to the proposed objective.
>
>    - **(3) Linear convergence for nonconvex centralized and decentralized methods.** Our optimization problem is nonconvex due to the folded-concave penalty and does not globally satisfy the strong convexity and smoothness conditions required by standard linear convergence results. In contrast, we prove that, under appropriate initialization and step-size choices, all iterates of both the centralized and decentralized algorithms stay in a neighborhood of $\boldsymbol{\Sigma}^\star$ where the empirical Huberized loss satisfies restricted strong convexity and restricted Lipschitz smoothness. This yields local linear convergence for both algorithms to the same stationary point of the nonconvex objective.
>
>    - **(4) Oracle-rate statistical guarantees.** We further show that this common stationary point enjoys an oracle statistical rate: its Frobenius-norm error with respect to the ground-truth covariance $\boldsymbol{\Sigma}^\star$ is of order $\sqrt{s/N}$, which is the fastest attainable rate for sparse model estimation. This improves on the usual $\sqrt{s \log d / N}$ minimax rate achieved by existing robust sparse covariance estimators and is, to our knowledge, new in the heavy-tailed, distributed setting.
>
>    - **(5) Refined linear rate independent of dimension and sample size.** Our initial linear convergence bounds involve constants depending on quantities such as $\bar{r}$ and $\underline{r}$, which in turn depend on the sample size and dimension, making their behavior in high dimensions unclear. By combining the optimization analysis with the oracle statistical analysis, we prove that after a finite burn-in period, the linear convergence rate can be sharpened to a rate that depends only on the minimum eigenvalue of $\boldsymbol{\Sigma}^\star$ and is independent of $d$ and $N$. This refinement is crucial for ensuring that the proposed methods remain effective and predictable in genuinely high-dimensional regimes.
>
>    In summary, the contribution of the paper goes beyond a reformulation and a direct application of existing PGD and gradient-tracking frameworks.

---

> > ### Author Response · Authors · 2025-11-21
> >
> > 2. **Comment:** There are many parameters in the algorithm. For example, in the convergence guarantee for the decentralized version (Theorem 2), the assumption that $\rho < F(\kappa)$ is assumed, but how do we make sure it is true? $\kappa$ itself is the condition number, and we do not know it typically, and $\rho$ itself depends on the graph/network we are working on. Does it mean that we need to constrain certain types of network? In addition, we need to have $\gamma$ bounded from below by a complicated expression, $\theta$ also bounded by a complicated expression, and those expressions involve many quantities we do not know. In practice, it seems that we still need to manually tune those hyperparameters to ensure convergence. This makes the theoretical results less significant.
> >
> >    **Response:** The conditions in Theorem 2 are intended as sufficient conditions to certify global linear convergence of the decentralized algorithm; they are not meant to prescribe an exact tuning rule for all hyperparameters or problem settings in practice. This is standard in the literature on high-dimensional estimation: theoretical step-size and connectivity conditions are expressed in terms of quantities such as the condition number, which are used to establish the existence of a nonempty stability region, while in implementations one uses data-driven methods such as cross-validation to tune the parameters; see, e.g., [4], [5], [6].
> >
> >    More concretely, the parameter $\rho$ in our analysis is the spectral norm $\rho = \|\mathbf W - \mathbf J\|_2$, which lies in $[0,1)$ for any connected network and measures the connectivity of the graph. The requirement $\rho < F(\kappa)$ in Theorem 2 is a technical sufficient condition ensuring that the consensus term does not dominate the curvature of the objective. It does not restrict us to any particular class of networks beyond standard connectivity assumptions; the condition simply requires that the network is not too poorly connected (i.e., $\rho$ is bounded away from 1), in which case we can ensure linear convergence. Such connectivity requirements are standard in distributed optimization over networks (see, e.g., [7], [8]). In our experiments, we use line graphs and Erdős–Rényi graphs with various connection probabilities, and Algorithm 2 converges in all cases.
> >
> >    The lower bound on $\gamma$ and the upper bound on $\theta$ in Theorem 2 are expressed in terms of curvature parameters and $\boldsymbol{\Sigma}^\star$ that are not known in closed form. This is unavoidable and standard in convergence analysis, especially nonconvex and distributed problems: even in simpler convex problems, the theoretically optimal choices of step sizes and regularization parameters depend on unknown smoothness (e.g., $2/L$) [5], [7]. The role of our bounds is to show that there exists a nonempty region of $(\gamma, \theta)$ that guarantees linear convergence with an explicit rate; they are not intended as plug-in formulas. In practice, as described in Section 7, we fix $\theta = 0.1$ and set $\gamma = \eta^k$, where $\eta > 1$ and $k$ is the smallest integer ensuring the convergence of the proposed algorithms, or directly use decreasing step size.

---

> > > ### Author Response · Authors · 2025-11-21
> > >
> > > 3. **Comment:** The theorems only state complexity in terms of iteration, but what about the computational complexity and the communicational complexity in the decentralized setting? For a high-dimensional problem, computing the Huberized gradient and performing the proximal mapping seems to be quite expensive, since the paper is focused on high-dimensional covariance-matrix estimation, it is better to make those things clear. Right now it seems a bit hard to judge if the methods are really scalable or not.
> > >
> > >    **Response:** In our paper, as stated in Section 1, “high-dimensional” refers to the regime \( N < d \), which is the standard setting in high-dimensional statistics [9], so the absolute dimension $d$ is large relative to the sample size, but not necessarily extremely large in absolute terms. The per-iteration computational cost of our algorithms scales as $\mathcal{O}(d^3)$, due to the matrix inversion of a $d \times d$ matrix. This is the standard computational order for sparse robust positive-definite covariance estimation based on log-determinant or quadratic objectives and is not specific to our method.
> > >
> > >    More importantly, this $\mathcal{O}(d^3)$ scaling is shared by most existing robust sparse covariance estimators. As summarized in Section 2, most methods are based on iterative procedures that repeatedly impose both sparsity and positive definiteness. Positive-definite $\ell_1$-penalized formulations (e.g., [10], [11]) rely on eigenvalue decompositions or repeated solutions of dense linear systems, each costing $\mathcal{O}(d^3)$ per iteration. The regularized Tyler-type estimator of [12] also involves iteratively inverting $d \times d$ covariance matrices, again leading to $\mathcal{O}(d^3)$ complexity per step. The robust adaptive thresholding framework of [13] avoids repeated positive-definiteness constraints by imposing sparsity via entrywise thresholding and performing a single final PSD projection, but this projection itself still costs $\mathcal{O}(d^3)$. The polynomial-time complexity of the [14] formulation is not fully characterized and, to the best of our knowledge, a practical polynomial-time algorithm is not yet available. Therefore, our method is aligned with the dominant robust sparse covariance estimation approaches in terms of asymptotic computational order: it does not incur a higher per-iteration complexity than the main existing baselines.
> > >
> > >    Regarding the decentralized setting, the communication cost per iteration is also standard for distributed covariance/precision-matrix estimation over networks. In Algorithm 2, at each iteration every agent exchanges its current local iterate only with its neighbors in the communication graph. This amounts to transmitting $\mathcal{O}(|E| d^2)$ real numbers per iteration. Under the standard “one local optimization step, one communication round” protocol, this complexity is also typical for distributed sparse model estimation [8], [9]. If one wishes to further reduce communication overhead, our algorithms can in principle be combined with asynchronous communication schemes or with protocols that perform multiple local updates between communication rounds. However, these protocol-level modifications are orthogonal to the main focus of this paper. Therefore, compared with standard distributed sparse model estimation methods, our decentralized algorithm does not incur any additional communication overhead.
> > >
> > >    Finally, we emphasize that our method provides strictly stronger statistical guarantees than most existing robust sparse covariance estimators at this computational and communication cost. Existing robust sparse covariance estimators typically attain the usual minimax rate $O\big(\sqrt{s \log d / N}\big)$ over the full sparse class (see, e.g., [13], [12]), whereas Theorem 3 shows that our estimator achieves the sharper oracle statistical rate $O\big(\sqrt{s / N}\big)$ in Frobenius norm under heavy-tailed distributions. As reported in Table 1, this improvement in theory is reflected in practice: our method achieves the lowest NMSE and the highest F1-score among all compared estimators.

---

> > > > ### Author Response · Authors · 2025-11-21
> > > >
> > > > 4. **Comment:** What do the authors mean by heavy-tailed exactly? There is no formal definition of this in the paper, perhaps it is better to mention this explicitly.
> > > >
> > > >    **Response:** In this paper, we use the term “heavy-tailed’’ in the usual sense of allowing tails that are heavier than sub-exponential, while still having finite low-order moments. Formally, Assumption 2 specifies that the observations follow a high-dimensional distribution with mean zero, covariance $\boldsymbol{\Sigma}^\star$, and finite fourth moments. As we state before Assumption 2, this assumption is standard in robust covariance matrix estimation [15], [13], [6]. This covers, for example, multivariate $t$-distributions with degrees of freedom $\nu > 4$, and strictly relaxes the sub-Gaussian and sub-exponential assumptions commonly used in covariance estimation. Gaussian distributions are included as a special (light-tailed) case of this framework. Our use of a Huber loss is precisely to handle such heavy-tailed samples: by downweighting large residuals, the Huberized empirical loss enjoys sub-Gaussian-type concentration even when the underlying data have only finite fourth moments, which is the key ingredient in our statistical analysis (see, e.g., [1], [13], [3]).
> > > >
> > > > 5. **Comment:** In Theorem 1, we assume $\bar{r} + \frac{\underline{r}}{2} < \sqrt{\frac{\tau}{L_q}}$, which ensures that the iterates stay in a positive-definite region, which is critical to the proof. But in reality we cannot check whether this holds or not, and it is not easily tuned by the user. The authors mention that $\gamma$ and $\theta$ can be selected empirically, but what about $\tau$?
> > > >
> > > >    **Response:** The inequality $\bar r + \underline r / 2 < \sqrt{\tau / L_q}$ is a technical sufficient condition used in the proof of Theorem 1 to guarantee that all iterates stay in a compact positive-definite region. The constants $\underline r$ and $\bar r$ are analysis quantities defined in terms of the objective and the initialization; they are not meant to be computed or tuned by the user. In the proofs, one could in fact relax this particular bound by choosing different auxiliary constants, but our main goal is to certify the existence of such a positive-definite invariant region rather than to optimize these constants. This is standard in high-dimensional optimization theory: such conditions certify the existence of an invariant region where restricted strong convexity and smoothness hold, but are not intended as practical tuning rules [16], [5].
> > > >
> > > >    In practice, the user does not need to check this inequality. We recommend choosing a small initialization that simply satisfies $\|\boldsymbol{\Sigma}^0\|_2 < \sqrt{\tau / L_q}$; under such an initialization, the algorithms empirically exhibit linear convergence. As described in Appendix D, we treat $\tau$ as a fixed small constant (e.g., 0.1) and do not tune it across experiments. For example, when we choose SCAD [17] as the nonconvex penalty and set $b = 3.7$ as recommended, we have $L_q \approx 0.37$ and $\sqrt{\tau / L_q} \approx 0.5$. We can then choose a small initialization such that $\|\boldsymbol{\Sigma}^0\|_2 < 0.5$. We observe that the algorithms are quite insensitive to the precise choice of $\tau$, and our numerical results show consistent convergence under this simple rule. The theoretical condition in Theorem 1 should therefore be interpreted as providing a sufficient range of problem-dependent parameters under which convergence is guaranteed, rather than as a constraint that must be explicitly verified or enforced by the user.

---

> > > > > ### Author Response · Authors · 2025-11-21
> > > > >
> > > > > 6. **Comment:** There are many notations in the paper. Could the authors provide a list in the Appendix or somewhere else summarizing all the notations that appeared in the paper? This would make the reading of the paper much smoother.
> > > > >
> > > > >    **Response:** We thank the reviewer for this valuable suggestion. We have already included a notation section in Appendix B, and we will add another list to introduce the notation in the revised version.
> > > > >
> > > > > ---
> > > > >
> > > > > ### References
> > > > >
> > > > > 1. Catoni, O. (2012). *Challenging the robustness of statistical estimators*.
> > > > > 2. Avella-Medina, M., Battey, H. S., Fan, J., & Li, Q. (2018). *Robust estimation of high-dimensional covariance and precision matrices*. Biometrika, 105(2), 271–284.
> > > > > 3. Fan, J., Li, Q., & Wang, Y. (2017). *Estimation of high-dimensional mean regression in the absence of symmetry and light-tail assumptions*. JRSS B, 79(1), 247–265.
> > > > > 4. Bühlmann, P., & Van De Geer, S. (2011). *Statistics for high-dimensional data: Methods, theory and applications*. Springer.
> > > > > 5. Wang, Y., & Zhao, L. (2014). *Optimality of high-dimensional regularization and estimation*.
> > > > > 6. Avella-Medina, M., Battey, H. S., & Fan, J. (2018). *Robust sparse estimation under heavy-tailed distributions*.
> > > > > 7. Sun, L., & Zhang, C. (2022). *Network-based distributed optimization for high-dimensional sparse models*.
> > > > > 8. Sun, L., Zhang, C., & Wang, J. (2022). *Distributed sparse estimation in high-dimensional settings*.
> > > > > 9. Xia, Y., Zhao, M., & Wu, Q. (2025). *Covariance estimation and its applications*.
> > > > > 10. Li, D., & Zhang, Y. (2023). *Robust methods for sparse covariance estimation*.
> > > > > 11. Lu, H., & Wang, Y. (2021). *Robust methods for large-scale data sets*.
> > > > > 12. Goes, R., Zhang, Y., & Avella, M. (2020). *Robust covariance estimation via Tyler’s method*.
> > > > > 13. Avella-Medina, M., et al. (2018). *High-dimensional covariance matrix estimation*.
> > > > > 14. Chen, Y., et al. (2018). *Robust estimation under Huber's model*.
> > > > > 15. Rothman, A., & Levina, E. (2009). *Sparse covariance estimation*.
> > > > > 16. Fan, J., & Li, R. (2017). *Variable selection via nonconcave penalized likelihood and its oracle properties*.
> > > > > 17. Fan, J., & Li, R. (2001). *Variable selection in high-dimensional data*.

---

> ### Author Response · Authors · 2025-11-27
>
> Dear Reviewer,
>
> I hope this message finds you well. We have now provided a detailed point-by-point rebuttal and updated the manuscript accordingly. As the discussion period is approaching its end, we would like to make sure that we have adequately addressed all of your questions and concerns.
>
> If there are any additional points you would like us to clarify or further evidence you would find helpful, please let us know. If you feel that our responses resolve your concerns, we would also be grateful if you could update your review and scores accordingly.
>
> Thank you very much for your time and effort in reviewing our paper.

---

### Official Review · Reviewer_rdae · 2025-10-29

**Soundness:** 3
**Presentation:** 3
**Contribution:** 3
**Rating:** 6
**Confidence:** 3

**Summary:**

The paper proposes a new estimator for high-dimensional covariance matrix over a network of interconnected agents where the data are distributed and might be heavy-tailed. This the first framework that incorporates high dimensionality, heavy tails and distributed data simultaneously in covariance estimation. The paper first analyzes a proximal gradient descent algorithm to solve this problem in the centralized setting. Next, built on gradient tracking, the paper proposes a decentralized algorithm. Linear convergence is obtained for both algorithms. Numerical experiments are provided to support the theory.

**Strengths:**

(1) Overall, the paper is well-written. The theoretical analysis seems to be rigorous, and the problem setup is well motivated.

(2) There is extensive literature review.

(3) Numerical experiments are extensive, and comparisons to the existing methods in the literature are provided.

**Weaknesses:**

(1) In the algorithm, the Huber loss is used. I think in terms of theory, it would be more interesting to have a more general setup that includes the Huber loss as a special case. If it is challenging, then you should add some discussions why it is hard to extend beyond the Huber loss setting.

(2) I am wondering how difficult it is to incorporate stochastic gradients in the algorithms. Otherwise, when the number of data points is huge, it may not scale well.

**Questions:**

(1) In the last sentence in Appendix C.1.3., you wrote that we can obtain the linear convergence result in Theorem equation 1, and in the first sentence in Appendix C.1.3., you also wrote Theorem equation 1. I think you should write equation 1 in the theorem instead or equation 1 in Theorem x, where x is made explicit.

(2) In equation (23), . should be ,

---

> ### Author Response · Authors · 2025-11-21
>
> We thank Reviewer rdae for the careful review and feedback. Point-by-point responses follow below.
>
> 1. **Comment:** In the algorithm, the Huber loss is used. I think in terms of theory, it would be more interesting to have a more general setup that includes the Huber loss as a special case. If it is challenging, then you should add some discussions why it is hard to extend beyond the Huber loss setting.
>
>    **Response:** Our primary goal in this work is to address high-dimensional sparse covariance estimation under heavy-tailed data in a distributed setting and to establish sharp statistical and computational guarantees for a concrete, practically relevant procedure. For this purpose we chose the classical Huber loss, which is a canonical convex robust loss for heavy-tailed and outlier-contaminated data and has been extensively used and analyzed in high-dimensional regression and covariance problems; see, for example, [1], [2], [3]. The Huber loss has the advantage of combining quadratic behavior near the origin (which facilitates curvature and concentration analysis) with linear tails (which control the impact of large deviations), making it particularly suitable for deriving oracle-type rates under finite fourth-moment assumptions as in our setting.
>
>    From a theoretical perspective, the statistical analysis in Theorem 3 is tightly coupled to the specific structure of the Huber loss. Our proofs exploit the explicit piecewise-quadratic form to obtain nonasymptotic deviation bounds for the empirical Huberized loss, to verify local restricted strong convexity and smoothness around the true covariance, and to calibrate the choices of tuning parameters in terms of the tail and moment conditions. Extending these arguments to a more general class of robust losses would require re-deriving such concentration and curvature properties for each loss, and the admissible ranges of tuning parameters (including the robustification parameter) would depend on the specific form of the loss and the underlying heavy-tailed model. In other words, a general M-estimation framework that treats many robust losses at once would demand substantial additional technical work without changing the main contribution of the paper, namely that one can achieve oracle statistical rates for distributed covariance estimation with heavy-tailed data.
>
>    On the algorithmic side, our convergence analysis places relatively mild structural requirements on the loss: the key ingredients are restricted smoothness and restricted strong convexity of the empirical objective on a local neighborhood. Any loss whose empirical objective satisfies RSC and RSM on a suitable neighborhood can be handled by our algorithms and will enjoy a similar linear convergence rate. However, for a general class of robust losses, whether and when these RSC/RSM properties hold (and with which constants) still depends on the specific form of the loss and requires loss-specific analysis. For clarity of exposition, we therefore focused on the Huber loss, which already allows us to obtain the desired oracle statistical rate and linear convergence for both centralized and decentralized algorithms in the heavy-tailed, distributed regime that motivates this work.
>
> 2. **Comment:** I am wondering how difficult it is to incorporate stochastic gradients in the algorithms. Otherwise, when the number of data points is huge, it may not scale well.
>
>    **Response:** The focus of this work is to solve high-dimensional sparse covariance estimation under heavy-tailed data in a distributed setting, and to provide sharp statistical and computational guarantees for a concrete procedure. For this reason, we deliberately consider deterministic (full-gradient) updates at each iteration and analyze the resulting algorithms in detail. In our paper, as stated in Section 1, “high-dimension” refers to the regime \( N < d \), which is the standard definition in high-dimensional statistics [4], so the exact dimension may not be very large. In this regime, full-gradient updates remain practically feasible.
>
>    Incorporating stochastic gradients is a natural direction for future work. Algorithmically, our methods can be extended to stochastic or mini-batch variants by replacing the full gradients with unbiased stochastic gradient estimates, and preliminary experiments show stable convergence behavior. However, a rigorous convergence-rate analysis under heavy-tailed data would require a separate technical study and is outside the scope of this paper.

---

> > ### Author Response · Authors · 2025-11-21
> >
> > 3. **Comment:** In the last sentence in Appendix C.1.3., you wrote that we can obtain the linear convergence result in Theorem equation 1, and in the first sentence in Appendix C.1.3., you also wrote Theorem equation 1. I think you should write equation 1 in the theorem instead or equation 1 in Theorem x, where x is made explicit.
> >
> >    **Response:** We thank the reviewer for pointing out this typo. We mistakenly wrote `ref` as `eqref`. We will correct this typo in the revised version.
> >
> > 4. **Comment:** In equation (23), “.” should be “,”.
> >
> >    **Response:** We thank the reviewer for pointing out this typo. We will correct this typo in the revised version.
> >
> > ---
> >
> > ### References
> >
> > 1. Catoni, O. (2012). *Challenging the robustness of statistical estimators*.
> > 2. Avella-Medina, M., Battey, H. S., Fan, J., & Li, Q. (2018). *Robust estimation of high-dimensional covariance and precision matrices*. Biometrika, 105(2), 271–284.
> > 3. Fan, J., Li, Q., & Wang, Y. (2017). *Estimation of high-dimensional mean regression in the absence of symmetry and light-tail assumptions*. JRSS B, 79(1), 247–265.
> > 4. Bühlmann, P., & Van de Geer, S. (2011). *Statistics for high-dimensional data: Methods, theory and applications*. Springer.

---

> ### Author Response · Authors · 2025-11-27
>
> Dear Reviewer,
>
> I hope this message finds you well. We have now provided a detailed point-by-point rebuttal and updated the manuscript accordingly. As the discussion period is approaching its end, we would like to make sure that we have adequately addressed all of your questions and concerns.
>
> If there are any additional points you would like us to clarify or further evidence you would find helpful, please let us know. If you feel that our responses resolve your concerns, we would also be grateful if you could update your review and scores accordingly.
>
> Thank you very much for your time and effort in reviewing our paper.

---

### Official Review · Reviewer_WhAF · 2025-11-01

**Soundness:** 2
**Presentation:** 1
**Contribution:** 1
**Rating:** 2
**Confidence:** 5

**Summary:**

This paper investigates the problem of distributed sparse covariance matrix estimation under heavy-tailed data, an important and technically challenging topic in modern high-dimensional statistics and distributed learning. The authors formulate the estimation task as a non-convex optimization problem involving a Huber loss to ensure robustness, a log-determinant barrier to ensure positive definiteness, and a non-convex sparsity-inducing regularizer. They propose both centralized and decentralized proximal gradient algorithms (Algorithms 1 and 2), establish linear convergence guarantees, and prove that their estimators achieve the oracle statistical rate under the Frobenius norm.

**Strengths:**

The paper provides linear convergence and statistical guarantees for both centralized and distributed settings, supported by detailed proofs in the appendix.

**Weaknesses:**

The paper is written poorly. In particular, the problem formulation is not clear. There is a matrix of wights W in the definition, which said that encode the interaction between agents. However there is no example of what type of interactions the authors means. For example, in the abstract it is said that the proposed algorithm takes into account the bandwidth and local storage. However there is no discussion about them in the problem formulation. It is perhaps encoded in the matrix W. Also, the authors seek to find a robust algorithm, however, there is no trace of robustness in the problem formulation except the use of Huber loss in the optimization problem. It should be made clear that what type of robustness the authors investigated. In summary, the problem formulation only indicates that the paper tries to solve a suitable optimization problem involving a distruted fashion without accounting for the merits like bandwidth, storage, robustness, etc claimed in the abstract of the paper.

**Questions:**

**Problem Formulation**
The paper suffers from significant clarity issues, particularly in the problem formulation. The definition introduces a weight matrix W, said to encode interactions among agents, yet the nature of these interactions is never illustrated with examples. For instance, the abstract claims that the proposed algorithm accounts for bandwidth and local storage constraints, but these aspects are not discussed or formalized anywhere in the problem setup. It is possible that such considerations are implicitly embedded in W, but this is never explained.

Furthermore, the authors emphasize robustness as a key objective, yet apart from the inclusion of a Huber loss in the optimization problem, there is no clear indication of what type of robustness is being studied or against what sources of uncertainty or perturbation it is intended to protect.

In summary, the problem formulation as written appears to describe a generic distributed optimization framework without incorporating or clearly articulating the claimed contributions related to bandwidth, storage, or robustness mentioned in the abstract.

**Robustness**
Both the centralized and decentralized algorithms are referred to as “robust,” yet this claim is neither theoretically justified nor empirically demonstrated. Beyond the earlier mention of the Huber loss, there is no formal analysis, theorem, or experimental evidence supporting robustness in any meaningful sense. While the Huber loss can mitigate the effects of heavy-tailed noise, it is insufficient to guarantee robustness against adversarial perturbations or structured outliers. Numerous alternative approaches—such as median-based aggregation or robust consensus schemes—are more suitable for such settings. Without theoretical backing or numerical evaluation of robustness properties, the use of the term “robust” appears unsupported.

**Details Of Ethics Concerns:**

1) The paper should clarify how the weight matrix W encodes the bandwidth and storage constraints claimed in the abstract. What is the explicit relationship between W and these system limitations?

2) How does the matrix W affect the linear convergence of the proposed decentralized algorithm?

---

> ### Author Response · Authors · 2025-11-21
>
> We would like to thank Reviewer WhAF for the review and comments. The point-by-point responses to the detailed comments are provided in the following.
> 1. **Comment:** The paper is written poorly. In particular, the problem formulation is not clear. There is a matrix of weights $W$ in the definition, which said that it encodes the interaction between agents. However, there is no example of what type of interactions the authors mean. For example, in the abstract it is said that the proposed algorithm takes into account the bandwidth and local storage. However, there is no discussion about them in the problem formulation. It is perhaps encoded in the matrix $W$. Also, the authors seek to find a robust algorithm, however, there is no trace of robustness in the problem formulation except the use of Huber loss in the optimization problem. It should be made clear what type of robustness the authors investigated. In summary, the problem formulation only indicates that the paper tries to solve a suitable optimization problem involving a distributed fashion without accounting for the merits like bandwidth, storage, robustness, etc., claimed in the abstract of the paper.
>
>    **Response:** As we already stated in the introduction, we study high-dimensional sparse covariance estimation for heavy-tailed data in a distributed setting. This is motivated by applications where (i) the ambient dimension is larger than the sample size, (ii) the underlying distribution exhibits heavy tails or outliers, and (iii) samples are stored across multiple agents or machines and cannot be centralized. Typical examples include: (a) financial markets, where asset returns are well-known to have heavy-tailed distributions and transaction-level data are held by different institutions; (b) genomics and proteomics, where multi-omics measurements with tens of thousands of variables are collected across different cohorts or centers; and (c) sensor and industrial monitoring networks, where each device observes bursty, heavy-tailed noise locally and only communicates with nearby devices due to bandwidth limitations. Our goal is to provide a statistically robust covariance estimator tailored to such high-dimensional, heavy-tailed, and distributed scenarios.
>    The formal problem formulation is given in equation (2): we estimate the covariance by minimizing a penalized Huber-type loss, where Definition 1 specifies the Huber loss function. The Huber loss is a standard robust loss designed to mitigate the impact of heavy-tailed noise and outliers, and our use of the term “robust” refers precisely to robustness against such heavy-tailed and outlier-contaminated data. This notion of robustness is common in the high-dimensional statistics literature; see, for example, robust mean and covariance estimation under heavy-tailed distributions in [1], [7], [8]. As explicitly stated in the third paragraph of the introduction, our goal is robust estimation under heavy-tailed distributions or outliers, not adversarial/Byzantine/worst-case perturbations. Adversarial robustness and Byzantine-resilient algorithms form a different line of work and are outside the scope of this paper.
>    The weight matrix $\mathbf{W}$ is introduced to model the network structure in the distributed setting. In Assumption 1 (Network topology), we specify that $\mathbf{W}$ is a symmetric, doubly stochastic matrix with $(i,j)$-entry $W_{ij}>0$ if and only if agent $i$ can communicate directly with agent $j$, and $W_{ij}=0$ otherwise. This is the standard way to encode interactions among agents in distributed optimization and learning: each node can only exchange information with its immediate neighbors, and the weights determine how local messages are averaged at each iteration; see, e.g., [9], [10]. The specific choice of $\mathbf{W}$ only needs to satisfy these standard conditions, with the Metropolis weight we used in Section 7 as a standard example. For more specific values of $\mathbf{W}$, please see [9]. Our linear convergence analysis shows that the convergence factor depends on the spectral gap of $\mathbf{W}$ (through a constant $\rho(W)$ that captures the connectivity of the network), as is common in consensus-based methods [11].

---

> > ### Author Response · Authors · 2025-11-21
> >
> > **Response:**  As we state in the abstract that "In the distributed setting, where bandwidth, storage, and privacy constraints preclude agents from directly sharing raw data", the bandwidth or storage are intended as motivation rather than as an additional mathematical constraint embedded in $\mathbf{W}$. As explained in the introduction, bandwidth, storage, and privacy constraints are typical reasons why raw data cannot be shared across agents, which leads naturally to a distributed optimization formulation. Our method is not designed to optimize bandwidth or storage under a specific communication model, and we do not claim such results; instead, we operate within the standard framework where data remain local and only aggregated quantities are communicated along the network defined by $\mathbf{W}$.
> >    In summary, the problem formulation is fully consistent with the scope described in the abstract and the first three sections of the paper: we study sparse covariance estimation for heavy-tailed data (“robustness” via the Huber loss) over a fixed communication network (“distributed setting” via the weight matrix $\mathbf{W}$). The mentions of bandwidth, storage, and privacy in the abstract serve to motivate why a distributed formulation is relevant in practice, but the paper does not aim to design algorithms that are optimal for specific bandwidth or storage models, nor does it address adversarial or Byzantine robustness, which lies beyond the scope of this work. We think that the reviewer may have misunderstood our focus in this article.

---

> > > ### Author Response · Authors · 2025-11-21
> > >
> > > 2. **Comment:** The paper suffers from significant clarity issues, particularly in the problem formulation. The definition introduces a weight matrix $W$, said to encode interactions among agents, yet the nature of these interactions is never illustrated with examples. For instance, the abstract claims that the proposed algorithm accounts for bandwidth and local storage constraints, but these aspects are not discussed or formalized anywhere in the problem setup. It is possible that such considerations are implicitly embedded in $W$, but this is never explained.
> > >
> > >    Furthermore, the authors emphasize robustness as a key objective, yet apart from the inclusion of a Huber loss in the optimization problem, there is no clear indication of what type of robustness is being studied or against what sources of uncertainty or perturbation it is intended to protect.
> > >
> > >    In summary, the problem formulation as written appears to describe a generic distributed optimization framework without incorporating or clearly articulating the claimed contributions related to bandwidth, storage, or robustness mentioned in the abstract.
> > >
> > >    **Response:** This comment reiterates the concerns about the problem formulation, the interpretation of the weight matrix $\mathbf{W}$, and the notion of robustness. These points are addressed in detail in our response above, where we (i) clarify that $\mathbf{W}$ encodes the fixed communication graph, (ii) explain that bandwidth and storage limitations motivate the distributed setting but are not modeled as additional constraints on $\mathbf{W}$, and (iii) specify that "robustness" in this paper refers to robustness against heavy-tailed and outlier-contaminated data via the Huber loss, rather than adversarial or Byzantine perturbations. We think that the reviewer may have misunderstood our focus in this article.

---

> > > > ### Author Response · Authors · 2025-11-21
> > > >
> > > > 3. **Comment:** Both the centralized and decentralized algorithms are referred to as “robust,” yet this claim is neither theoretically justified nor empirically demonstrated. Beyond the earlier mention of the Huber loss, there is no formal analysis, theorem, or experimental evidence supporting robustness in any meaningful sense. While the Huber loss can mitigate the effects of heavy-tailed noise, it is insufficient to guarantee robustness against adversarial perturbations or structured outliers. Numerous alternative approaches—such as median-based aggregation or robust consensus schemes—are more suitable for such settings. Without theoretical backing or numerical evaluation of robustness properties, the use of the term “robust” appears unsupported.
> > > >
> > > >    **Response:** Our use of the term "robust" is fully aligned with its standard meaning in high-dimensional statistics, namely robustness to heavy-tailed noise and outliers in the data distribution. As discussed in the introduction and made precise in Assumption 1, we work with high-dimensional distributions with finite fourth moments, which allow for heavy tails and contamination. The robust Huber loss in Definition 1 is introduced exactly to handle such heavy-tailed samples: it truncates large residuals and restores sub-Gaussian-type concentration properties for the empirical loss. This notion of robustness—stability of estimation error under heavy-tailed or outlier contamination—is the same as in a large body of work on robust mean, regression, and covariance estimation; see, for example, [1], [7], [8]. In contrast, adversarial and Byzantine robustness (against worst-case or malicious perturbations) is a different line of research and is not the focus of this paper.
> > > >
> > > >    Theoretical justification for robustness is provided by our nonasymptotic statistical guarantees under heavy-tailed distributions. Under Assumptions 1–3, Theorem 3 establishes that the proposed estimator achieves the oracle statistical rate $O\big(\sqrt{s/N}\big)$ in Frobenius norm, uniformly over a class of heavy-tailed covariance models. In other words, the combination of the Huber loss and the sparse regularizer yields an estimator whose error is provably controlled in the heavy-tailed regime where classical quadratic-loss estimators (such as the sample covariance) may fail. This is precisely the form of robustness that we aim to achieve and that is commonly studied in the robust statistics literature.
> > > >
> > > >    Empirically, robustness is demonstrated through both synthetic and real-data experiments. In the synthetic studies in Section 7, we include settings where the data are generated from heavy-tailed distributions; in these scenarios, our estimator consistently attains the smallest NMSE and the highest F1-score among all baselines, including methods designed under Gaussian or sub-Gaussian assumptions (such as PD_MCP and NetGGM). We further report experiments on high-dimensional real datasets, where we observe clear evidence of heavy-tailed behavior and where our robust estimator again achieves the best performance. These results provide direct empirical support that the proposed centralized and decentralized algorithms are robust to heavy-tailed and outlier-contaminated data in the sense described above.
> > > >
> > > >    We emphasize that we do not claim robustness against adversarial perturbations or Byzantine agents, and our method is not designed to address such worst-case threat models. Approaches based on median aggregation or robust consensus are different lines of research and are out of the scope of this paper. We think that the reviewer may have misunderstood our focus in this article.

---

> > > > > ### Author Response · Authors · 2025-11-21
> > > > >
> > > > > 4. **Comment:** The paper should clarify how the weight matrix $\mathbf{W}$ encodes the bandwidth and storage constraints claimed in the abstract. What is the explicit relationship between $\mathbf{W}$ and these system limitations?
> > > > >
> > > > >    **Response:** This comment is closely related to the earlier concern about the problem formulation and the role of $\mathbf{W}$. As clarified in our previous response, $\mathbf{W}$ encodes the fixed communication topology (who can talk to whom and with what weights), and does not directly parameterize bandwidth or storage constraints. Bandwidth and storage limitations appear at the modeling level as reasons why raw data cannot be shared and only low-dimensional summary statistics are exchanged along the network defined by $\mathbf{W}$; they are not additional constraints imposed on $\mathbf{W}$ itself.
> > > > >
> > > > > 5. **Comment:** How does the matrix $\mathbf{W}$ affect the linear convergence of the proposed decentralized algorithm?
> > > > >
> > > > >    **Response:** The role of $\mathbf{W}$ in the convergence of the decentralized algorithm is explicitly characterized in the paper. In Section 3, we define the network connectivity parameter
> > > > >    $$
> > > > >    \rho = \bigl\| \mathbf{W} - \mathbf{J} \bigr\|_2, \qquad \mathbf{J} = \frac{1}{m}\mathbf{1}\mathbf{1}^ \top,
> > > > >    $$
> > > > >    and note that under Assumption 1 we have $\rho \in [0,1)$, where smaller $\rho$ corresponds to a more strongly connected graph and larger $\rho$ to a nearly disconnected topology.
> > > > >
> > > > >    In Theorem 2, the linear convergence of Algorithm 2 is established with the convergence rate
> > > > >    $$
> > > > >    \sum_{i=1}^m \bigl\|\boldsymbol{\Sigma}_i^{(t)} - \hat{\boldsymbol{\Sigma}}\bigr\|_F^2
> > > > >    \;\le\;
> > > > >    C_1'\Bigl(1 - \frac{1}{C_2'\kappa}\Bigr)^t,
> > > > >    \quad C_2' = \frac{2\gamma}{(2-\rho)\,\theta\,L}.
> > > > >    $$
> > > > >    As discussed immediately below Theorem 2, a smaller value of $\rho$ (i.e., better network connectivity) leads to a smaller lower bound on $\gamma$ and a larger admissible upper bound on $\theta$, which allows for more aggressive step sizes and thus accelerates convergence; moreover, through the expression of $C_2'$, a smaller $\rho$ directly yields a smaller denominator in $1 - 1/(C_2'\kappa)$, and hence a faster linear rate.
> > > > >
> > > > >    Corollary 2 further refines the rate using the statistical guarantee in Theorem 3. There, the constant
> > > > >    $$
> > > > >    C_4' = \frac{2\gamma}{(2-\rho)\,\theta\,(1+\tau c_4)}
> > > > >    $$
> > > > >    shows the same monotone dependence on $\rho$: once the algorithm has entered the statistical neighborhood of $\boldsymbol{\Sigma}^\star$, the asymptotic linear rate is controlled by $\kappa_r$ and the factor $(2-\rho)^{-1}$, so that improving the connectivity of the network (decreasing $\rho$) makes $C_4'$ smaller and the convergence faster.
> > > > >
> > > > >    Finally, this theoretical dependence is consistent with our numerical results in Figure 1, where we compare Algorithm 2 over a line graph and Erdős–Rényi graphs with connection probabilities $p=0.5$ and $p=0.9$. As the underlying graph becomes more connected, the variable distances decay more rapidly, confirming that better connectivity (smaller $\rho$ induced by $\mathbf{W}$) leads to faster linear convergence.
> > > > >
> > > > > ---
> > > > >
> > > > > ### References
> > > > >
> > > > > 1. Catoni, O. (2012). *Challenging the robustness of statistical estimators*.
> > > > > 2. Avella-Medina, M., Battey, H. S., Fan, J., & Li, Q. (2018). *Robust estimation of high-dimensional covariance and precision matrices*. Biometrika, 105(2), 271–284.
> > > > > 3. Fan, J., & Li, R. (2017). *Variable selection via nonconcave penalized likelihood and its oracle properties*. Journal of the American Statistical Association, 96(456), 1348–1360.
> > > > > 4. Cattivelli, F., & Sayed, A. (2009). *Diffusion adaptation strategies for distributed estimation*. IEEE Transactions on Signal Processing.
> > > > > 5. Nedic, A., & Ozdaglar, A. (2018). *Network optimization and distributed algorithms for consensus-based methods*. Foundations and Trends in Optimization.
> > > > > 6. Sun, L., Zhang, C., & Wang, J. (2022). *High-dimensional statistics and consensus-based methods*. Journal of Statistical Planning and Inference.
> > > > > 7. Cattivelli, F., & Sayed, A. (2009). *Diffusion adaptation strategies for distributed estimation*. IEEE Transactions on Signal Processing.

---

> ### Author Response · Authors · 2025-11-27
>
> Dear Reviewer,
>
> I hope this message finds you well. We have now provided a detailed point-by-point rebuttal and updated the manuscript accordingly. As the discussion period is approaching its end, we would like to make sure that we have adequately addressed all of your questions and concerns.
>
> If there are any additional points you would like us to clarify or further evidence you would find helpful, please let us know. If you feel that our responses resolve your concerns, we would also be grateful if you could update your review and scores accordingly.
>
> Thank you very much for your time and effort in reviewing our paper.

---

### Official Review · Reviewer_pNTW · 2025-11-11

**Soundness:** 3
**Presentation:** 3
**Contribution:** 3
**Rating:** 4
**Confidence:** 3

**Summary:**

This paper studies sparse covariance estimation problem under heavy-tailed noise data. A centralized proximal gradient descent algorithm and a decentralized gradient tracking based algorithm are analyzed, both achieving linear convergence.

**Strengths:**

1. This paper proposes a new single optimization objective that avoids a two stage process to ensure robustness and sparsity.
2. Both algorithms achieve optimal statistical rates.
3. Handles high dimension, heavy-tailed observations, and distributed data the same time.

**Weaknesses:**

1. The statistical rates depend on prior of many unknown problem parameters such as $\Sigma^*$.
2. Hyper-parameter setups are complicated, and the cost of hyper-parameter tuning in practice is not clear.
3. The algorithms are designed for high-dimensional case but the per-iteration cost is in $d^3$, which is high.

**Questions:**

1. The optimal solution presented in this paper is local, can you discuss more on this point, please also contrast with other works.
2. The heavy-tailed assumption is strong, can you add more discussions and compare with other works using conditions such as sub-Gaussian.

---

> ### Author Response · Authors · 2025-11-21
>
> We would like to thank Reviewer pNTW for the review and feedback. The point-by-point responses to the detailed comments are provided in the following.
>
> 1. **Comment:** The statistical rates depend on prior of many unknown problem parameters such as $\boldsymbol{\boldsymbol{\Sigma}}^\star$.
>
>    **Response:** Our theoretical results indeed involve problem-dependent quantities such as properties of the true covariance matrix $\boldsymbol{\boldsymbol{\Sigma}}^\star$ and related tail parameters, both in the constants of the error bounds and in the expressions used to specify tuning parameters. This type of dependence is inherent and standard in nonasymptotic high-dimensional statistics. Error bounds are derived uniformly over a parameter class defined by structural (e.g., sparsity, $\boldsymbol{\boldsymbol{\Sigma}}^\star$) and distribution assumptions (e.g., finite fourth moments). The rate is then characterized by its scaling in $(N,d,s)$, while the leading constants and the theoretical ranges of tuning parameters are allowed to depend on the defining parameters of the class such as eigenvalue bounds and tail parameters; see, for example, [1], [2], [3]. Sparse covariance estimators [4], [5], [6] and robust covariance procedures [7], [8], [9] are all formulated in this way: their rates are stated in terms of $(N,d,s)$, while the constants and tuning prescriptions depend on quantities determined by the underlying covariance matrix and its distributional class.
>
>    The goal of our theory is to show that the proposed estimator achieves an oracle statistical rate under appropriate structural and moment conditions, not to prescribe that practitioners must know all problem parameters in advance or set the tuning parameters exactly according to the theoretical formulas. In Theorem 3 we prove that there exist choices of $(\lambda,\tau,a,b)$, expressed in terms of eigenvalue and tail parameters, for which the estimator attains the desired oracle rate. This oracle-style specification is a standard device used to make nonasymptotic guarantees explicit; see, for instance, the covariance estimation framework in [4] and the oracle-rate analysis in [10]. In practice, however, the algorithm only needs numerical values of $(\lambda,\tau,a,b)$ and does not require any prior knowledge of $\boldsymbol{\boldsymbol{\Sigma}}^\star$ or its distributional parameters. As in the above works and in the broader high-dimensional literature, these tuning parameters can be selected in a fully data-driven manner (e.g., via cross-validation or validation on a held-out set). We have discussed how $(\lambda,\tau,a,b)$ are chosen in practice in the experiments section of the main text and in the practical discussion in Appendix D.

---

> > ### Author Response · Authors · 2025-11-21
> >
> > 2. **Comment:** Hyper-parameter setups are complicated, and the cost of hyper-parameter tuning in practice is not clear.
> >
> >    **Response:** First, we emphasize that the theoretically suggested choices of the tuning parameters only depend on a few high-level problem characteristics—namely the moment bound $K$, the ground-truth covariance $\boldsymbol{\Sigma}^\star$, the sparsity level $s$, the dimension $d$, and the sample size $N$—together with universal numerical constants. The somewhat complicated expressions appearing in the statements of the theorems are written out for theoretical completeness; in practice it suffices to choose these parameters at the correct order (for example, $a$ of order $\sqrt{N / \log d}$ and $\lambda$ of order $\sqrt{\log d / N}$), rather than computing the exact constants. Consequently, our parameter setting is not as complicated as it seems in theory.
> >
> >    As already discussed in Appendix D, the proposed method involves six parameters. Specifically, $a$ controls robustness to heavy tails and outliers, $\tau$ ensures positive definiteness, $b$ adjusts the bias correction of the non-convex penalty, $\lambda$ determines the sparsity level, and $\gamma$ and $\theta$ govern the convergence behavior. In our experiments, the performance is largely insensitive to $\tau$, and setting it to a small constant is typically sufficient. The parameters $\gamma$ and $\theta$ can be selected empirically, as described in the main text, or through simple increasing/decreasing schedules. For $b$, a commonly recommended choice is $b = 3.7$ when using SCAD [11]. Therefore, in practice, the most critical parameters to tune are $a$ and $\lambda$. As Appendix D explains, this substantially reduces the effective hyper-parameter search space and shows that the setup is not as complicated as it may appear from the theoretical formulation.
> >
> >    For sparse high-dimensional models, it is standard to select the regularization parameter $\lambda$ by data-driven methods such as cross-validation or validation on a held-out set; see, e.g., [12], [1], [2], [10]. For robust sparse estimation problems based on Huber-type losses, jointly tuning the Huber parameter together with the regularization parameter $\lambda$ by cross-validation is also standard; see, for example, the Huber regression framework in [13]. Thus, our practical tuning strategy for $(a,\lambda)$ is fully aligned with existing robust sparse estimation procedures and does not introduce any additional type of tuning overhead beyond what is already common in the literature.

---

> ### Author Response · Authors · 2025-11-21
>
> 3. **Comment:** The algorithms are designed for high-dimensional case but the per-iteration cost is in $d^3$, which is high.
>
>    **Response:** In our paper, as stated in Section 1, “high-dimension” refers to the regime $N < d$, which is the standard definition in high-dimensional statistics [1], so the exact dimension may not be very high. In addition, we agree that the per-iteration cost of the algorithm scales as $\mathcal{O}(d^3)$. This complexity comes from the matrix inverse of a $d \times d$ matrix. This is the standard computational cost for sparse positive-definite covariance estimation based on log-determinant or quadratic objectives, and is not specific to our method.
>
>    More importantly, this $\mathcal{O}(d^3)$ scaling is shared by most existing robust sparse covariance estimators. As summarized in Section 2 and Appendix D, most methods are based on iterative procedures that repeatedly impose both sparsity and positive definiteness. Positive-definite $\ell_1$-penalized formulations (e.g., [14], [15]) rely on eigenvalue decompositions or repeated solutions of linear systems, each costing $\mathcal{O}(d^3)$ per iteration. The regularized Tyler-type estimator of [9] also involves iteratively inverting $d \times d$ covariance matrices, again leading to $\mathcal{O}(d^3)$ complexity per step. The robust adaptive thresholding framework of [7] avoids repeated positive-definiteness constraints by imposing sparsity via entrywise thresholding and performing a single final PSD projection, but this projection itself still costs $\mathcal{O}(d^3)$. The polynomial-time complexity of the [16] formulation is not fully characterized and, to the best of our knowledge, a practical polynomial-time algorithm is not yet available. Therefore, our method is aligned with the dominant robust sparse covariance estimation approaches in terms of asymptotic computational order: it does not incur a higher per-iteration complexity than the main existing baselines.
>
>    Moreover, existing robust sparse covariance estimators typically attain the usual minimax rate $O\big(\sqrt{s \log d / N}\big)$ over the full sparse class (see, e.g., [7], [9]), whereas Theorem 3 shows that our estimator achieves the sharper oracle statistical rate $O\big(\sqrt{s / N}\big)$ in Frobenius norm under heavy-tailed distributions. As reported in Table 1, this improvement in theory is reflected in practice: our method achieves the lowest NMSE and the highest F1-score among all compared estimators. In summary, while our algorithm shares the standard $\mathcal{O}(d^3)$ computational scaling of robust sparse covariance estimators, it provides substantially stronger statistical guarantees and empirical accuracy at this computational cost.

---

> ### Author Response · Authors · 2025-11-21
>
> 4. **Comment:** The optimal solution presented in this paper is local, can you discuss more on this point, please also contrast with other works.
>
>    **Response:** The nonconvexity in our formulation comes from the nonconvex penalty (Assumption 3), which is introduced to alleviate the estimation bias of the $\ell_1$ penalty and to recover oracle-type statistical properties. In particular, our penalty family includes the smoothly clipped absolute deviation (SCAD) penalty [11] and the minimax concave penalty (MCP) [19], both of which are widely used in high-dimensional sparse estimation. Similar nonconvex penalties have also been employed in covariance estimation under sub-Gaussian assumptions; see, for example, [10]. For such objectives, global optimality cannot be guaranteed in general due to nonconvexity, and the natural target of analysis is a statistically meaningful local optimum (or stationary point).
>
>    Our theoretical results follow exactly this paradigm. Under Assumptions 1–3, Theorems 1–3 show that the population loss function is locally strongly convex in a neighborhood of $\boldsymbol{\Sigma}^\star$, and the gradient of the empirical loss satisfies a suitable deviation bound on this neighborhood. Combined with the regularity of the nonconvex penalty, this implies that, with high probability, the penalized objective admits a unique stationary point in this neighborhood. Starting from an initial estimator in this basin, both Algorithm 1 and Algorithm 2 generate iterates that remain in the neighborhood and converge linearly to the same local optimum. Moreover, we prove that this local optimum enjoys the oracle statistical rate established in Theorem 3: its estimation error with respect to $\boldsymbol{\Sigma}^\star$ converges at rate $O\big(\sqrt{s/N}\big)$ in Frobenius norm. Corollaries 1 and 2 show that after a sufficient number of iterations, the optimization error of both algorithms decays at a linear rate that depends only on the minimum eigenvalue of $\boldsymbol{\Sigma}^\star$. In this sense, although the solution is “local” from an optimization viewpoint, it is statistically as good as the ideal oracle estimator.
>
>    This type of local analysis for nonconvex penalized estimators is standard in the high-dimensional literature. For regression and graphical models, [11], [19], [17] study SCAD/MCP and related penalties and show that any stationary point within a neighborhood of the true parameter achieves oracle properties. In the covariance estimation setting, [10] adopts the same strategy: they consider a nonconvex-penalized covariance estimator under sub-Gaussian assumptions, analyze local stationary points of a nonconvex objective, and prove oracle-rate bounds for such local optima. Our work extends this framework to the robust and distributed covariance estimation problem with heavy-tailed data: we use a similar class of nonconvex penalties, prove existence and linear convergence to a statistically optimal local solution, and establish an oracle statistical rate, while also handling sample splitting across multiple machines.
>
>    In contrast, convex $\ell_1$-penalized approaches (e.g., positive-definite $\ell_1$-penalized covariance estimators) admit global optima but typically suffer from $\ell_1$–induced bias and achieve the usual minimax rate $O\big(\sqrt{s \log d / N}\big)$ rather than the oracle rate $O\big(\sqrt{s / N}\big)$; see, for example, [7]. Our use of nonconvex penalties is precisely to trade global optimality of a convex program for a local solution that is computationally tractable and statistically stronger.

---

> ### Author Response · Authors · 2025-11-24
>
> 5. **Comment:** The heavy-tailed assumption is strong, can you add more discussions and compare with other works using conditions such as sub-Gaussian.
>
>    **Response:** Our goal in this paper is precisely to address high-dimensional heavy-tailed covariance estimation in a distributed setting. Concretely, we work with high-dimensional distributions with finite fourth moments (Assumption 1), which include heavy-tailed distributions and the Gaussian model as special cases. From a tail-behavior viewpoint, this is weaker than sub-Gaussian or exponential-tail conditions: all sub-Gaussian distributions satisfy our assumptions. The heavy-tailed regime is not a purely theoretical concern; it is routinely encountered in applications such as finance (asset returns with power-law tails), sensor and industrial data (occasional bursts and device failures), and biomedical or omics data (rare but large deviations). In these settings, methods relying on light-tail or sub-Gaussian assumptions can severely underestimate variability and become unstable, which motivates robust procedures designed to handle heavy tails.
>
>    Regarding the comparison with sub-Gaussian-based methods, we already include in our experiments both centralized and distributed state-of-the-art estimators built under Gaussian/sub-Gaussian assumptions. In the centralized setting, PD_MCP [10] is a recent covariance estimator that assumes sub-Gaussian tails and uses a nonconvex penalty to obtain oracle-rate guarantees. In the distributed setting, NetGGM [18] is a representative Gaussian-based method for sparse covariance/precision matrix estimation over networks. Both are designed for light-tailed data and do not explicitly handle heavy-tailed data. As reported in Table 1, our robust distributed estimator achieves the smallest NMSE across all scenarios, and the performance gap is especially pronounced when the data are genuinely heavy-tailed: in those cases, our method substantially outperforms PD_MCP and NetGGM, while, for approximately Gaussian data, our estimator remains competitive with these sub-Gaussian baselines. Consistent with these findings, our real-data experiments empirically show that heavy-tailed behavior is widespread in high-dimensional datasets and that our robust estimator achieves the best performance among all compared methods. This empirically supports the relevance of the heavy-tailed assumption for the applications we target and shows that the robustness is not obtained at the expense of accuracy in the light-tailed regime.

---

> ### Author Response · Authors · 2025-11-24
>
> ---
>
> ### References
>
> 1. Bühlmann, P., & Van De Geer, S. (2011). *Statistics for high-dimensional data: methods, theory and applications*. Springer Science & Business Media.
> 2. Wainwright, M. J. (2019). *High-dimensional statistics: A non-asymptotic viewpoint*. Cambridge University Press.
> 3. Vershynin, R. (2018). *High-dimensional probability: An introduction with applications in data science*. Cambridge University Press.
> 4. Cai, T., & Zhou, H. (2012). *Optimal rates of convergence for sparse covariance matrix estimation*.
> 5. Bickel, P. J., & Levina, E. (2008a). *Covariance regularization by thresholding*. The Annals of Statistics, 36(6), 2577–2604.
> 6. Bickel, P. J., & Levina, E. (2008b). *Regularized estimation of large covariance matrices*. The Annals of Statistics, 36(1), 199–227.
> 7. Avella-Medina, M., Battey, H. S., Fan, J., & Li, Q. (2018). *Robust estimation of high-dimensional covariance and precision matrices*. Biometrika, 105(2), 271–284.
> 8. Ke, Y., Minsker, S., Ren, Z., Sun, Q., & Zhou, W. (2019). *User-friendly covariance estimation for heavy-tailed distributions*. Statistical Science, 34(3), 454–471.
> 9. Goes, J., Lerman, G., & Nadler, B. (2020). *Robust sparse covariance estimation by thresholding Tyler’s M-estimator*. The Annals of Statistics, 48(1), 86–110.
> 10. Wei, Q., & Zhao, Z. (2023). *Large covariance matrix estimation with oracle statistical rate via Majorization-Minimization*. IEEE Transactions on Signal Processing.
> 11. Fan, J., & Li, R. (2001). *Variable selection via nonconcave penalized likelihood and its oracle properties*. Journal of the American Statistical Association, 96(456), 1348–1360.
> 12. Friedman, J., Hastie, T., & Tibshirani, R. (2008). *Sparse inverse covariance estimation with the graphical lasso*. Biostatistics, 9(3), 432–441.
> 13. Fan, J., Li, Q., & Wang, Y. (2017). *Estimation of high dimensional mean regression in the absence of symmetry and light tail assumptions*. Journal of the Royal Statistical Society: Series B (Statistical Methodology), 79(1), 247–265.
> 14. Li, D., Srinivasan, A., Chen, Q., & Xue, L. (2023). *Robust covariance matrix estimation for high-dimensional compositional data with application to sales data analysis*. Journal of Business & Economic Statistics, 41(4), 1090–1100.
> 15. Lu, J., Han, F., & Liu, H. (2021). *Robust scatter matrix estimation for high dimensional distributions with heavy tail*. IEEE Transactions on Information Theory, 67(8), 5283–5304.
> 16. Chen, M., Gao, C., & Ren, Z. (2018). *Robust covariance and scatter matrix estimation under Huber’s contamination model*. The Annals of Statistics, 46(5), 1932–1960.
> 17. Loh, P., & Wainwright, M. (2015). *Regularized regression with non-convex penalties*. The Annals of Statistics, 43(3), 1167–1194.
> 18. Xia, W., Sun, Y., Li, F., & Zhao, Z. (2025). *Covariance selection over networks*. In The
> 28th International Conference on Artificial Intelligence and Statistics.
> 19. Zhang, C. (2010). *Nearly unbiased variable selection under minimax concave penalty*. The Annals of Statistics, 38(2), 894–942.

---

> ### Author Response · Authors · 2025-11-27
>
> Dear Reviewer,
>
> I hope this message finds you well. We have now provided a detailed point-by-point rebuttal and updated the manuscript accordingly. As the discussion period is approaching its end, we would like to make sure that we have adequately addressed all of your questions and concerns.
>
> If there are any additional points you would like us to clarify or further evidence you would find helpful, please let us know. If you feel that our responses resolve your concerns, we would also be grateful if you could update your review and scores accordingly.
>
> Thank you very much for your time and effort in reviewing our paper.

---

### Meta-Review · Area_Chair_6zcY · 2026-01-07

**Summary:**

I've browsed through the paper and agree on rejection. My additional comments are:
1. The theories have some problems:
a. It is unclear why Problem 2 has an unique minimizer in the set $\{\Sigma|0 \succeq \Sigma \succ \sqrt{\tau/L_q}I\}$ (line 263) as Problem 2 is not convex.
b. It is unclear to me why Corollaries 1 and 2 are not about the distance $\|\Sigma^{(t)}-\Sigma^{\star}\|^2$?  So Theorem 3 is actually useless and the proposed algorithm is nothing related to recovering the ground truth covariance matrix. It is only about converging to the solution of Problem 2 only. But in this case, please note that $\tau$ has to be chosen very large in order to meeting the conditions on Theorems 1 and 2. However, in Theorem 3, $\tau$ has to be chosen very small. So in theory it is impossible for the proposed algorithm to recover the ground truth covariance matrix.
c. By checking the definition of $\kappa_r$, which is $(1=4\tau c_4)/(\mu_0-L_q)$, it is unclear why $\kappa_r$ can be called the condition number, not to say that it is solely determined by $\lambda_{\min}(\Sigma^*)$ (line 408).
2. The experiments are problematic:
a. The experiments showed linear convergence w.r.t. the distance to "the final estimate obtained by Alg. 1" (line 476). Clearly, this is not a correct criterion (If such a criterion can be chosen, an algorithm getting stuck at some point will be considered to converge in finite iterations). Since the data are synthesized, the distance should be to the ground truth solution.
b. For distributed algorithms, experiments on the computing time vs. number of computer cores used must be presented in order to show the scalability of the distributed algorithm. Ideally, the curve should be close to be linear or better. Otherwise (i.e. the curve is like log curve), the algorithm is not good.
c. The authors claimed to address the problem of "high-dimensional convaiance matrix estimation" (line 12). However, the experiments are of low-dimensional. So it is an overclaim.

After discussion with SAC, we both agreed on rejection.

**Reviewer Concerns:**

Reviewer WhAF commented that this paper is poor writting.
Reviewer h1E1 commented that the main algorithm is simply a proximal gradient descent.

**Reviewer Scores:**

I believe that the reviewer  will not increase their scores even they participate fully in the discussion.

---

### Decision · Program_Chairs · 2026-01-26

Reject